# Hom-PGD$^+$: Fast Reparameterized Optimization over Non-convex Ball-Homeomorphic Set

**Chenghao Liu**[1] **Enming Liang**[1] **Minghua Chen**[1][2]

## Abstract

We study optimization over non-convex constraint sets that are homeomorphic to a ball, encompassing important problem classes such as star-shaped sets that frequently arise in machine learning and engineering applications. We propose **Hom-PGD$^+$**, a learning-based and projection-efficient first-order method that efficiently solves such problems without requiring expensive projection or optimization oracles. Our approach leverages an invertible neural network (INN) to learn the homeomorphism between the non-convex constraint set and a unit ball, transforming the original problem into an equivalent ball-constrained optimization where projections admit efficient solutions. We establish that Hom-PGD$^+$ achieves an $\mathcal{O}(\epsilon^{-2})$ convergence rate to an $(\epsilon + \mathcal{O}(\sqrt{\epsilon_{\mathrm{inn}}}))$-approximate stationary solution, where $\epsilon_{\mathrm{inn}}$ denotes the homeomorphism learning error. This rate significantly improves upon existing methods for optimization over non-convex sets, while maintaining a per-iteration complexity of only $\mathcal{O}(W)$ for $W$ INN parameters. Extensive experiments, including QCQP, chance-constrained power-system optimization, and non-uniform adversarial attacks, demonstrate that Hom-PGD$^+$ achieves competitive solution quality while delivering speedups of up to one order of magnitude.

## 1. Introduction

We consider a class of non-convex constrained optimization problems where the constraint set is homeomorphic to a unit ball, also known as *ball-homeomorphic* (BH) sets.

BH sets encompass any *compact convex set* and a class of *simply-connected non-convex* sets, such as star-shaped and geodesic-convex sets. This problem is fairly general and covers numerous optimization classes, including standard convex programming (Boyd et al., 2004), chance-constrained programming (Nemirovski & Shapiro, 2006; Pagnoncelli et al., 2009), and $\ell_p$-constrained regression (Xu et al., 2010; Jiang et al., 2016). These optimization problems naturally arise in real-world applications in machine learning and engineering, such as chance-constrained power grid optimization (Pagnoncelli et al., 2009) and non-uniform adversarial attacks in neural networks (Erdemir et al., 2021). While convex constrained optimization has been extensively studied and can be solved efficiently, this paper focuses on optimization over non-convex constraint sets, which present significant additional challenges.

Optimization over non-convex sets is highly challenging. Even establishing the feasibility of a general non-convex set can be *NP-hard* (Park & Boyd, 2017). Furthermore, in many real-time operational scenarios, one must repeatedly solve the same class of problems with varying parameters, introducing uncertainty and variability in a setting known as *parametric optimization* (Grancharova & Johansen, 2012). This scenario poses significant computational challenges.

Traditional approaches to non-convex constrained optimization include convex relaxation (Low, 2014a;b; Diamond et al., 2018; Anstreicher, 2012), reformulation-linearization (Sherali & Adams, 2013), and sequential convex approximation (Marks & Wright, 1978; Beck et al., 2010; Tran et al., 2013; Scutari et al., 2014). However, they may require expensive subproblems/oracles, can be conservative, or rely on structural assumptions not available for general BH constraints. Recent state-of-the-art works (Lin et al., 2022; Kume & Yamada, 2024; Ma et al., 2019) have proposed efficient alternatives under various structural assumptions with established convergence guarantees, yet challenges remain, including slower convergence rates, expensive per-iteration oracles, and reliance on strong assumptions.

*Reparameterization* has emerged as a powerful technique for transforming difficult optimization problems into more tractable forms. The core idea is to apply invertible or smooth mappings that preserve optimal solutions while alle-

---

[1]Department of Data Science, City University of Hong Kong [2]School of Data Science, The Chinese University of Hong Kong, Shenzhen. Correspondence to: Enming Liang <enming.cityu@gmail.com>.

*Proceedings of the 43$^{rd}$ International Conference on Machine Learning*, Seoul, South Korea. PMLR 306, 2026. Copyright 2026 by the author(s).

*Table 1.* Summary of parameterization or iterative methods for (non)-convex constrained optimization.

| Reference | Settings | | Key Assumption | Parameterization Techniques | Algorithm | Per-iteration Complexity | Convergence Rate |
|---|---|---|---|---|---|---|---|
| | Obj. | Ctr. | | | | | |
| (Li et al., 2023) | NC | Simplex | – | *Hadamard Transformation* | Perturbed RGD | $\mathcal{O}(n)$ | $\mathcal{O}(\epsilon^{-2})$ |
| (Chok & Vasil, 2025) | C | Simplex | – | | Cauchy-simplex | $\mathcal{O}(n)$ | $\mathcal{O}(\epsilon^{-1})$ |
| (Tang & Toh, 2024) | (N)C | Polyhedra | Full-rank constraints. | | RGD + PGD | RO + PO | N/A |
| (Liu et al., 2026) | C SC NC | Convex | Non-degeneracy. – – | *Gauge Mapping* | PGD over ball | $\mathcal{O}(n^2)$ + MO | $\mathcal{O}(\epsilon^{-1})$ $\mathcal{O}(\log \epsilon^{-1})$ $\mathcal{O}(\epsilon^{-2})$ |
| (Barber & Ha, 2018) | SC | NC | Small local concavity coefficients of constraints. | – | PGD | PO | $\mathcal{O}(\log \epsilon^{-1})$ |
| (Lin et al., 2022) | WC | WC | Certain non-singularity. Initial feasible points. | – | Proximal-point penalty method | SCOO | $\tilde{\mathcal{O}}(\epsilon^{-3})$ $\tilde{\mathcal{O}}(\epsilon^{-4})$ |
| (Barik et al., 2023) | IV SIV | IV | Contraction and triangle inequality w.r.t. invexity. | – | Invex PGD | Invex PO | $\mathcal{O}(\epsilon^{-1})$ $\mathcal{O}(\log \epsilon^{-1})$ |
| Theorem 1 | NC | NC | Ball-homeomorphic. | *Invertible Neural Network* | Bisected-PGD | $\mathcal{O}(W)$+MO | $\mathcal{O}(\epsilon^{-2})$ |

[1] **Abbreviations**: C = convex, NC = non-convex, WC = weakly convex, SC = strongly convex, IV = invex, SIV = strongly invex, Obj = objective, Ctr = constraint, GD = gradient descent, PGD = projected gradient descent, RGD = Riemannian gradient descent, SCOO = strongly convex optimization oracle, MO = membership oracle, PO = Projection oracle, RO = Retraction oracle.

[2] **Convergence rate**: Number of iterations for finding an $\epsilon$-approximate global optimum or stationary point for convex and non-convex optimization problems, respectively. Here $\tilde{\mathcal{O}}(\cdot)$ hides logarithmic factors.

[3] **Complexity**: Here $W$ denotes the size of the neural network we use to learn a homeomorphic mapping, referring to Sec. 3. In practice, we choose $W = \mathcal{O}(n^2)$ where $n$ is the problem size. Notably, Membership oracle (MO) enjoys the lowest complexity compared with other optimization-based oracles in general settings (Mhammedi, 2022).

viating difficulties arising from non-smoothness or complex constraints. This approach has been successfully applied in semidefinite programming (Cifuentes, 2021), low-rank optimization (Mishra et al., 2014; Ha et al., 2020), and risk minimization (Bah et al., 2022). Recent extensions address optimization over simplices (Li et al., 2023), polyhedra (Tang & Toh, 2024), and compact convex sets (Liu et al., 2026), as well as smoothing non-smooth objectives (Poon & Peyré, 2023) and modeling discrete data (Davis et al., 2024). However, most existing applications are confined to convex settings and require carefully hand-crafted transformations. See Table 1 for a summary of those methods.

For more complex non-convex constraints, recent work (Liang et al., 2024) proposes using invertible neural networks (INNs) (Papamakarios et al., 2021) to learn constraint reparameterizations. However, this line of work focuses on single-step projection in the transformed space to correct infeasible neural network predictions, rather than iteratively solving the original optimization problem from an initial point. Consequently, *convergence analysis*—a central concern for iterative optimization—remains unaddressed. We also refer readers to Appendix A for a more detailed discussion on reparameterization and non-convex constrained optimization.

Despite recent progress in constrained optimization, a fundamental research gap remains: *"Can we design an efficient algorithm for optimization over non-convex ball-homeomorphic sets that achieves fast convergence while avoiding expensive projection/optimization oracles?"*

In this work, we provide an affirmative answer by proposing **Hom-PGD$^+$**, an efficient method for solving *parametric* optimization over *non-convex BH* sets. As detailed in Sec. 3, it leverages an INN to learn a homeomorphism between a unit ball and the BH constraint set, reformulates the original non-convex problem into an equivalent ball-constrained one, applies projected gradient descent in the transformed space where projections admit efficient solutions, and maps the convergent solution back to the original domain.

We establish convergence and complexity guarantees for **Hom-PGD$^+$** in Sec. 4. It finds an $\epsilon + \mathcal{O}(\sqrt{\epsilon_{\mathrm{inn}}})$-stationary point in $\mathcal{O}(\epsilon^{-2})$ iterations, where $\epsilon_{\mathrm{inn}}$ denotes the INN learning error. This convergence rate improves upon existing first-order methods for optimization over non-convex sets (see Table 1); It achieves a per-iteration complexity of $\mathcal{O}(W)$, where $W$ is the number of INN parameters, and setting $W = \mathcal{O}(n^2)$ empirically performs well. This demonstrates the scalability of our method compared to existing approaches requiring expensive optimization oracles.

We evaluate **Hom-PGD$^+$** on non-convex QCQPs, chance-constrained power-grid optimization, and non-uniform adversarial attacks in Sec. 5. Across these settings, our method consistently improves computational efficiency over existing approaches, achieving faster convergence with lower per-iteration cost. Together, these results show that our INN-driven reparameterized optimization with learned homeomorphism provides a practical and theoretically grounded route for first-order optimization over non-convex ball-homeomorphic constraint sets.

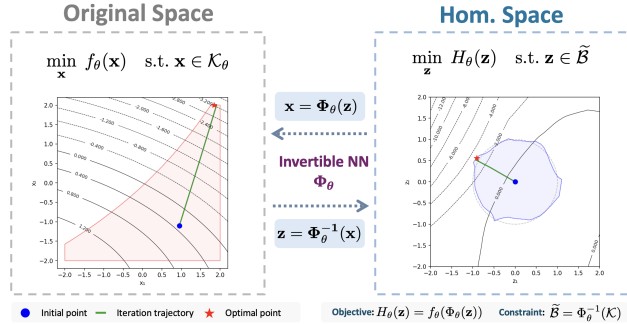

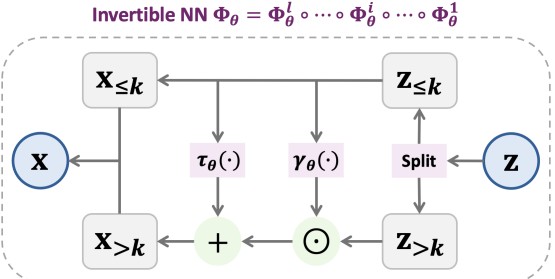

*Figure 1.* **Hom-PGD$^+$ Framework.** It applies projection-based GD methods in a transformed space via an INN-learned homeomorphism $\Phi_\theta(\cdot)$, where the transformed constraint set $\tilde{\mathcal{B}}$ is an approximated ball and $h_\theta$ is the transformed objective. The iterative trajectory is visualized in the transformed homeomorphic space and also mapped back to the original space for comparison.

*Figure 2.* **Illustration of a coupling layer in the INN.** The input is split as $\mathbf{z} = [\mathbf{z}_{\leq k}, \mathbf{z}_{>k}]$. The forward pass computes $\mathbf{x} = [\mathbf{z}_{\leq k}, \ \gamma_\theta(\mathbf{z}_{\leq k}) \odot \mathbf{z}_{>k} + \tau_\theta(\mathbf{z}_{\leq k})]$, and the inverse is $\mathbf{z} = [\mathbf{x}_{\leq k}, \ (\mathbf{x}_{>k} - \tau_\theta(\mathbf{x}_{\leq k}))/\gamma_\theta(\mathbf{x}_{\leq k})]$. Here $\gamma_\theta$ and $\tau_\theta$ are regular NNs (e.g., fully-connected) that take input parameter $\theta$ and $\mathbf{x}_{\leq k}$ to produce element-wise scaling and translation for $\mathbf{x}_{>k}$.

## 2. Problem Statement

We consider *parametric* constrained optimization problem:

$$\min_{\mathbf{x}} \ f_\theta(\mathbf{x}), \quad \text{s.t. } \mathbf{x} \in \mathcal{K}_\theta, \quad \text{(P)}$$

where $\mathbf{x} \in \mathbb{R}^n$ is the decision variable and $\theta \in \Theta \subseteq \mathbb{R}^d$ is the input parameters. The objective function $f_\theta(\cdot)$ is continuous and smooth, and the constraint set $\mathcal{K}_\theta \subset \mathbb{R}^n$ is compact. For ease of analysis and without loss of generality, we assume the constraint set $\mathcal{K}_\theta$ is defined by inequalities[1] as $\mathcal{K}_\theta = \{\mathbf{x} \in \mathbb{R}^n \mid \mathbf{g}_\theta(\mathbf{x}) \leq \mathbf{0}\}$ with $\mathbf{g}_\theta = (g_{1,\theta}, \cdots, g_{m,\theta})$, where $g_{i,\theta} : \mathbb{R}^n \to \mathbb{R}$ (for $i = 1, 2, \cdots, m$) are continuous functions. We further impose the following topological assumption on the constraint set $\mathcal{K}_\theta$.

**Assumption 1.** The set $\mathcal{K}_\theta$ is homeomorphic to a unit ball $\mathcal{B} = \{\|\mathbf{z}\|_2 \leq 1\}$, denoted as $\mathcal{K}_\theta \cong \mathcal{B}, \forall \theta \in \Theta$.

Homeomorphism (or homeomorphic mapping) is a bi-continuous bijection from two topological spaces, guaranteeing the topological equivalence. The non-convex BH constraint is fairly general, covering *a broad class of compact and simply-connected non-convex sets*[2], and many real-world applications as discussed in Sec.1.

While convex optimization has been extensively studied, optimization over non-convex constraint sets such as (P) remains challenging. Existing methods typically require strong regularity assumptions, or suffer from slow convergence rates, or incur high per-iteration costs due to expensive projection or optimization oracles. In this work, we exploit the ball-homeomorphic structure for non-convex constraints and propose an efficient first-order method with fast convergence while avoiding expensive projections.

[1]Certain equality constraints can be removed without loss of generality, see Appendix B.1 for discussions.

[2]For example, any contractible manifold with simply connected boundary over $\mathbb{R}^n$ for $n \geq 6$ (Smale, 1962).

## 3. Homeomorphic Optimization Approach

Motivated by reparameterization frameworks for solving optimization problems efficiently over *convex* sets (Li et al., 2023; Liu et al., 2026), we propose to transform the original *non-convex* problem (P) through a homeomorphism between the constraint set $\mathcal{K}_\theta$ and a unit ball $\mathcal{B}$, preserving the problem structure while simplifying the constraint set.

**Definition 3.1** (Homeomorphic Constrained Optimization (Liu et al., 2026)). Given a homeomorphism $\psi_\theta : \mathcal{B} \to \mathcal{K}_\theta$, we define the transformed parametric optimization problem with objective function $h_\theta(\mathbf{z}) = f_\theta(\psi_\theta(\mathbf{z}))$ and constraint set as a unit ball $\mathcal{B} = \psi_\theta^{-1}(\mathcal{K}_\theta)$ as:

$$\min_{\mathbf{z}} \ h_\theta(\mathbf{z}), \quad \text{s.t. } \mathbf{z} \in \mathcal{B}. \quad \text{(H)}$$

Under Assumption 1, we can transform any optimization problem (P) over a (non-convex) BH set into a simple ball-constrained program (H) via homeomorphism. Notably, under the homeomorphic transformation, the original problem and its homeomorphic counterpart are equivalent, i.e., there exists a bijective correspondence between their optimal solution sets $\mathbf{P}^*$ and $\mathbf{H}^*$, where $\mathbf{P}^* = \{\mathbf{x} \mid \mathbf{x} \in \arg\min\{(\mathbf{P})\}\}$ and similarly for $\mathbf{H}^*$. Specifically, for any $\mathbf{x} \in \mathbf{P}^*$, there exists a unique $\mathbf{z} \in \mathbf{H}^*$ such that $\mathbf{x} = \psi(\mathbf{z})$, and vice versa. Thus, we can solve the equivalent problem (H) without expensive projection to obtain the corresponding optimal solution of the original problem (P).

However, constructing homeomorphic transformations for general BH constraints remains non-trivial. Existing reparameterization methods rely on explicitly constructed transformations tailored to specific constraint structures. For instance, the *Hadamard transformation* (Li et al., 2023) maps a simplex to a sphere, while the *Gauge mapping* (Liu et al., 2026) transforms a compact convex set to a unit ball. Despite their success in specific settings, these methods face

fundamental limitations: (i) explicit or analytical homeomorphisms do not exist for general non-convex BH sets; and (ii) constructing different homeomorphisms for varying constraint sets incurs prohibitive computational overhead when input parameters change frequently, limiting their applicability in real-time operational settings.

To address these limitations, we propose **Hom-PGD$^+$**, illustrated in Fig. 1. Our method leverages an INN to transform the original non-convex constrained problem into a ball-constrained problem across varying input parameters (Sec. 3.1 and 3.2). We then apply projected gradient descent (PGD) to solve the reformulated problem (**H**). Under an exact homeomorphism, projection onto the unit ball admits a closed-form solution. In practice, however, the INN provides only an approximated homeomorphism, so we apply a bisection scheme to compute projections onto this approximated ball. Complete algorithmic details are provided in Alg. 1 and 2.

### 3.1. Homeomorphic Parameterization via INN

We utilize an invertible neural network (INN) to learn the homeomorphic mapping for general BH sets. An INN is a neural network $\Phi : \mathbb{R}^n \to \mathbb{R}^n$ whose inverse $\Phi^{-1}$ is both well-defined. To handle input-dependent homeomorphisms $\psi_{\boldsymbol{\theta}}$, we adopt conditional INNs (Winkler et al., 2019; Lyu et al., 2022), which treat the parameters $\boldsymbol{\theta}$ as additional inputs and learn augmented mappings $\Phi_{\boldsymbol{\theta}} : \mathcal{B} \to \mathcal{K}_{\boldsymbol{\theta}}$, where $\mathcal{K}_{\boldsymbol{\theta}}$ denotes the target constraint set under $\boldsymbol{\theta}$. For a detailed introduction to INNs, see Appendix B.2.

We employ standard coupling-layer INNs (Fig. 2) for their computational efficiency (Jin et al., 2024; Ishikawa et al., 2022; Lyu et al., 2022). Crucially, coupling-layer INNs can universally approximate any (differentiable) homeomorphism given sufficient depth (Jin et al., 2024; Ishikawa et al., 2022; Lyu et al., 2022), providing theoretical justification for learning the homeomorphic mapping.

### 3.2. INN Training for Obtaining the Homeomorphism

We now describe how to train an INN to approximate the homeomorphism between the BH constraint set and the unit ball. Following (Liang et al., 2023), we train an INN $\Phi_{\boldsymbol{\theta}}$ by *maximizing* the following loss function:

$$\mathcal{L}\left(\Phi_{\boldsymbol{\theta}}\right) = \widehat{V}\left(\Phi_{\boldsymbol{\theta}}(\mathcal{B})\right) - \lambda_1 P\left(\Phi_{\boldsymbol{\theta}}(\mathcal{B})\right) - \lambda_2 \widehat{L}\left(\Phi_{\boldsymbol{\theta}}\right) \quad (1)$$

where $\lambda_1, \lambda_2 > 0$ are balancing coefficients and the three terms are defined as follows:
▷ **Volume term**: $\widehat{V}\left(\Phi_{\boldsymbol{\theta}}(\mathcal{B})\right)$ is a computable approximation of the log-volume term $\log V\left(\Phi_{\boldsymbol{\theta}}(\mathcal{B})\right)$.
▷ **Penalty term**: $P\left(\Phi_{\boldsymbol{\theta}}(\mathcal{B})\right)$ is the penalty term for the constraint violation of $\Phi_{\boldsymbol{\theta}}(\mathcal{B}) \subseteq \mathcal{K}_{\boldsymbol{\theta}}$.
▷ **Lipschitz term**: $\widehat{L}\left(\Phi_{\boldsymbol{\theta}}\right)$ is a computable approximation of the log-Lipschitz term $\log L\left(\Phi_{\boldsymbol{\theta}}\right)$.

Details on computing these terms are provided in Appendix B.4. Intuitively, the first two terms encourage the transformed set to maximize volume while remaining within the BH constraint, thereby yielding the target homeomorphism. The third term regularizes the Lipschitz constant, which improves optimization performance in the subsequent stage (see Sec. 4.2 for formal analysis).

We then uniformly sample from a unit ball to prepare the training data for the loss function. Further, to train the INN for learning the homeomorphism under different $\boldsymbol{\theta}$, we uniformly sample input parameters $\{\boldsymbol{\theta}_i\}_{i=1}^N$ and train the INN following $\frac{1}{N} \sum_{i=1}^N \mathcal{L}(\Phi_{\boldsymbol{\theta}_i})$. After finite-sample training, the trained INN only approximates the homeomorphism, i.e., it does not perfectly map the constrained set to the unit ball, or vice versa. However, for our purposes on algorithm design in the next section, it suffices that the following validity condition holds:

**Definition 3.2** (Valid INN). The INN approximated mapping $\Phi_{\boldsymbol{\theta}}$ is valid for $\mathcal{K}_{\boldsymbol{\theta}}$ if $\Phi_{\boldsymbol{\theta}}(\mathbf{0}) \in \mathcal{K}_{\boldsymbol{\theta}}$, i.e., it maps the origin in the unit ball to a feasible point in $\mathcal{K}_{\boldsymbol{\theta}}$.

Theoretically, validity extends to all $\boldsymbol{\theta} \in \Theta$ given that it holds for a finite covering $\{\boldsymbol{\theta}_i\}_{i=1}^N$ (Liang et al., 2023). Empirically, we observe that validity is consistently satisfied across both training and test inputs, which is expected since the loss encourages the entire mapped set to be feasible $\Phi_{\boldsymbol{\theta}}(\mathcal{B}) \subseteq \mathcal{K}_{\boldsymbol{\theta}}$, while validity only requires the origin to map to a feasible point $\Phi_{\boldsymbol{\theta}}(\mathbf{0}) \in \mathcal{K}_{\boldsymbol{\theta}}$. If $\Phi_{\boldsymbol{\theta}}(\mathbf{0}) \notin \mathcal{K}_{\boldsymbol{\theta}}$, validity can be enforced via a shifted INN: $\Phi'_{\boldsymbol{\theta}}(\cdot) = \Phi_{\boldsymbol{\theta}}(\cdot) - \Phi_{\boldsymbol{\theta}}(\mathbf{0}) + \mathbf{x}^\circ$, where $\mathbf{x}^\circ \in \mathcal{K}_{\boldsymbol{\theta}}$ is an interior point. Such an interior/feasible point requirement for worst-case feasibility guarantees aligns with existing works on non-convex constrained optimization (Barber & Ha, 2018; Lin et al., 2022).

### 3.3. Hom-PGD$^+$: Projected Gradient Descent with INN

In the ideal setting with perfect homeomorphism, we perform regular PGD to problem (**H**) where the constrained set is a unit ball. However, in practice, due to the non-perfect training, the INN homeomorphic mapping is inexact, i.e., $\Phi_{\boldsymbol{\theta}} \neq \psi_{\boldsymbol{\theta}}$, thereby transforming $\mathcal{K}_{\boldsymbol{\theta}}$ into a non-perfect (and a non-convex)[3] ball $\tilde{\mathcal{B}} = \Phi_{\boldsymbol{\theta}}^{-1}(\mathcal{K}_{\boldsymbol{\theta}})$. To clarify the reformulated optimization problem we address, we denote the reformulated version induced by the INN as follows:

$$\min_{\mathbf{z}} \ H_{\boldsymbol{\theta}}(\mathbf{z}), \quad \text{s.t.} \ \mathbf{z} \in \tilde{\mathcal{B}}. \qquad (\mathbf{H}_{\mathrm{inn}})$$

where $H_{\boldsymbol{\theta}} = f_{\boldsymbol{\theta}} \circ \Phi_{\boldsymbol{\theta}}$. It is worth noting that the orthogonal projection onto the approximate ball $\tilde{\mathcal{B}}$ is computationally challenging. To mitigate this, we employ a bisection-based

---

[3]Here "non-perfect ball" means the learned ball $\tilde{B}$ is just an approximate ball, i.e., the shape is close to a unit ball, thus might exhibit non-convexity (e.g., see Fig. 1).

---

**Algorithm 1** Hom-PGD$^+$

**Input:** initial point $\mathbf{z}_0$, valid INN $\Phi_{\boldsymbol{\theta}}$, reformulated problem ($\mathbf{H}_{\text{inn}}$), and maximum iterations $K$
**for** $k = 0$ **to** $K$ **do**
  Compute stepsize $\alpha_k$ and gradient $\nabla H_{\boldsymbol{\theta}}(\mathbf{z}_k)$
  **Gradient descent:** $\mathbf{z}_{k+1} = \mathbf{z}_k - \alpha_k \nabla H_{\boldsymbol{\theta}}(\mathbf{z}_k)$
  **Projection:** $\mathbf{z}_{k+1} \leftarrow \text{BP}_{\tilde{\mathcal{B}}}(\mathbf{z}_{k+1})$ **if** $\Phi_{\boldsymbol{\theta}}(\mathbf{z}_{k+1}) \notin \mathcal{K}_{\boldsymbol{\theta}}$
**end for**
**Output:** $\mathbf{x}_K = \Phi_{\boldsymbol{\theta}}(\mathbf{z}_K)$

---

**Algorithm 2** BP Operator

**Input:** input point $\mathbf{z}$, lower bound $\beta_l = 0$, upper bound $\beta_u = 1$, and maximum iterations $B$
**for** $t = 0$ **to** $B$ **do**
  **Bisection:** $\beta_m = (\beta_l + \beta_u)/2$
  **Update:** $\begin{cases} \beta_l \leftarrow \beta_m, & \text{if } \Phi_{\boldsymbol{\theta}}(\beta_m \cdot \mathbf{z}) \in \mathcal{K}_{\boldsymbol{\theta}}, \\ \beta_u \leftarrow \beta_m, & \text{otherwise.} \end{cases}$
**end for**
**Output:** $\hat{\mathbf{z}} = \beta_l \cdot \mathbf{z}$

---

projection operator to approximate the orthogonal projection in each iteration, formally defined below.

**Definition 3.3** (Bisected projection). The bisected projection operator $\text{BP}_{\tilde{\mathcal{B}}}(\mathbf{z})$ for $\mathbf{z} \in \mathbb{R}^n$ is as $\text{BP}_{\tilde{\mathcal{B}}}(\mathbf{z}) \in$ segment$(\mathbf{oz}) \cap \partial\tilde{\mathcal{B}}$ for $\mathbf{z} \notin \tilde{\mathcal{B}}$ and $\text{BP}_{\tilde{\mathcal{B}}}(\mathbf{z}) = \mathbf{z}$ for $\mathbf{z} \in \tilde{\mathcal{B}}$, where $\mathbf{o}$ is the origin.

We note the following properties of the bisected projection operator: **(i)** The bisected projection may have multiple solutions when the line segment intersects the boundary $\partial\tilde{\mathcal{B}}$ at multiple points; in such cases, the operator returns one of the valid solutions. **(ii)** The projected solution can be computed efficiently using bisection methods (Alg. 2) with linear convergence rate (Liang et al., 2023). Importantly, each bisection iteration requires a simple feasibility check (i.e., membership oracle queries). **(iii)** When the trained INN satisfies validity conditions (Def. 3.2), the composition $\Phi_{\boldsymbol{\theta}}(\text{BP}_{\tilde{\mathcal{B}}}(\mathbf{z}))$ guarantees feasibility in $\mathcal{K}_{\boldsymbol{\theta}}$ for any $\mathbf{z} \in \mathbb{R}^n$.

We then apply the PGD with the bisection projection operator for the INN-transformed problem ($\mathbf{H}_{\text{inn}}$) (shown in Alg. 1). Finally, we map the obtained converged solution back to the original space to recover the corresponding solution for the original problem.

## 4. Performance Analysis

In this section, we present a comprehensive performance analysis for Hom-PGD$^+$, including the landscape analysis, convergence rate, and run-time complexity.

**General Assumptions and Notations** (detailed in Appendix C.2): For notational simplicity, we fix the input parameter $\boldsymbol{\theta}$ and omit it, writing $f$ in place of $f_{\boldsymbol{\theta}}(\cdot)$, and similarly for other functions and mappings. We assume that:

- The objective $f$ and each constraint function $g_i$ ($i \in [m]$) are $L_{f,0}$-Lipschitz ($L_{g_i,0}$ resp.) continuous, and $L_f$-smooth ($L_{g_i}$ resp.).
- The homeomorphic mapping $\psi$ is invertible, bi-Lipschitz continuous, and has a non-singular, Lipschitz continuous Jacobian matrix, denoted by $\mathbf{J}_{\psi}$.

We remark that the learned INN $\Phi$ inherently satisfies the same assumptions as $\psi$, including bi-Lipschitz continuity over compact domains and the existence of the Jacobian, by the design of the INN architectures (Appendix B.3). Moreover, the composited function $H = f \circ \Phi$ and $G_i = g_i \circ \Phi$ for $i \in [m]$ inherit the same regularity properties as $f$ and $g_i$. Thus, by Lemma D.1, we can make the following assumptions without loss of generality:

- The learned INN is $(l_\Phi, u_\Phi)$-bi-Lipschitz continuous and $L_\Phi$-smooth.
- The composited functions $H = f \circ \Phi$ and $G_i = g_i \circ \Phi$ ($i \in [m]$) are $L_{H,0}$-Lipschitz ($L_{G_i,0}$ resp.) continuous, and $L_H$-smooth ($L_{G_i}$ resp.).

Furthermore, we make assumptions on the training errors of the INN, which is essential for convergence analysis:

**Assumption 2** (INN Approximation Errors). We assume the INN-approximated homeomorphism $\Phi : \mathbb{R}^n \to \mathbb{R}^n$ has (i) a bounded approximation error: $\mathcal{B}(0, 1 - \epsilon_{\text{inn}}) \subseteq \Phi^{-1}(\mathcal{K}) \subseteq \mathcal{B}(0, 1 + \epsilon_{\text{inn}})$, $\|\psi - \Phi\| \le \epsilon_{\text{inn}}$, and (ii) a bounded Jacobian approximation error: $\|\mathbf{J}_{\psi} - \mathbf{J}_{\Phi}\| \le \epsilon_{\text{inn}}$.

**Remarks** Assumption 2 formalizes the approximation quality required from the learned INN $\Phi$ to the target homeomorphism $\psi$ with $\mathcal{K} = \psi(\mathcal{B})$. At the set level, it requires the learned image $\Phi(\mathcal{B})$ to be close to the true feasible set $\mathcal{K}$, which can be measured through a Hausdorff-distance condition. At the map level, it requires the induced function and Jacobian approximation errors to be controlled, since these errors directly enter the KKT residual and convergence bound.

This assumption is plausible for two reasons. First, recent universality results for INNs show that, with sufficient depth and width, INNs can approximate smooth diffeomorphisms arbitrarily well under Sobolev-type norms (Jin et al., 2024; Ishikawa et al., 2022; Lyu et al., 2022). Such approximation controls both the function approximation error and the Jacobian approximation error, providing a theoretical basis for the error terms in Assumption 2. Second, our training objective in Eq. (1) is designed to learn a map from the unit

ball $\mathcal{B}$ to the constraint set $\mathcal{K}$, so the Hausdorff-distance condition can be encouraged and empirically checked through sampled feasibility and coverage tests. Therefore, the assumption serves as the technical bridge between INN training quality and the convergence guarantee of Hom-PGD$^+$. Refer to Appendix E.1 for a detailed discussion.

## 4.1. Landscape Analysis

In this subsection, we analyze the landscape of (H) under the homeomorphic transformation. The following lemma establishes a one-to-one correspondence between KKT stationary points (Def. D.2) of (P) and (H), where the relevant definitions and the proofs are provided in Appendix D.3.

**Proposition 4.1.** *Suppose the strict complementary condition holds for both problem* (P) *and* (H)*. Then* $\mathbf{x}^*$ *is a first-order, second-order and non-degenerate KKT stationary point of* (P) *if and only if* $\mathbf{z}^*$ *is a corresponding KKT stationary point of* (H) *where* $\mathbf{x}^* = \boldsymbol{\psi}(\mathbf{z}^*)$.

The significance of this proposition lies in its ability to establish a fundamental equivalence between the solution properties of two distinct formulations of an optimization problem. Specifically, it guarantees that the optimality conditions under the Karush-Kuhn-Tucker (KKT) framework are preserved under a homeomorphism.

## 4.2. Convergence Analysis

We present the convergence guarantees for Hom-PGD$^+$, with proofs deferred to Appendix E.2.

**Theorem 1** (Convergence of Hom-PGD$^+$)**.** *Let INN* $\Phi$ *satisfy Assumption 2. Then Hom-PGD$^+$ with constant step-size* $\alpha \in (0, \frac{1}{L_H}]$ *can find an* $\epsilon + \mathcal{O}\left(\sqrt{L_H \epsilon_{inn}}\right)$*-approximate KKT stationary point (Def. E.1) for problem* (P) *in* $\mathcal{O}\left(L_H \epsilon^{-2}\right)$ *iterations.*

To contextualize the significance of this result, we compare it against established complexity bounds over convex and non-convex constraint sets. For non-convex optimization over **convex constraints**, established methods like PGD and augmented-Lagrangian approaches (Beck, 2014; Zhang et al., 2022; Liu et al., 2026) achieve $\mathcal{O}(\epsilon^{-2})$ rates. Under perfect INN training ($\epsilon_{\text{inn}} = 0$), our result recovers this result. The additional $\mathcal{O}\left(\sqrt{L_H \epsilon_{\text{inn}}}\right)$ term reflects INN approximation error and is consistent with analyses of optimization under inexact oracles (Devolder et al., 2014; Barber & Ha, 2018; Liu et al., 2025).

Optimization over **non-convex constraints** is substantially harder. Existing PGD-type methods require restrictive assumptions such as bounded local concavity (Barber & Ha, 2018), hidden convexity (Barik et al., 2023; Fatkhullin et al., 2023), or specialized manifold structure (Balashov et al., 2020). Proximal-point-based algorithms (Boob et al., 2019;

Ma et al., 2019; Lin et al., 2022) achieve $\tilde{\mathcal{O}}(\epsilon^{-3})$ complexity under non-singularity assumptions and $\tilde{\mathcal{O}}(\epsilon^{-4})$ without them.

Our key insight is that the BH structure closes this complexity gap. Although $\mathcal{K}$ may be highly non-convex, the homeomorphic mapping $\Phi$ transform the problem into an (approximately) ball-constrained problem, enabling the use of convex optimization techniques in the transformed space. Theorem 1 thus achieves the convex-constraint rate of $\mathcal{O}(\epsilon^{-2})$ for non-convex constrained problems–a significant theoretical improvement. Moreover, since $L_H = u_\Phi^2 L_f + L_\Phi L_{f,0}$ depends on the forward Lipschitz constant $u_\Phi$ of the INN (Lemma D.1), the Lipschitz-regularized training scheme in Section 3.2 can improve the convergence rate by a constant factor. Therefore, the BH structural assumption is both practically motivated by real-world applications and analytically useful: it enables convex-constraint-style iteration complexity guarantees for a broad class of non-convex constrained problems.

## 4.3. Run-time Complexity

We analyze the run-time complexity of Hom-PGD$^+$. Since INN training is performed offline as a one-time cost, it does not affect real-time performance; we defer its analysis to Appendix B.5. Once trained, the mapping $\Phi_{\boldsymbol{\theta}}$ can be directly applied for any given parameter $\boldsymbol{\theta}$. Below, we focus on online complexity.

**Oracles.** Hom-PGD$^+$ relies on the *Membership oracle* besides regular zeroth- and first-order information oracles. The *Membership oracle* $\mathcal{M}_\mathcal{K}(\mathbf{x}) : \mathbb{R}^n \to \{0, 1\}$ returns 1 if and only if $\mathbf{x} \in \mathcal{K}$. The membership oracle is generally more efficient than optimization oracles (Mhammedi, 2022), particularly for non-convex constraint sets.

**Basic operations.** Let $W$ denote the size of the INN (i.e., trainable parameter count). The complexity of each basic operation is as follows (see Appendix B.3 for details):

**(i)** *Computing* $\text{BP}_{\tilde{\mathcal{B}}}(\cdot)$*:* $\tilde{\mathcal{O}}(W \log(1/\epsilon))$. This bisection-based operator is computed via Alg. 2, which converges linearly (Liang et al., 2023). Each iteration requires one forward pass through the INN and 1 membership oracle queries for $\mathcal{K}_{\boldsymbol{\theta}}$.

**(ii)** *Gradient of* $h$*:* $\mathcal{O}(W)$. By the chain rule, $\nabla h(\mathbf{z}) = \text{J}_\Phi(\mathbf{z})^\top \nabla f(\mathbf{x})$. The Jacobian $\text{J}_\Phi$ is obtained via backpropagation at a cost of $\mathcal{O}(W)$.

**Total run-time complexity.** Given a trained INN $\Phi$, the online cost of Hom-PGD$^+$ consists of the following components.

**(i)** *Per-iteration cost:* $\tilde{\mathcal{O}}(W)$. Each iteration evaluates the transformed gradient $\nabla h(\mathbf{z}) = \text{J}_\Phi(\mathbf{z})^\top \nabla f(\mathbf{x})$ and computes the bisected projection. Both operations require INN

forward/backward evaluations and membership-oracle calls, leading to a per-iteration cost of $\tilde{\mathcal{O}}(W)$, where $W$ denotes the number of INN parameters.

**(ii)** *Final mapping:* $\mathcal{O}(W)$. After convergence, the solution in the transformed space is mapped back to the original domain by one forward pass through $\Phi$.

**(iii)** *Iteration count:* I. The iteration count is governed by the convergence guarantee in Section 4.2.

Therefore, the total online complexity is $\tilde{\mathcal{O}}(W\mathrm{I})$, plus a one-time final mapping cost $\mathcal{O}(W)$.

**Remarks** The above bound is stated in terms of the INN size $W$ and is independent of the offline training cost. In our implementation, we use a three-layer INN with width $\mathcal{O}(n)$, which gives $W = \mathcal{O}(n^2)$. This architectural choice is empirically supported by the approximation-quality results in Section 5 and Appendix G.1, where such INNs achieve small set-approximation errors and high feasibility rates across the tested constraint families. Thus, under the architecture used in our experiments, Hom-PGD$^+$ has online per-iteration cost $\tilde{\mathcal{O}}(n^2)$. This scaling is favorable compared with second-order methods with $\mathcal{O}(n^3)$ per-iteration costs, while retaining the practical expressiveness needed to approximate the considered BH constraint sets.

### 4.4. Discussion and Remarks

**Beyond ball-homeomorphic constraints**. While we assume the constraint set is homeomorphic to a ball, our framework can, in principle, be extended to general compact non-convex sets with non-empty interior, albeit with a potentially large worst-case optimality gap due to INN approximation errors. Specifically, we have the following key observations. **(i)** For non-BH constrained sets, one can still train an INN to learn an invertible mapping from the unit ball to a ball-homeomorphic *subset* of the constraint set, ideally, the largest subset via volume maximization in the loss function in Sec 3.2. This INN still yields a reformulated problem ($\mathbf{H}_{\text{inn}}$) equivalent only to a restricted version of the original problem (Lee et al., 2019). **(ii)** The Hom-PGD$^+$ (Alg. 1) can still be applied to the reformulated problem ($\mathbf{H}_{\text{inn}}$) under the valid INN condition. **(iii)** The convergence rate of Theorem 1 still holds, but the stationary point corresponds to the restricted problem over the subset. Consequently, the optimality gap with respect to the original problem cannot be directly quantified.

## 5. Empirical Study

We conduct extensive experiments to demonstrate the efficiency of Hom-PGD$^+$. **(i)** We evaluate Hom-PGD$^+$ on quadratically constrained quadratic programming (QCQP) problems. **(ii)** We scale the QCQP problem dimension and

compare Hom-PGD$^+$ with IPOPT on scalability. **(iii)** We consider real-world power grid optimization under uncertainty with joint chance constraints (JCC). **(iv)** We examine non-uniform adversarial attacks, which induce non-convex perturbation sets but admit a closed-form homeomorphic mapping. **(v)** We conduct ablation studies including INN complexity and optimality gaps. Detailed experimental settings, problem formulation, data generation, baseline description, and supplementary results are provided in Appendices F and G.

**Baselines**: For non-convex constrained optimization problems, we consider the following baselines following the state-of-the-art work considering optimization over non-convex constrained sets (Lin et al., 2022). (i) **EPM** (Cartis et al., 2011): *exact penalty methods* iteratively solve subproblems by adding a penalty for constraint violations to the objective. (ii) **ALM** (Sahin et al., 2019; Xie & Wright, 2019; Birgin et al., 2003): *augmented Lagrangian methods* for problem **P** that alternately update primal and dual variables for an unconstrained Lagrangian formulation. (iii) **PPP** (Lin et al., 2022): *proximal-point penalty method* iteratively solves subproblems by augmenting the objective with a proximal term and quadratic penalty terms. (iv) **Hom-PGD$^+$** shown in Sec. 3.

### 5.1. Illustrative Examples over Non-convex QCQP

As shown in Fig. 3, in the randomly generated non-convex QCQP instances, our Hom-PGD$^+$ method achieves fast convergence compared to other first-order algorithms. In terms of running time, compared to methods requiring expensive inner minimization problems such as Lagrangian or proximal-point methods, we only need bisection to project infeasible solutions back to the transformed constraint set, reaching linear convergence with low complexity through membership oracle queries.

We train one INN to transform the constraint set under different input parameters and deploy it for optimization, *amortizing* the homeomorphism construction complexity across different constraints and reducing online complexity. Furthermore, our method empirically works for non-BH constraint settings as long as the valid INN conditions hold, despite lacking tight theoretical bounds on INN approximation error $\epsilon_{\text{inn}}$, as discussed in Sec. 4.4.

### 5.2. Scaling tests of Hom-PGD$^+$ *vs* IPOPT

We scale our method to high-dimensional QCQP problems (which may be non-homeomorphic) along two axes: the number of decision variables $n$ and the number of quadratic constraints $m$, yielding $\mathcal{O}(n^2 \cdot m)$ problem parameters. Hom-PGD$^+$ demonstrates superior scaling compared to the well-optimized second-order industrial solver IPOPT. As $m$ increases by two orders of magnitude ($10 \rightarrow 1000$), IPOPT's

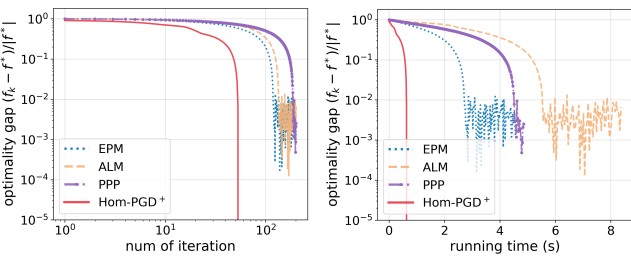

(a) Convergence with respect to iterations and running time.

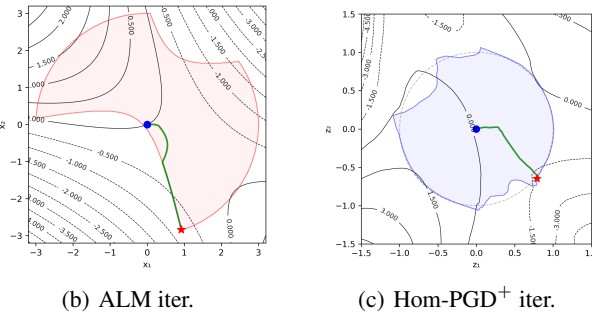

(b) ALM iter.  (c) Hom-PGD$^+$ iter.

*Figure 3.* Illustrative examples of Hom-PGD$^+$ for solving QCQP. The optimality gap is evaluated over the IPOPT solver. One INN is trained to map the unit ball to the constraint set under different input parameters (detailed in Appendix G.1). Performance comparisons under various inputs, including non-convex BH and non-BH constraints, are included in the Appendix G.3.

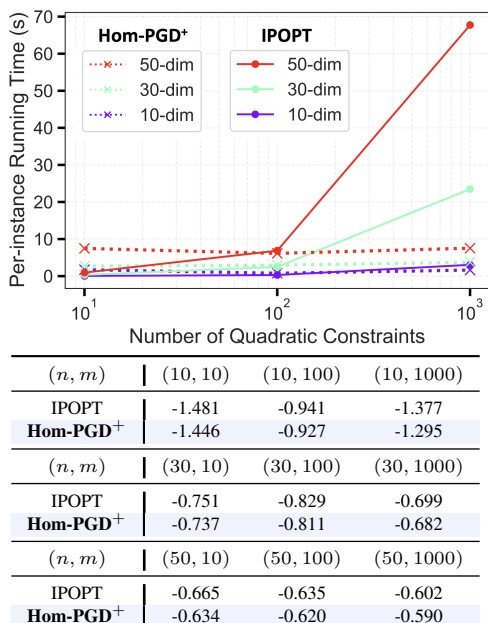

| $(n, m)$ | $(10, 10)$ | $(10, 100)$ | $(10, 1000)$ |
|---|---|---|---|
| IPOPT | -1.481 | -0.941 | -1.377 |
| **Hom-PGD$^+$** | -1.446 | -0.927 | -1.295 |
| $(n, m)$ | $(30, 10)$ | $(30, 100)$ | $(30, 1000)$ |
| IPOPT | -0.751 | -0.829 | -0.699 |
| **Hom-PGD$^+$** | -0.737 | -0.811 | -0.682 |
| $(n, m)$ | $(50, 10)$ | $(50, 100)$ | $(50, 1000)$ |
| IPOPT | -0.665 | -0.635 | -0.602 |
| **Hom-PGD$^+$** | -0.634 | -0.620 | -0.590 |

*Figure 4.* Scalability analysis of Hom-PGD$^+$ with constraints $m \in \{10, 100, 1000\}$ and variables $n \in \{10, 30, 50\}$. Problem dimensions scale as $\mathcal{O}(m \cdot n^2)$. The figure shows average per-instance solving time; the table reports average converged objective values.

per-instance time grows steeply—most notably for $n = 50$, where runtime jumps from 3 to 70 seconds. In contrast, Hom-PGD$^+$ exhibits near-constant runtime as $m$ scales and only mild growth with $n$, owing to efficient GPU-accelerated INN computation and batched constraint verification. Solution quality remains competitive: Hom-PGD$^+$ achieves an average objective gap of 2.9% on average with zero constraint violations. These results demonstrate that Hom-PGD$^+$ maintains efficiency as problem size grows, while IPOPT's computational cost escalates rapidly, particularly for large $n$ and $m$.

### 5.3. Chance-Constrained Optimization for Power Grid

Modern power grids face uncertainties from renewable generation and load fluctuations, requiring generator settings that ensure safe operation with high probability. This leads to non-convex joint chance-constrained (JCC) formulations, which are computationally prohibitive for large-scale grids when solved exactly via mixed-integer programming (Pagnoncelli et al., 2009), as integer variables and operational constraints scale with both scenarios and grid size.

Our method demonstrates strong performance on this challenging problem. As shown in Table 2, we significantly outperform baselines in running time while maintaining approximately 3% optimality gap compared to GUROBI

and achieving exact chance constraint satisfaction. This efficiency stems from our bisection-based projection algorithm requiring only function evaluation (membership oracle) without gradient calculations for constraints, unlike other first-order methods that require both evaluations at each iteration, with computational burden growing linearly with scenarios.

### 5.4. Non-Uniform Adversarial Attack

We next test Hom-PGD$^+$ on non-uniform adversarial attacks, where the inner maximization searches for perturbations under feature-dependent budgets $\Delta(\mathbf{x}) = \{\boldsymbol{\delta} : \|\mathbf{W}\boldsymbol{\delta}\|_p \leq \epsilon_p, \ 0 \leq \mathbf{x} + \boldsymbol{\delta} \leq 1\}$. Here $\mathbf{W}$ encodes channel-spatial weights; for $0 < p < 1$, the feasible region is non-convex, making projection-based PGD difficult, while Hom-PGD$^+$ uses a closed-form homeomorphic parameterization from the unit ball to preserve feasibility. We evaluate on CIFAR-10 with a trained CNN and $N = 500$ correctly classified samples, comparing against the PGD baseline in (Erdemir et al., 2021) that uses radial feasibility scaling instead of exact Euclidean projection. Detailed settings and more visualizations are in Appendix G.4.

Table 3 reports the quantitative results, and Fig. 5 provides qualitative examples on CIFAR-10. Hom-PGD$^+$ is most effective in the non-convex regimes: for $p = 0.5$, it increases the attack success rate from 2.0% to 14.6%, and

*Table 2.* Performance comparison over JCC optimal power flow.

| Case | Method | 100 scen. | | | 1000 scen. | | |
|---|---|---|---|---|---|---|---|
| | | Obj. | Vio. | Time | Obj. | Vio. | Time |
| **200-bus** | MIP | 0.679 | 0 | 95 | *failed* | | |
| | EPM | 0.690 | 0.9 | 76 | 0.933 | 1 | 801 |
| | ALM | 0.693 | 0.9 | 141 | 0.927 | 1 | 1452 |
| | PPP | 0.698 | 0.9 | 75 | 0.927 | 1 | 799 |
| | **Hom-PGD$^+$** | 0.688 | 0 | **44** | 0.768 | 0 | **246** |
| **500-bus** | MIP | 7.43 | 0 | 1259 | *failed* | | |
| | EPM | 8.63 | 1 | 109 | 8.65 | 1 | 1107 |
| | ALM | 8.66 | 1 | 205 | 8.67 | 1 | 2061 |
| | PPP | 8.62 | 1 | 108 | 8.66 | 1 | 1102 |
| | **Hom-PGD$^+$** | 7.66 | 0 | **103** | 8.56 | 0 | **396** |

[1] Performance comparison over JCC optimal power flow on PGLIB 200- and 500-bus systems with 100 and 1000 uncertainty scenarios.

[2] (Obj., Vio., Time) denote the objective value, chance-constraint excess violation, and inference time (in seconds), respectively.

[3] GUROBI is applied to compute the optimum with equivalent mixed-integer programming (MIP) formulations in 3,600 seconds. All first-order baseline methods are executed in a maximum of 100 iterations.

for $p = 1$, from 2.8% to 16.4%. For the convex weighted $\ell_2$ and $\ell_4$ regions, Hom-PGD$^+$ performs comparably to PGD. Across all settings, the per-iteration time remains nearly unchanged, showing that the homeomorphic reparameterization improves attack generation when projection is challenging while adding negligible computational overhead.

### 5.5. Ablation Study and Sensitivity Analysis

With details in Appendix G.2, we analyze: **(i)** *INN Complexity*: varying INN depth (1/3/5 layers) affects approximation error, Lipschitz constants, and downstream optimization, with 3 layers achieving the best trade-off. **(ii)** *Bisection Iterations*: fewer iterations reduce per-iteration cost but may increase the optimality gap.

## 6. Conclusion and Limitations

In this work, we proposed Hom-PGD$^+$, a fast projection-efficient for optimizing over non-convex constraint sets homeomorphic to a ball. Exploiting the topological structure of constraint sets, we leverage INN as a homeomorphism to transform the problem into a (approximately) ball-constrained optimization problem where projections admit efficient solutions. We prove that Hom-PGD$^+$ achieves an $\mathcal{O}(\epsilon^{-2})$ convergence rate to an $(\epsilon + \mathcal{O}(\sqrt{\epsilon_{\text{inn}}}))$-approximate stationary solution, where the rate significantly improves upon existing methods for optimization over non-convex sets. We evaluate **Hom-PGD$^+$** on non-convex QC-QPs, chance-constrained power-grid optimization, and non-

*Table 3.* Non-uniform adversarial attacks on CIFAR-10 under weighted $\ell_p$ constraints.

| $p$ | Method | Adv. acc. ↓ (%) | Success ↑ (%) | Time/iter ↓ (s) |
|---|---|---|---|---|
| **0.5** | PGD | 98.0 | 2.0 | 0.04195 |
| | Hom-PGD$^+$ | 85.4 | 14.6 | 0.04242 |
| **1** | PGD | 97.2 | 2.8 | 0.04138 |
| | Hom-PGD$^+$ | 83.6 | 16.4 | 0.04237 |
| **2** | PGD | 79.4 | 20.6 | 0.04141 |
| | Hom-PGD$^+$ | 80.0 | 20.0 | 0.04239 |
| **4** | PGD | 78.6 | 21.4 | 0.04133 |
| | Hom-PGD$^+$ | 79.2 | 20.8 | 0.04229 |

[1] Results use a shared fixed subset of $N = 500$ correctly classified CIFAR-10 samples. PGD uses radial feasibility scaling for its projection step; full experimental details and qualitative examples are in Appendix G.4.

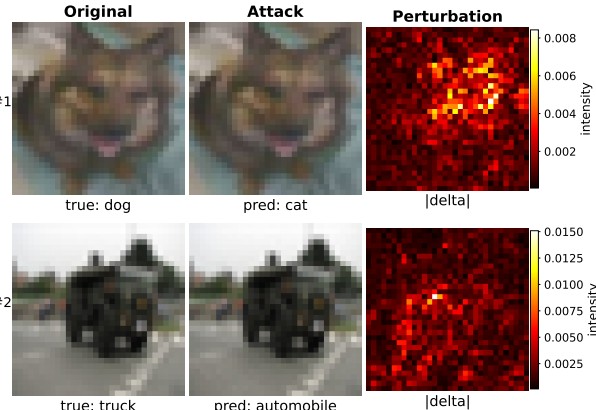

*Figure 5.* Qualitative adversarial examples on CIFAR-10 under weighted non-uniform $\ell_{1/2}$ perturbation constraints. It shows sampled successful attacks for the corresponding method.

uniform adversarial attacks in Section 5, where it consistently achieves faster convergence and lower per-iteration cost than existing methods. These results demonstrate that INN-based reparameterization with learned homeomorphisms offers a practical and theoretically grounded approach to first-order optimization over non-convex ball-homeomorphic constraint sets.

There are several **limitations** remain for future work: (i) INN-based homeomorphism learning theoretically incurs high worst-case complexity. Developing tighter bounds for learning homeomorphisms could improve practical efficiency. (ii) Our convergence guarantee yields an $(\epsilon + \mathcal{O}(\sqrt{\epsilon_{\text{inn}}}))$-approximate stationary point. This square-root dependence for INN errors $\epsilon_{\text{inn}}$ may be suboptimal. (iii) While designed for Euclidean ball-homeomorphic constraints, our framework may also extend to manifold-constrained problems with favorable topology conditions.

## Acknowledgments

This work is supported in part by a General Research Fund from Research Grants Council, Hong Kong (Project No. 11200124), a Collaborative Research Fund from Research Grants Council, Hong Kong (Project No. C1049-24G), an InnoHK initiative, The Government of the HKSAR, Laboratory for AI-Powered Financial Technologies, a Shenzhen-Hong Kong-Macau Science & Technology Project (Category C, Project No. SGDX20220530111203026), and a Start-up Research Grant from The Chinese University of Hong Kong, Shenzhen (Project No. UDF01004086). The authors would also like to thank the anonymous reviewers for their helpful comments.

## Impact Statement

This paper presents work aimed at advancing the field of Machine Learning. There are many potential societal consequences of our work, none of which we feel must be specifically highlighted here.

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

# Contents

# A. Related Work

Non-convex optimization is notoriously challenging and is NP-hard in general. To better understand its structure and design more efficient algorithms, researchers have explored strong structural assumptions that enable convergence, sometimes even to global optima, as well as advanced techniques such as reparameterization and hidden convexity. We review these developments in the following sections.

## A.1. Conditions for Global Convergence in Non-convex Optimization

**Invexity.** Invexity (Hanson, 1981) is a generalization of convexity, with a property that stationary points are global optima (Martin, 1985; Ben-Israel & Mond, 1986). The classical theory of invexity is detailed in (Mishra & Giorgi, 2008). Recent work (Barik et al., 2023) develops projected invex gradient descent algorithms that find global optima for invex programs under certain assumptions. Additionally, the invex structure has been applied to learning tasks, such as image reconstruction (Pinilla et al., 2022; Pinilla & Thiyagalingam, 2024), to achieve global optima instead of merely critical points.

**PL/KL conditions.** Kurdyka-Łojasiewicz (KL) condition (Lojasiewicz, 1963a; Kurdyka, 1998) is widely used to analyze local convergence in non-convex minimization. The Polyak-Łojasiewicz (PL) condition (Polyak, 1963; Lojasiewicz, 1963b), a global variant of the KL condition, ensures that stationarity implies optimality and serves as a sufficient condition for global linear convergence in non-convex problems. This condition has been applied to non-convex, non-smooth optimization (Bento et al., 2024) and learning tasks such as training neural networks (Reddi et al., 2016; Lei et al., 2019) and stochastic risk minimization (Foster et al., 2018). Theoretical studies have explored the relationship between (generalized) PL and other conditions (Karimi et al., 2016), the calculus of KL functions (Li & Pong, 2018), and convergence rates for functions satisfying the KL condition with varying exponents (Frankel et al., 2015).

**Quasar-convexity.** Quasar-convexity (Hardt et al., 2018) is a relaxation of convexity parameterized by $\gamma \in (0, 1]$, with $\gamma = 1$ implying star-convexity. This property arises in various optimization and learning tasks such as the objectives in, learning linear dynamical systems (Hardt et al., 2018), positive semidefinite matrix completion (Ge et al., 2016), and neural network training tasks (Zhou et al., 2019; Kleinberg et al., 2018). For quasar-convex objectives, gradient-based methods can achieve a comparable convergence rate as convex objectives to a global optimum, with convergence analyses available for standard algorithms (Gower et al., 2021; Guminov et al., 2017) and accelerated methods (Guminov et al., 2017; Hinder et al., 2020; Nesterov et al., 2018a; Fu et al., 2023).

## A.2. Non-Convex Constrained Optimization

For optimization problems with non-convex constraints, convergence guarantees for standard PGD algorithms are rarely provided. The existing literature often imposes extremely stringent conditions, such as assumptions on local concavity coefficients (Barber & Ha, 2018) or adopts a manifold optimization framework (Balashov et al., 2020; Balashov, 2021; Boumal, 2023).

In fact, convergence analysis for non-convex constrained optimization is generally scarce and frequently relies on inconsistent or overly restrictive assumptions, not just for projection-based algorithms but across other approaches as well. To address these challenges, several works have proposed alternative methodologies, including regularized subgradient methods (Ma et al., 2020), inexact Lagrangian augmented methods (Sahin et al., 2019; Xie & Wright, 2019; Birgin et al., 2003) and proximal-point-based algorithms (Boob et al., 2019; Ma et al., 2019; Lin et al., 2022). Among these works, the state-of-the-art work (Lin et al., 2022) achieves the fastest convergence rate $\mathcal{O}(\epsilon^{-3})$ for non-convex optimization problems with weakly convex constraints, under some regularization assumption. We refer readers to this paper for a comprehensive discussion of the assumptions and convergence analysis in related work.

## A.3. Recent Advances for Non-Convex Optimization

To reduce the cost and accelerate the convergence for solving (non-)convex constrained optimization, recent novel projection-free methods and other advanced techniques involve inexact projection, radial dual formulation, reparameterizing optimization problems, and uncovering hidden convexity.

**Inexact projection**. In many cases, the projection operator lacks an analytic solution or is computationally expensive to compute exactly, motivating the analysis of inexact projected methods. For convex optimization, such methods achieve the same convergence rate as PGD if the cumulative projection error is bounded (Schmidt et al., 2011; Patrascu & Necoara,

2018), with new results derived under specific settings (Patrascu & Irofti, 2021). For nonconvex objectives with convex constraints, their convergence has been analyzed in (Birgin et al., 2003; Wang & Liu, 2006; Zhang et al., 2020). Recent advances further generalize inexact projection operators to broader settings (Ferreira et al., 2022; Aguiar et al., 2023).

**Radial duality.** Beyond classical projection-free methods, recent advancements have introduced novel approaches based on gauge and radial duality theory. Radial duality theory for nonnegative optimization problems (Grimmer, 2024a;b) demonstrates that constrained optimization problems can be reformulated as unconstrained problems using the gauge of their constraints. This framework has led to the development of new families of projection-free methods with optimal convergence guarantees (Liu & Grimmer, 2023), as well as relaxed conditions (Samakhoana & Grimmer, 2024) that enable more efficient line search operators for the reformulated unconstrained problems.

**Reparameterization.** Reparameterizing optimization problems aims to mitigate challenging properties, such as non-smoothness or non-convexity, via invertible transformations while preserving equivalent optima. Parameterization is widely used in optimization and learning tasks, including semi-definite programming (Cifuentes, 2021), low-rank optimization (Mishra et al., 2014; Ha et al., 2020), and risk minimization (Bah et al., 2022). Recent advancements include parameterizing simplex (Li et al., 2023) and polyhedron (Tang & Toh, 2024) optimization via Hadamard transformation to reduce projection complexity, smooth over-parameterization to accelerate non-smooth optimization algorithms (Poon & Peyré, 2023), parameterizing discrete data as continuous for generative learning (Davis et al., 2024), and analyzing the optimization landscape under parameterization transformations in non-convex settings (Levin et al., 2024).

**Hidden convexity.** Hidden convexity refers to transformations that reveal the convex structure of non-convex sets or functions, which has been exploited in problems such as rotation matrix optimization (Ramachandran et al., 2024), non-linear least squares (Drusvyatskiy & Paquette, 2019), revenue management and inventory control (Chen et al., 2022), and quadratically constrained quadratic programming (QCQP) with Toeplitz-Hermitian quadratics (Konar & Sidiropoulos, 2015). For non-convex stochastic optimization with hidden structure, projected gradient-based algorithms can achieve the same convergence rate as in convex optimization for both strongly convex (Fatkhullin et al., 2023) and convex objectives (Chen et al., 2022) under certain assumptions. Furthermore, QCQP, which is generally NP-hard, can be solved in polynomial time when hidden convexity is present (Konar & Sidiropoulos, 2015).

# B. Learning Homeomorphism via INN

In this section, we provide the omitted details in Sec. 2 and 3.

## B.1. Handling Constraint Set with Equality

We first explain how to handle equality constraints as mentioned in Sec. 2. Consider the constrained set $\mathcal{K}_{\boldsymbol{\theta}}$ as follows

$$\mathcal{K}_{\boldsymbol{\theta}} = \{\mathbf{x} \mid \mathbf{q}_{\boldsymbol{\theta}}(\mathbf{x}) = 0, g_{1,\boldsymbol{\theta}}(\mathbf{x}) \leq 0, \cdots, g_{m,\boldsymbol{\theta}}(\mathbf{x}) \leq 0\}$$

where $\mathbf{q} = (q_1, q_2, \cdots, q_{m_{\mathrm{eq}}})$ with continuous functions $q_{i,\boldsymbol{\theta}}(\mathbf{x}) : \mathbb{R}^n \to \mathbb{R}$ with respect to $\mathbf{x}$ and $\boldsymbol{\theta}$.

Suppose the rank of the equality constrained function is constant for all $\mathbf{x} \in \mathcal{K}_{\boldsymbol{\theta}}$, i.e.,

$$\mathrm{rank}\left(\mathrm{J}_{\mathbf{q}}(\mathbf{x})\right) = r, \quad \forall \mathbf{x} \in \mathcal{K}_{\boldsymbol{\theta}}.$$

Then $\{\mathbf{q}_{\boldsymbol{\theta}}(\mathbf{x}) = \mathbf{0}\}$ is of dimension $n - r$ by the Constant-Rank Level Set Theorem (Lee & Lee, 2012). In other words, we can use a subset of decision variables $\mathbf{x}_1 \in \mathbb{R}^{n-r}$ and reconstruct full decision variable $[\mathbf{x}_1, \mathbf{x}_2] \in \mathbb{R}^n$ via the equality constraint, where $\mathbf{x}_2 = \boldsymbol{\phi}_{\boldsymbol{\theta}}(\mathbf{x}_1)$ and $\mathbf{q}_{\boldsymbol{\theta}}([\mathbf{x}_1, \boldsymbol{\phi}_{\boldsymbol{\theta}}(\mathbf{x}_2)]) = \mathbf{0}$. Such a reconstruction process ensures the feasibility of the equality constraint. Hence, the constraint $\mathcal{K}_{\boldsymbol{\theta}}$ can be reformulated as

$$\mathcal{K}_{\boldsymbol{\theta}}^s = \{\mathbf{x}_1 \in \mathbb{R}^{n-r} \mid g_{1,\boldsymbol{\theta}}(\mathbf{x}_1, \boldsymbol{\phi}_{\boldsymbol{\theta}}(\mathbf{x}_1)) \leq 0, \cdots, g_{m,\boldsymbol{\theta}}(\mathbf{x}_1, \boldsymbol{\phi}_{\boldsymbol{\theta}}(\mathbf{x}_1)) \leq 0\}.$$

It follows from the reconstruction that

$$(\mathbf{x}_1, \mathbf{x}_2 = \boldsymbol{\phi}_{\boldsymbol{\theta}}(\mathbf{x}_2)) \in \mathcal{K}_{\boldsymbol{\theta}} \Leftrightarrow \mathbf{x}_1 \in \mathcal{K}_{\boldsymbol{\theta}}^s.$$

It is noteworthy that the constant rank assumption for $\mathbf{q}_{\boldsymbol{\theta}}(\cdot)$ holds globally for linear equalities and locally for nonlinear manifold equalities (see, e.g., (Lee, 2010; Boumal, 2023)), which encompasses a majority of practical optimization applications. Based on the foregoing analysis, this paper assumes that the constrained set $\mathcal{K}_{\boldsymbol{\theta}}$ includes only equality constraints. For a detailed discussion on managing linear equalities, nonlinear inequalities, and manifold equalities, the reader is referred to Appendix A and Appendix B in (Liang et al., 2023).

## B.2. Introduction of Invertible Neural Networks

The INN $\Phi : \mathbb{R}^n \to \mathbb{R}^n$ is a class of neural networks that is a continuous bijection. It is a finite composition of invertible layers, where each layer is also a homeomorphic mapping with tunable parameters. In the following, we introduce several commonly used invertible layers for INN, and refer readers to (Papamakarios et al., 2021) for a more comprehensive introduction. *Moreover, denote $\mathcal{H}^n$ the set of homeomorphisms from $\mathbb{R}^n$ to $\mathbb{R}^n$.*

**Remark.** In this work, we follow the GLOW architecture (Kingma & Dhariwal, 2018) for INN design, which consists of a composition of finite affine coupling layers and invertible linear layers. Specifically, an $l$-layer INN is defined as

$$\Phi = \Phi^l \circ \Phi^{l-1} \cdots \circ \Phi^1$$

where each layer $\Phi^j = f^j_{\text{coup}} \circ \mathcal{L}^j$ ($j \in [l]$) consists of an invertible linear transformation $\mathcal{L}^j(\mathbf{x}) = \mathbf{Q}_j \mathbf{x}$ for some rotation matrix $\mathbf{Q}_j$ and a coupling layer $f_{\text{coup}}$ of fixed splitting strategy $k = \lfloor n/2 \rfloor$.

This structure offers several key advantages: (i) it admits closed-form forward and inverse computations through neural network propagation, (ii) it enables closed-form calculation of Jacobian singular values, which are essential for computing the log-determinant and Lipschitz constant required in our INN loss function, and (iii) affine coupling layers are universal approximators for any differentiable homeomorphism (Teshima et al., 2020). Given these theoretical and computational advantages, we adopt the coupling layer-based INN architecture for our framework.

## B.3. Computational Issues of Invertible Neural Networks

In this section, we analyze the computational issues of INNs $\Phi$. There are several requirements for the Invertible Neural Network (INN):

- (i) The forward and inverse mappings of the INN must be efficiently computable, as they are required to map solutions between the original space and the transformed space within Hom-PGD$^+$.

- (ii) The Jacobian of the INN must be computable, as it is essential for evaluating the gradient of the composite function $H = f \circ \Phi$ in the Hom-PGD$^+$ algorithm.

- (iii) The singular values of the Jacobian matrix must be accessible, as they are necessary for estimating terms in the loss function defined in Eq. (6) during the INN training process.

- (iv) The INN should have bounded distortion to ensure the worst-case performance for homeomorphic projection. Furthermore, the INN should be a universal approximator of homeomorphic mappings. This enables it to handle complex transformations involving a broad range of constraints.

*Since this paper adopts the coupling-layer-based INN architecture, we focus our analysis specifically on this type of INN.* For conciseness of notations, we fix $\boldsymbol{\theta}$ and omit it. For an $l$-layer INN denoted as $\Phi = \Phi^l \circ \cdots \circ \Phi^j \circ \cdots \circ \Phi^1$, we denote $\mathbf{x}^j = \Phi^{j-1}\left(\mathbf{x}^{j-1}\right)$ for $j = 2, \cdots, l$ and $\mathbf{x}^1 = \mathbf{x}$. Moreover, we denote $W$ as the size (number of parameters) of an INN.

(i) In each affine coupling layer, the forward and inverse could be computed directly by the definition, i.e., for $\mathbf{x} = (\mathbf{x}_1 \in \mathbb{R}^{n_1}, \mathbf{x}_2 \in \mathbb{R}^{n_2})$ with $n_1 + n_2 = n$ and two arbitrary NNs $\boldsymbol{\gamma} > 0, \boldsymbol{\tau} : \mathbb{R}^{n_1} \to \mathbb{R}^{n_2}$, we have

$$
\begin{aligned}
\text{Forward}: \quad & (\mathbf{y}_1, \mathbf{y}_2) = (\mathbf{x}_1, \mathbf{x}_2 \odot \boldsymbol{\gamma}(\mathbf{x}_1) + \boldsymbol{\tau}(\mathbf{x}_1)), \\
\text{Inverse}: \quad & (\mathbf{x}_1, \mathbf{x}_2) = (\mathbf{y}_1, (\mathbf{y}_2 - \boldsymbol{\tau}(\mathbf{y}_1)) / \boldsymbol{\gamma}(\mathbf{y}_1))
\end{aligned}
\tag{2}
$$

where $/$ is applied element-wise to vector computation. For the conditional layer, we augment the input parameters $\boldsymbol{\theta}$ as, $\boldsymbol{\gamma}_{\boldsymbol{\theta}}(\cdot)$ and $\boldsymbol{\tau}_{\boldsymbol{\theta}}(\cdot)$. *Therefore, the complexity of computing $\Phi$ and $\Phi^{-1}$ is $\mathcal{O}(W)$.*

(ii) The Jacobian of such a composited mapping and its determinant can be expressed as

$$\mathrm{J}_\Phi(\mathbf{x}) = \prod_{j=1}^{l} \mathrm{J}_{\Phi^j}\left(\mathbf{x}^j\right), \quad \left|\det \mathrm{J}_\Phi(\mathbf{x})\right| = \prod_{j=1}^{l} \left|\det \mathrm{J}_{\Phi^j}\left(\mathbf{x}^j\right)\right|.$$

For each affine coupling layer, the Jacobian can be expressed as

$$\frac{\partial \mathbf{y}}{\partial \mathbf{x}} = \left[ \begin{array}{cc} \mathbf{I}_{n_1} & 0 \\ \frac{\partial \mathbf{y}_2}{\partial \mathbf{x}_1} & \operatorname{diag}\left(\boldsymbol{\gamma}(\mathbf{x}_1)\right) \end{array} \right],$$

where $\mathrm{diag}(\mathbf{v})$ returns a diagonal matrix whose diagonal elements are given by the vector $\mathbf{v}$. *It follows that the complexity of computing $\mathrm{J}_\Phi(\mathbf{x})$ is $\mathcal{O}(W)$.*

(iii) For each layer, the Jacobian determinant can be expressed as the product of singular values:

$$\left| \det \mathrm{J}_{\Phi^j} \left( \mathbf{x}^j \right) \right| = \prod_{i=1}^{n} \sigma_i \left( \mathrm{J}_{\Phi^j} \left( \mathbf{x}^j \right) \right)$$

where $\sigma_1(\cdot) \geq \ldots \geq \sigma_n(\cdot) > 0$ are the sorted singular values of the Jacobian matrix of the mapping $\Phi^j(\cdot)$ at $\mathbf{x}$. By the design of each affine coupling layer, such an invertible transformation has a closed-form expression of singular values, which is 1 or elements of $\boldsymbol{\gamma}(\mathbf{x}_1)$. *Therefore, the complexity to compute the determinant or singular values of an coupling layer INN is still $\mathcal{O}(W)$.*

(iv) The bounded distortion property of an INN constructed with affine coupling layers is inherently guaranteed by its architectural design. Moreover, its universal approximation capability for homeomorphic mappings over compact domains has been established in the existing literature. These two properties are formally stated below.

**Proposition B.1.** *Suppose $\Phi$ is an INN composed of affine coupling layers. Then:*

*(i) $\Phi$ is capable of approximating any $n$-dimensional differentiable homeomorphism over a compact domain, given a sufficiently large number of layers (Jin et al., 2024; Liang et al., 2024; Ishikawa et al., 2022).*

*(ii) $\Phi$ exhibits bounded distortion, where the bound depends on the number of layers (Liang et al., 2024).*

### B.4. Unsupervised INN Training

We denote

$$\mathcal{H}^n := \{\boldsymbol{\phi} : \mathbb{R}^n \to \mathbb{R}^n \mid \boldsymbol{\phi} \text{ is a homeomorphism}\}, \mathcal{H}^n(\mathcal{K}_{\boldsymbol{\theta}}, \mathcal{B}) := \{\boldsymbol{\psi} \in \mathcal{H}^n \mid \boldsymbol{\psi}(\mathcal{B}) = \mathcal{K}_{\boldsymbol{\theta}}\}.$$

Moreover, the feasible set $\mathcal{H}^n (\mathcal{K}_{\boldsymbol{\theta}}, \mathcal{B})$ is equivalent to the set of optimal solutions to the problem (Liang et al., 2023; 2024):

$$\max_{\boldsymbol{\psi}_{\boldsymbol{\theta}} \in \mathcal{H}^n} \log \mathrm{V}\left(\boldsymbol{\psi}_{\boldsymbol{\theta}}(\mathcal{B})\right) \quad \text{s.t. } \boldsymbol{\psi}_{\boldsymbol{\theta}}(\mathcal{B}) \subseteq \mathcal{K}_{\boldsymbol{\theta}} \tag{3}$$

where $\mathrm{V}\left(\boldsymbol{\psi}_{\boldsymbol{\theta}}(\mathcal{B})\right)$ computes the volume of set $\boldsymbol{\psi}_{\boldsymbol{\theta}}(\mathcal{B})$ and the constraint means that the set $\boldsymbol{\psi}_{\boldsymbol{\theta}}(\mathcal{B})$ is a subset of $\mathcal{K}_{\boldsymbol{\theta}}$. While there might be multiple homeomorphisms in the set $\mathcal{H}^n(\mathcal{K}_{\boldsymbol{\theta}}, \mathcal{B})$ (e.g., through composition with rotations over the ball, we get an additional such homeomorphism), we wish to learn one with minimum Lipschitz constant. To this end, we define the Lipschitz constant of a mapping $\boldsymbol{\psi}$ over a set $\mathcal{K}$ as

$$\mathrm{L}(\boldsymbol{\psi}) = \sup_{\mathbf{z} \neq \mathbf{u} \in \mathcal{K}} \frac{\|\boldsymbol{\psi}(\mathbf{z}) - \boldsymbol{\psi}(\mathbf{u})\|}{\|\mathbf{z} - \mathbf{u}\|}. \tag{4}$$

Intuitively, the minimum Lipschitz homeomorphical (MLH) mapping problem can be reformulated to the following bi-level problem:

$$\min_{\boldsymbol{\psi}_{\boldsymbol{\theta}} \in \mathcal{H}^n} \log \mathrm{L}\left(\boldsymbol{\psi}_{\boldsymbol{\theta}}\right) \text{ s.t. } \boldsymbol{\psi}_{\boldsymbol{\theta}} \in \arg\max\{ \text{ Problem in (3) }\}. \tag{5}$$

We employ the following loss function and maximize it to train an INN $\Phi_{\boldsymbol{\theta}}$ with $l$ layers for learning the homeomorphic mapping $\boldsymbol{\theta}$ in an unsupervised manner:

$$\mathcal{L}\left(\Phi_{\boldsymbol{\theta}}\right) = \widehat{\mathrm{V}}\left(\Phi_{\boldsymbol{\theta}}(\mathcal{B})\right) - \lambda_1 \mathrm{P}\left(\Phi_{\boldsymbol{\theta}}(\mathcal{B})\right) - \lambda_2 \widehat{\mathrm{L}}\left(\Phi_{\boldsymbol{\theta}}\right) \tag{6}$$

where $\lambda_1$ and $\lambda_2$ are positive coefficients to balance among the three terms. For ease of analysis of how to compute the three terms, we denote an $l$-layer INN as $\Phi_{\boldsymbol{\theta}} = \Phi_{\boldsymbol{\theta}}^l \circ \ldots \circ \Phi_{\boldsymbol{\theta}}^2 \circ \Phi_{\boldsymbol{\theta}}^1$, where each layer is either a bi-Lip affine coupling layer or an invertible linear layer.

(i) $\widehat{\mathrm{V}}\left(\Phi_{\boldsymbol{\theta}}(\mathcal{B})\right)$ is a computable approximation of the log-volume term $\log \mathrm{V}\left(\Phi_{\boldsymbol{\theta}}(\mathcal{B})\right)$ in (3) as:

$$\widehat{V}\left(\Phi_{\boldsymbol{\theta}}(\mathcal{B})\right) = \frac{1}{V(\mathcal{B})} \int_{\mathcal{B}} \sum_{i=1}^{n} \sum_{j=1}^{l} \log \sigma_i \left( J_{\Phi_{\boldsymbol{\theta}}^j} \left( \mathbf{z}^j \right) \right) d\mathbf{z} + \log V(\mathcal{B}) \tag{7}$$

where $\mathbf{z}^j = \Phi_{\boldsymbol{\theta}}^{j-1} \left( \mathbf{z}^{j-1} \right)$ for $j = 2, \cdots, l$, and $\mathbf{z}^1 \in \mathcal{B}$, $J_{\Phi_{\boldsymbol{\theta}}^j} \left( \mathbf{z}^j \right)$ denotes the Jacobian matrix of $\Phi_{\boldsymbol{\theta}}^j(\cdot)$ at $\mathbf{z}^j$.

(ii) $P\left(\Phi_{\boldsymbol{\theta}}(\mathcal{B})\right)$ is the penalty term for the constraint violation of $\Phi_{\boldsymbol{\theta}}(\mathcal{B}) \subseteq \mathcal{K}_{\boldsymbol{\theta}}$ in (3) as:

$$P\left(\Phi_{\boldsymbol{\theta}}(\mathcal{B})\right) = \int_{\mathcal{B}} \|\text{ReLU}\left(\mathbf{g}\left(\Phi_{\boldsymbol{\theta}}(\mathbf{z}), \boldsymbol{\theta}\right)\right)\|_1 \, d\mathbf{z}, \tag{8}$$

where $\text{ReLU}(\cdot) = \max\{0, \cdot\}$ and $\mathbf{g}\left(\Phi_{\boldsymbol{\theta}}(\mathbf{z}), \boldsymbol{\theta}\right)$ calculates the residual for each inequality constraint as $[g_1\left(\Phi_{\boldsymbol{\theta}}(\mathbf{z}), \boldsymbol{\theta}\right), \ldots, g_m\left(\Phi_{\boldsymbol{\theta}}(\mathbf{z}), \boldsymbol{\theta}\right)]$.

(iii) $\widehat{L}\left(\Phi_{\boldsymbol{\theta}}^{-1}, \mathcal{K}_{\boldsymbol{\theta}}\right)$ is a computable approximation of the log-Lipschitz term $\log L\left(\Phi_{\boldsymbol{\theta}}^{-1}, \mathcal{K}_{\boldsymbol{\theta}}\right)$ as:

$$\widehat{L}\left(\Phi_{\boldsymbol{\theta}}\right) = \sup_{\mathbf{z}^1 \in \mathcal{Z}_{\boldsymbol{\theta}}} \left\{ \sum_{j=1}^{l} \log \sigma_1 \left( J_{\Phi_{\boldsymbol{\theta}}^j} \left( \mathbf{z}^j \right) \right) \right\} \tag{9}$$

where $\mathbf{z}^j = \Phi_{\boldsymbol{\theta}}^{j-1} \left( \mathbf{z}^{j-1} \right)$ for $j = 2, \cdots, l$, and $\mathbf{z}^1 \in \mathcal{Z}_{\boldsymbol{\theta}} = \Phi_{\boldsymbol{\theta}}^{-1}\left(\mathcal{K}_{\boldsymbol{\theta}}\right)$.

We have the following bounds for the approximations (Liang et al., 2023; 2024). The two approximation terms in (7) and (9) satisfy $\log V\left(\Phi_{\boldsymbol{\theta}}(\mathcal{B}) \geq \widehat{V}\left(\Phi_{\boldsymbol{\theta}}(\mathcal{B})\right)$ and $\log L\left(\Phi_{\boldsymbol{\theta}}\right) \leq \widehat{L}\left(\Phi_{\boldsymbol{\theta}}\right)$.

The above proposition implies that the loss function in (6) is actually a lower bound to the Lagrangian of the problem in (5). Therefore, we can maximize the loss function in (6) to approximate the MLH mapping under the equivalent reformulation in (5). Further, to train one conditional INN $\Phi \in \mathcal{H}^{n+d}$ to learn the $\boldsymbol{\theta}$-dependent MLH mappings for any $\boldsymbol{\theta} \in \Theta$, we generalize the loss in (6) to

$$\mathcal{L}(\Phi) = \mathbb{E}_{\boldsymbol{\theta}} \left[ \mathcal{L}\left(\Phi_{\boldsymbol{\theta}}\right) \right]$$

where $\boldsymbol{\theta} \in \Theta$ is uniformly sampled. For the INN training, we prepare quasi Monte Carlo (QMC) samples $\{\mathbf{z}_i\}_{i=1}^{N} \subset \mathcal{B}$ to approximate the integration in (7) and (8). When evaluating the distortion in (9), since we may not know $\mathcal{Z}_{\boldsymbol{\theta}}$ in advance, we sample from $\mathcal{Z}_{\boldsymbol{\theta}} = \Phi_{\boldsymbol{\theta}}^{-1}\left(\mathcal{K}_{\boldsymbol{\theta}}\right) \subset \mathcal{B}$ over a unit ball as $\{\mathbf{z}_i\}_{i=1}^{N}$. In each iteration, we sample a batch of collected data and employ the Adam optimizer to maximize the loss function $\mathcal{L}(\Phi)$, similar to training standard NNs (Kingma & Ba, 2014).

### B.5. Offline Complexity to Obtain a Trained Valid INN

In this section, we will discuss the theoretical complexity of obtaining a trained, valid INN $\Phi_{\boldsymbol{\theta}}$ which approximates $\psi_{\boldsymbol{\theta}}$ where $\psi_{\boldsymbol{\theta}}(\mathcal{B}) = \mathcal{K}_{\boldsymbol{\theta}}$ for the optimization **P**.

**Complexity of obtaining a valid INN.** To obtain a valid invertible neural network (INN) $\Phi_{\boldsymbol{\theta}} \approx \psi_{\boldsymbol{\theta}}$ given a ball-homeomorphic constrained set $\mathcal{K}_{\boldsymbol{\theta}}$, one must incur the following cost.

- Training. Training a neural network is an unconstrained non-convex optimization, which is NP-hard to find a global optimum in general. In practice, we use Adam optimizer to maximize the loss function, similar to the process of training regular NNs (Kingma & Ba, 2014). Typically, the run-time is poly($\epsilon^{-1}$) to find an approximate stationary solution.

- #Samples of $\mathcal{B}$. As discussed in Sec. B.4, one will prepare samples $\{\mathbf{z}_i\} \subset \mathcal{B}$ to approximate the integration (7), (8) and (9) using QMC. The integration error for the QMC approach is $\mathcal{O}\left((\log N)^{n-1} / N\right)$ where $N$ is the number of samples, which is faster in the rate of convergence than Monte Carlo using a pseudorandom sequence (Dick & Pillichshammer, 2010).

- INN size. For the INN size to approximate a bi-continuous $n$-dimensional homeomorphism to an error $\epsilon$, the theoretical upper bound $\mathcal{O}(\epsilon^{-n})$ derived from (Jin et al., 2024) is high due to the worst-case analysis. Meanwhile, the lower bound is an open question so far. Note that the theoretical bound of INN size is high and grows exponentially with the input dimension $n$ due to a worst-case analysis. However, in practice, the target homeomorphism may be much simpler,

requiring significantly fewer parameters for the INN to approximate it effectively. For instance, in our empirical study, we found that approximately three coupling layers with width $\mathcal{O}(n)$ are sufficient to learn the homeomorphic mapping from a non-convex set to a ball.

**Remark.** Although training the INN offline incurs additional computational cost, this expense is only one-time and can be amortized over numerous online problem instances. Moreover, modern deep learning frameworks, such as PyTorch coupled with GPU acceleration, render the training process efficient (e.g., less than 10 minutes for high-dimensional chance-constrained problems). Once the INN is appropriately trained, the framework achieves a convergence rate comparable to optimization over convex constraint sets $\left(\mathcal{O}\left(\epsilon^{-2}\right)\right)$ with a low per-iteration cost, significantly improving on state-of-the-art rates of $\mathcal{O}\left(\epsilon^{-4}\right)$ or $\mathcal{O}\left(\epsilon^{-3}\right)$ under regularity conditions (see Table 1 for details).

In practice, it is often necessary to verify whether a constrained set is homeomorphic to a ball. This question can generally be divided into two cases:

(i) *Special cases with known topological properties*. Certain sets are naturally homeomorphic to a ball, such as compact convex sets (Bredon, 2013) and star-shaped sets (Appendix B.6). In particular, for compact convex sets, an explicit ball-homeomorphic mapping can be directly constructed using the gauge mapping, as discussed in (Liu et al., 2026). For star-shaped sets, a ball-homeomorphic mapping can also be constructed; however, it may depend on certain unknown parameters specific to the star-shaped set. As a result, it is often more practical to use an INN to approximate the homeomorphic mapping. Further details are provided in Appendix B.6.

(ii) *General non-convex sets*. For general compact non-convex constrained sets, we may apply topological data analysis (TDA) (Chazal & Michel, 2021; Otter et al., 2017) to determine whether the set satisfies the ball-homeomorphic property.

**Remark.** While verifying the ball-homeomorphism property through sampling and topological data analysis can be computationally expensive, explicit verification is often unnecessary in practice. Many common constraint sets—including convex and star-shaped sets—possess known topological properties that naturally guarantee ball-homeomorphism.

More generally, our method can be applied whenever the *valid INN condition* (Definition 3.2) is satisfied, which requires only that *the INN maps the center of the unit ball to a feasible point in the constraint set*. As discussed in Section 4.4, our theoretical guarantees (feasibility preservation and convergence rate) hold under this valid INN condition alone.

This makes ball-homeomorphism verification a *sufficient but not necessary* prerequisite—the valid INN condition provides a more practical and verifiable criterion that can be easily checked without expensive topological analysis. In essence, practitioners need only verify that their trained INN satisfies the valid INN condition, which is straightforward to evaluate through simple feasibility checking.

### B.6. Homeomorphisms from a Star-Shaped Set to a Ball

**Definition B.2** (Star-shaped set)**.** A set is called a *star-shaped* set if it has the property that all interior and boundary points are visible from a point $\mathbf{x}^{\circ}$ (called *star center*) in the set. Note that the set of star centers of a star-shaped set might have multiple and even infinite elements.

For the geometric, analytical, combinatorial, and topological properties of star-shaped sets, and their broad applicability in many mathematical fields, we refer readers to (Hansen et al., 2020) for a comprehensive discussion and review.

Importantly, a star-shaped set is homeomorphic to a unit ball. The formal statement is given below, where one could refer to, e.g., Page 60 (Gonnord & Tosel, 1998) and Theorem 237 of the handbook *Analysis III* by Dirk Ferus, for its proof.

**Proposition B.3.** *Open star-shaped sets are diffeomorphic to open balls, where a diffeomorphism is a smooth homeomorphism.*

For a star-shaped set $\mathcal{S}$, using $\mathbf{x}^{\circ}$ as the center, one can construct an explicit homeomorphism $\psi$ that continuously and bijectively sends points in $\mathcal{S}$ to points in a unit ball $\mathcal{B}$. Such a homeomorphism is termed a gauge mapping (Tabas & Zhang, 2022a;b; Liang & Chen, 2025; Li et al., 2025; 2026; Liu et al., 2026) defined below.

**Definition B.4** (Gauge mapping)**.** Suppose $\mathcal{S}$ is a star-shaped set with star center $\mathbf{x}^{\circ}$. Let $\gamma_{\mathcal{S}}(\mathbf{x}, \mathbf{x}^{\circ}) = \inf\{\lambda \geq 0 \mid \mathbf{x} \in \lambda(\mathcal{S} - \mathbf{x}^{\circ})\}$ be the Gauge/Minkowski function (Blanchini & Miani, 2008) given a star center $\mathbf{x}^{\circ} \in \text{int}(\mathcal{S})$. The gauge

mapping $\psi : \mathcal{B} \to \mathcal{S}$ is defined between a unit ball and a compact star-shaped set:

$$\psi(\mathbf{z}) = \frac{\|\mathbf{z}\|}{\gamma_{\mathcal{S}}(\mathbf{z}, \mathbf{x}^\circ)} \mathbf{z} + \mathbf{x}^\circ, \ \forall \mathbf{z} \in \mathcal{B}; \qquad \psi^{-1}(\mathbf{x}) = \frac{\gamma_{\mathcal{S}}(\mathbf{x} - \mathbf{x}^\circ, \mathbf{x}^\circ)}{\|\mathbf{x} - \mathbf{x}^\circ\|}(\mathbf{x} - \mathbf{x}^\circ), \ \forall \mathbf{x} \in \mathcal{S}. \tag{10}$$

Based on the explicit construction of homeomorphisms between the unit ball and a star-shaped set, the gauge mapping can be efficiently computed by evaluating the gauge function using a bisection-based algorithm (Liu et al., 2026). Moreover, it is important to note that the above construction depends on the center of a star-shaped set. However, in general, finding a star center of a star-shaped set is very challenging and can be NP-hard (O'Rourke & Supowit, 1983; Lee & Lin, 1986). In such cases, one can utilize an INN to learn the ball-homeomorphic mapping directly as discussed in Sec. 3, avoiding the need to verify whether the star-shaped set is ball-homeomorphic.

# C. Preliminaries for Technical Proof

In this section, we summarize the related basic concepts, notations, assumptions, and fundamental propositions and lemmas.

## C.1. Basic Concepts

We list the basic concepts used in this paper below.

- Distance between a point and a set. For a closed set $\mathcal{X} \in \mathbb{R}^n$ and any $\mathbf{x} \in \mathbb{R}^n$, the distance between $\mathbf{x}$ and $\mathcal{X}$ is defined as $\mathrm{dist}(\mathbf{x}, \mathcal{X}) = \inf_{\mathbf{y} \in \mathcal{X}} \|\mathbf{x} - \mathbf{y}\|$.

- Orthogonal projection. For a closed set $\mathcal{X}$, the orthogonal projection of a point $\mathbf{x} \in \mathbb{R}^n$ onto $\mathcal{X}$ is defined as $\Pi_{\mathcal{X}}(\mathbf{x}) \in \arg\min_{\mathbf{y} \in \mathcal{X}} \|\mathbf{x} - \mathbf{y}\|$.

- Function convexity. For a differentiable function $f : \mathcal{X} \subseteq \mathbb{R}^n \to \mathbb{R}$, it is said to be convex if one of the following holds:

  1) Jensen's inequality. For $\theta$ with $0 \le \theta \le 1$, we have $f(\theta \mathbf{x} + (1 - \theta)\mathbf{y}) \le \theta f(\mathbf{x}) + (1 - \theta)f(\mathbf{y})$ for all $\mathbf{x}, \mathbf{y} \in \mathcal{X}$.
  2) first-order condition. $f(\mathbf{y}) \ge f(\mathbf{x}) + \langle \nabla f(\mathbf{x}), \mathbf{y} - \mathbf{x} \rangle, \forall \mathbf{x}, \mathbf{y} \in \mathcal{X}$.
  3) monotone gradient. $(\nabla f(\mathbf{x}) - \nabla f(\mathbf{y}))^T(\mathbf{x} - \mathbf{y}) \ge 0$ for all $\mathbf{x}, \mathbf{y} \in \mathcal{X}$.

- $L$-Smoothness. A differentiable function $f : \mathcal{X} \subseteq \mathbb{R}^n \to \mathbb{R}$ is said be $L$-smooth if one of the following holds:

  1) zeroth-order condition. $f(\lambda \mathbf{x} + (1 - \lambda)\mathbf{y}) \ge \lambda f(\mathbf{x}) + (1 - \lambda)f(\mathbf{y}) - \frac{L}{2}\lambda(1 - \lambda)\|\mathbf{y} - \mathbf{x}\|^2$, for all $\mathbf{x}, \mathbf{y} \in \mathcal{X}, \lambda \in [0, 1]$.
  2) first-order condition. $f(\mathbf{y}) \le f(\mathbf{x}) + \langle \nabla f(\mathbf{x}), \mathbf{y} - \mathbf{x} \rangle + \frac{L}{2}\|\mathbf{y} - \mathbf{x}\|^2$, for all $\mathbf{x}, \mathbf{y} \in \mathcal{X}$.
  3) Lipschitz gradient. $\|\nabla f(\mathbf{y}) - \nabla f(\mathbf{x})\| \le L\|\mathbf{y} - \mathbf{x}\|$, for all $\mathbf{x}, \mathbf{y}$.

- Weak convexity. A function $f : \mathbb{R}^d \to \mathbb{R}$ is said to be weakly convex with constant $\ell_f > 0$ if the function $f(\mathbf{x}) + (\ell_f/2)\|\mathbf{x}\|^2$ is convex.

- Jacobian matrix. Suppose $\mathbf{f} : \mathbb{R}^n \to \mathbb{R}^m$ is a function such that each of its first-order partial derivatives exists on $\mathbb{R}^n$. Then the Jacobian matrix of $\mathbf{f}$, denoted $J_{\mathbf{f}} \in \mathbb{R}^{m \times n}$, is defined as $J_{\mathbf{f}} = (\frac{\partial f_i}{\partial x_j})_{ij}$.

- A Hessian of a function $f : \mathbb{R}^n \to \mathbb{R}$ is defined as $\nabla^2 f = (\frac{\partial^2 f}{\partial x_i \partial x_j})_{ij} \in \mathbb{R}^{n \times n}$, if its second-order partial derivatives exist. Moreover, for a mapping $\mathbf{f} : \mathbb{R}^n \to \mathbb{R}^m$ with existed second-order partial derivatives of each component $f_i$ $(i = 1, 2, \cdots, m)$. The Hessian of $\mathbf{f}$ is defined as

$$\mathrm{H}(\mathbf{f}) = (\nabla^2 f_1, \cdots, \nabla^2 f_m).$$

## C.2. Basic Assumptions and Notations

**Remark.** For conciseness of notation, we fix the input parameter $\boldsymbol{\theta}$ in problem **P** (and **H**) and omit it, by which we write $\psi, \Phi, f, g_i, h, \mathcal{K}$ to replace $\psi_{\boldsymbol{\theta}}, \Phi_{\boldsymbol{\theta}}, f_{\boldsymbol{\theta}}(\cdot), g_{i,\boldsymbol{\theta}}(\cdot), h_{\boldsymbol{\theta}}(\cdot), \mathcal{K}_{\boldsymbol{\theta}}$ respectively. In the following, we make assumptions throughout the paper.

- Assumptions on $f$ and constraints $g_i$ $(i = 1, 2, \cdots, m)$ in problem **P**:

    1) $f$ is $L_{f,0}$-Lipschitz continuous, i.e., $\|f(\mathbf{x}) - f(\mathbf{y})\| \leq L_{f,0}\|\mathbf{x} - \mathbf{y}\|$ for any $\mathbf{x}, \mathbf{y}$.
    2) $f$ in problem **P** is differentiable and $L_f$-smooth.
    3) $f^* > -\infty$ where $f^* := \min_{\mathbf{x} \in \mathcal{K}} f(\mathbf{x})$.
    4) Each $g_i$ is $L_{g_i,0}$-Lipschitz continuous, differentiable, and $L_{g_i}$-smooth.

- Assumptions on the homeomorphic mapping $\boldsymbol{\psi} : \mathbb{R}^n \to \mathbb{R}^n$:

    1) $\boldsymbol{\psi}$ is differentiable with non-singular Jacobian $\mathrm{J}_{\boldsymbol{\psi}}(\cdot)$,
    2) $\boldsymbol{\psi}$ is $(\kappa_1, \kappa_2)$-bi-Lipschitz continuous for $\kappa_2 \geq \kappa_1 > 0$, i.e.,

    $$\kappa_1\|\mathbf{u} - \mathbf{v}\| \leq \|\boldsymbol{\psi}(\mathbf{u}) - \boldsymbol{\psi}(\mathbf{v})\| \leq \kappa_2\|\mathbf{u} - \mathbf{v}\|.$$

    Then the Jacobian matrix, $\mathrm{J}_{\boldsymbol{\psi}}(\cdot)$ and $\mathrm{J}_{\boldsymbol{\psi}^{-1}}(\cdot)$ will satisfy

    $$\|\mathrm{J}_{\boldsymbol{\psi}}(\mathbf{z})\| \leq \kappa_2, \ \ \forall \mathbf{z}, \quad \|\mathrm{J}_{\boldsymbol{\psi}^{-1}}(\mathbf{x})\| \leq \frac{1}{\kappa_1}, \ \forall \mathbf{x}.$$

    3) $\boldsymbol{\psi}$ has $L_{\boldsymbol{\psi}}$-Lipschitz continuous Jacobian matrix, i.e.,

    $$\|\mathrm{J}_{\boldsymbol{\psi}}(\mathbf{u}) - \mathrm{J}_{\boldsymbol{\psi}}(\mathbf{v})\| \leq L_{\boldsymbol{\psi}}\|\mathbf{u} - \mathbf{v}\|, \ \forall \mathbf{u}, \mathbf{v}.$$

    4) $\boldsymbol{\psi}$ has continuous Hessian, i.e.,
    $$\mathrm{H}_{\boldsymbol{\psi}}(\mathbf{z}) = (\nabla^2 \boldsymbol{\psi}_1, \cdots, \nabla^2 \boldsymbol{\psi}_n)$$

    exists and is continuous.

**Remark.** Given a compact constrained set $\mathcal{K}$, we can relax these global assumptions to hold on a compact domain, including Lipschitz continuity and smoothness. Specifically, we only require $f$ and $\boldsymbol{\psi}$ to be Lipschitz continuous on a compact set containing the feasible constrained set $\mathcal{K}$. The following are detailed explanations. In our convergence analysis of the Hom-PGD$^+$ algorithm, we only require that the composite function $H = f \circ \Phi$ satisfies: (i) $L_H$-smoothness, and (ii) $L_{H,0}$-Lipschitz continuity on the iterates (with both constants depending on the Lipschitz constant of $f$; see Lemma D.1). Since each iterate $\mathbf{z}_k$ is feasible in the ball $\mathcal{B}$, the update $\mathbf{z}_{k+1}^+ = \mathbf{z}_k - \alpha_k \nabla H(\mathbf{z}_k)$ remains in a compact set $\mathcal{M}$ (which contains $\mathcal{B}$) for bounded $\alpha_k$ and $\|\nabla H(\mathbf{z})\|$. Thus, it suffices for $H$ to be smooth and Lipschitz continuous over $\mathcal{M}$, meaning that $f$ need only be Lipschitz continuous on the compact set $\Phi(\mathcal{M}) \supseteq \mathcal{K}$.

In addition, we summarize the commonly used notations in this paper in Table 4.

*Table 4.* Summary of Notations. The notations shown in the table is for problem **P** and we use the same type notations for problem **H**.

| Notation | Definition |
|---|---|
| $\|\cdot\|$ | $l2$-norm $\|\cdot\|_2$ |
| $\mathcal{B}$ | unit ball centered at $0$ |
| $L_{f,0}$ | Lipschitz constant of $f$ |
| $L_f$ | $L_f$-smooth property of $f$ |
| $\mu_f$ | $\mu_f$-strong convexity of $f$ |
| $\kappa_1, \kappa_2$ | bi-Lipschitz constant of $\boldsymbol{\psi}$ |
| $D$ | distortion of $\boldsymbol{\psi}$, i.e., $\kappa_2/\kappa_1$ |
| $L_{\boldsymbol{\psi}}$ | Lipschitz constant of $\mathrm{J}_{\boldsymbol{\psi}}$ |
| $\mathrm{int}(\mathcal{K}), \partial\mathcal{K}$ | the interior, boundary of $\mathcal{K}$ |

## C.3. Basic Facts

In this section, we list the fundamental facts we will use in this paper.

**Proposition C.1** (Properties of Orthogonal Projection, see e.g., (Beck, 2014)). *The projection operator $\Pi_{\mathcal{C}}$ over a closed and convex set $\mathcal{C}$ satisfies the following properties.*

1) *Optimality condition:* $\forall \mathbf{y} \in \mathcal{C}$, $\langle \mathbf{x} - \Pi_{\mathcal{C}}(\mathbf{x}), \mathbf{y} - \Pi_{\mathcal{C}}(\mathbf{x}) \rangle \leq 0$.

2) *Non-Expansiveness:* $\|\Pi_{\mathcal{C}}(\mathbf{x}) - \Pi_{\mathcal{C}}(\mathbf{y})\| \leq \|\mathbf{x} - \mathbf{y}\|$.

3) *Monotonicity:* $\langle \Pi_{\mathcal{C}}(\mathbf{x}) - \Pi_{\mathcal{C}}(\mathbf{y}), \mathbf{x} - \mathbf{y} \rangle \geq 0$.

We have the following lemma related to $\psi$ to help with the computation.

**Lemma C.2.** *Suppose $\mathrm{J}_{\psi}$ is $L_{\psi}$ Lipschitz, i.e., $\|\mathrm{J}_{\psi}(\mathbf{u}) - \mathrm{J}_{\psi}(\mathbf{z})\| \leq L_{\psi}\|\mathbf{u} - \mathbf{z}\|$ for any $\mathbf{u}$ and $\mathbf{z}$. Then, we have*

$$\|\psi(\mathbf{u}) - \psi(\mathbf{z}) - \mathrm{J}_{\psi}(\mathbf{z})(\mathbf{u} - \mathbf{z})\| \leq \frac{L_{\psi}\|\mathbf{u} - \mathbf{z}\|^2}{2}, \ \forall \mathbf{u}, \mathbf{z}.$$

One can refer to Lemma 1.2.3 (Nesterov et al., 2018b) for the proof.

Next, we list the following rules for basic computation:

- Jacobian equivalence: $\mathrm{J}_{\psi^{-1}}(\mathbf{x}) = \mathrm{J}_{\psi}^{-1}(\mathbf{z})$ for $\mathbf{z} = \psi(\mathbf{x})$.

- Chain rule for computing gradient of $h = f \circ \psi$:

$$\nabla h(\mathbf{z}) = \mathrm{J}_{\psi}(\mathbf{z})^{\top} \nabla f(\psi(\mathbf{z})) = \mathrm{J}_{\psi}(\mathbf{z})^{\top} \nabla f(\mathbf{x}).$$

- Chain rule for computing gradient of $f$:

$$\nabla f(\mathbf{x}) = \mathrm{J}_{\psi^{-1}}(\mathbf{x})^{\top} \nabla h(\mathbf{z}) = \mathrm{J}_{\psi}^{-1}(\mathbf{z})^{\top} \nabla h(\mathbf{z}).$$

- Chain rule for computing Hessian of $h = f \circ \psi$:

$$\nabla^2 h(\mathbf{z}) = \mathrm{J}_{\psi}(\mathbf{z})^{\top} \nabla^2 f(\psi(\mathbf{z})) \mathrm{J}_{\psi}(\mathbf{z}) + \sum_{i=1}^{n} \frac{\partial f}{\partial \mathbf{x}_i}(\psi(\mathbf{z})) \nabla^2 \psi_i(\mathbf{z}).$$

## D. Landscape Analysis

In this section, we provide landscape analysis to understand important relationships between problem (P) and (H).

### D.1. Action of Homeomorphism on a Constrained Set

Recall that the constrained set is $\mathcal{K} = \{\mathbf{x} \in \mathbb{R}^n \mid \mathbf{g}(\mathbf{x}) \leq 0\}$ with $\mathbf{g} = (g_1, g_2, \cdots, g_m)$ where each $g_i$ $(i = 1, 2, \cdots, m)$ is a continuous function. For problem (H),

$$\mathcal{B} = \psi^{-1}(\mathcal{K}) = \{\mathbf{z} \in \mathbb{R}^n \mid \psi(\mathbf{z}) \in \mathcal{K}\} = \{\mathbf{z} \in \mathbb{R}^n \mid \mathbf{G}(\mathbf{z}) := \mathbf{g}(\psi(\mathbf{z})) \leq \mathbf{0}\}$$

where $G_i$ is non-convex in general. However, $\mathcal{B}$ is assumed to be convex (actually a ball set) in this paper. One can refer to Fig. 6 for an illustration.

Moreover, we assume there are no redundant inequalities in $\mathcal{K}$, i.e., there is no $g_i$ such that $\mathcal{K} = \{\mathbf{x} \mid \mathbf{g}_{-i}(\mathbf{x}) \leq \mathbf{0}\}$ where $\mathbf{g}_{-i} = (g_1, \cdots, g_{i-1}, g_{i+1}, \cdots, g_m)$. In this case, any feasible point $\mathbf{x}$ satisfying $g_i(\mathbf{x}) = 0$ for some $i$ is on the boundary of the set $\mathcal{K}$. Thus, we have

$$\{\mathbf{x} \in \mathcal{K} \mid g_j(\mathbf{x}) = 0, g_k(\mathbf{x}) \neq 0\} \bigcap \{\mathbf{x} \in \mathcal{K} \mid g_k(\mathbf{x}) = 0, g_j(\mathbf{x}) \neq 0\} = \emptyset$$

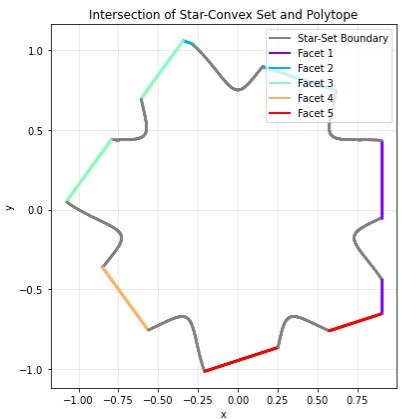 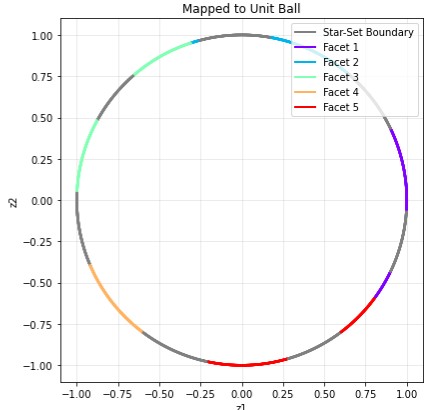

*Figure 6.* Illustration of the action of homeomorphism on a star-shaped set. The left figure shows the star-shaped constraints of problem (**P**). Each color of line represents the boundary characterized by a constraint inequality $\{\mathbf{a}_i^\top \mathbf{x} \leq b_i\}$ for some $i$. Under a homeomorphic mapping $\psi$, the constrained set is transformed to a ball (right figure). Each constraint inequality $\{G_i(\mathbf{z}) \leq 0\}$ (colored differently) is non-convex in general.

for any $k \neq j$. Note $\mathcal{B} = \{\mathbf{z} \mid G_i(\mathbf{z}) \leq 0, i = 1, 2, \cdots, m\} = \{\mathbf{z} \mid \|\mathbf{z}\|^2 \leq 1\}$. Moreover, $\{G_i(\mathbf{z}) \leq 0, i = 1, 2, \cdots, m\}$ also has no redundant constraints by the non-singularity of the Jacobian of $\psi$ and similarly,

$$\{\mathbf{z} \in \mathcal{B} \mid G_j(\mathbf{z}) = 0, G_k(\mathbf{z}) \neq 0\} \bigcap \{\mathbf{z} \in \mathcal{B} \mid G_k(\mathbf{z}) = 0\} = \emptyset$$

for any $j \neq k$. Hence if $\mathbf{z} \in \mathcal{B}$ satisfies $G_i(\mathbf{z}) = 0$ for some $i$, it lies on the boundary of $\mathcal{B}$. Clearly, we have

$$G_i(\mathbf{z}) = \|\mathbf{z}\|^2 - 1 \quad \text{at} \quad \mathbf{z}' \in \partial\mathcal{B}, G_i(\mathbf{z}') = 0, \tag{11}$$

and

$$\nabla G_i(\mathbf{z}) = 2\mathbf{z}, \nabla^2 G_i(\mathbf{z}) = 2\mathbf{I}_n \quad \text{at} \quad \mathbf{z}' \in \partial\mathcal{B}, G_i(\mathbf{z}') = 0. \tag{12}$$

where $\mathbf{I}_n$ is the identity matrix of $n$ by $n$.

**D.2. Properties of Function $h = f \circ \psi$**

**Lemma D.1** (Properties of $h = f \circ \psi$). *Under the general assumptions C.2, $h = f \circ \psi$ has the following properties.*

1) *$h$ is $L_{h,0} := L_{f,0}\kappa_2$ Lipschitz continuous.*

2) *$h$ is $L_h$-smooth with $L_h = \kappa_2^2 L_f + L_\psi L_{f,0}$.*

3) *If $f$ is convex, then $h$ is $\ell_h$-weakly convex with $\ell_h = L_{f,0}L_\psi$.*

*Proof.* We prove them one by one in the following.

1) We can directly derive from basic definitions:

$$\begin{aligned}
\|h(\mathbf{u}) - h(\mathbf{v})\| &\leq \|f(\psi(\mathbf{u})) - f(\psi(\mathbf{v}))\| \\
&\leq L_{f,0} \|\psi(\mathbf{u}) - \psi(\mathbf{v})\| \\
&\leq L_{f,0}L_\psi \|\mathbf{u} - \mathbf{v}\|.
\end{aligned}$$

2) From $L_f$-smoothness of $f$, we have

$$\|\nabla f(\mathbf{x}) - \nabla f(\mathbf{y})\| \leq L_f \|\mathbf{x} - \mathbf{y}\|. \tag{13}$$

Then we derive with $\mathbf{x} = \psi(\mathbf{z}), \mathbf{v} = \psi(\mathbf{y})$,

$$
\begin{aligned}
\|\nabla h(\mathbf{z}) - \nabla h(\mathbf{v})\| &= \left\| \mathbf{J}_{\psi}(\mathbf{z})^{\top} \nabla f(\mathbf{x}) - \mathbf{J}_{\psi}(\mathbf{v})^{\top} \nabla f(\mathbf{y}) \right\| \\
&= \left\| \mathbf{J}_{\psi}(\mathbf{z})^{\top} (\nabla f(\mathbf{x}) - \nabla f(\mathbf{y})) + (\mathbf{J}_{\psi}(\mathbf{z}) - \mathbf{J}_{\psi}(\mathbf{v}))^{\top} \nabla f(\mathbf{y}) \right\| \\
&\leq \left\| \mathbf{J}_{\psi}(\mathbf{z})^{\top} (\nabla f(\mathbf{x}) - \nabla f(\mathbf{y})) \right\| + \left\| (\mathbf{J}_{\psi}(\mathbf{z}) - \mathbf{J}_{\psi}(\mathbf{v}))^{\top} \nabla f(\mathbf{y}) \right\| \\
&\leq \kappa_2 L_f \|\psi(\mathbf{z}) - \psi(\mathbf{v})\| + L_{\psi} L_{f,0} \|\mathbf{z} - \mathbf{v}\| \\
&\leq \left( \kappa_2^2 L_f + L_{\psi} L_{f,0} \right) \|\mathbf{z} - \mathbf{v}\|.
\end{aligned}
$$

Let $L_h = \kappa_2^2 L_f + L_{\psi} L_{f,0}$. We have the conclusion.

3) One hope to show $h(\cdot) + \frac{\ell_h}{2} \| \cdot + \mathbf{v}\|^2$ is a convex function, i.e.,

$$
h(\mathbf{v}) + \frac{\ell_h}{2} \|\mathbf{v}\|^2 \geq h(\mathbf{z}) + \frac{\ell_h}{2} \|\mathbf{z}\|^2 + \langle \nabla h(\mathbf{z}) + \ell_h \mathbf{z}, \mathbf{v} - \mathbf{z} \rangle, \ \forall \mathbf{z}, \mathbf{v}.
$$

This is equivalent to show

$$
h(\mathbf{v}) + \frac{\ell_h}{2} \|\mathbf{v} - \mathbf{z}\|^2 \geq h(\mathbf{z}) + \langle \nabla h(\mathbf{z}), \mathbf{v} - \mathbf{z} \rangle, \ \forall \mathbf{z}, \mathbf{v}.
$$

We drive with $\mathbf{x} = \psi(\mathbf{z}), \mathbf{y} = \psi(\mathbf{v})$ as follows,

$$
\begin{aligned}
\langle \nabla h(\mathbf{z}), \mathbf{v} - \mathbf{z} \rangle &= \left\langle \nabla \mathbf{J}_{\psi}(\mathbf{z})^{\top} f(\mathbf{x}), \mathbf{v} - \mathbf{z} \right\rangle \\
&= \langle \nabla f(\mathbf{x}), \mathbf{J}_{\psi}(\mathbf{z})(\mathbf{v} - \mathbf{z}) \rangle \\
&= \langle \nabla f(\mathbf{x}), -\psi(\mathbf{v}) + \psi(\mathbf{z}) + \mathbf{J}_{\psi}(\mathbf{z})(\mathbf{v} - \mathbf{z}) \rangle + \langle \nabla f(\mathbf{x}), \psi(\mathbf{v}) - \psi(\mathbf{z}) \rangle \\
&\leq \|\nabla f(\mathbf{x})\| \cdot \|\psi(\mathbf{v}) - \psi(\mathbf{z}) - \mathbf{J}_{\psi}(\mathbf{z})(\mathbf{v} - \mathbf{z})\| + \langle \nabla f(\mathbf{x}), \mathbf{y} - \mathbf{x} \rangle \\
&\leq L_{f,0} L_{\psi} \|\mathbf{z} - \mathbf{v}\|^2 + f(\mathbf{y}) - f(\mathbf{x}) \\
&= L_{f,0} L_{\psi} \|\mathbf{z} - \mathbf{v}\|^2 + h(\mathbf{v}) - h(\mathbf{z})
\end{aligned}
$$

where the first inequality is from triangular inequality and the second inequality is from Lemma C.2 and the convexity of $f$.

$\square$

### D.3. KKT Conditions of Problem (P) and (H)

First, we recall some basic definitions.

Consider a general optimization problem

$$
\begin{aligned}
\min_{\mathbf{x} \in \mathbb{R}^n} \ & f(\mathbf{x}), \\
\text{s.t.} \quad & g_i(\mathbf{x}) \leq 0, \forall i = 1, 2, \cdots, m; \\
& q_i(\mathbf{x}) \leq 0, \forall i = 1, 2, \cdots, p.
\end{aligned} \tag{G}
$$

The Lagrangian function of problem (G) is defined as

$$
\mathcal{L}(\mathbf{x}, \boldsymbol{\lambda}, \boldsymbol{\nu}) = f(\mathbf{x}) + \sum_{i=1}^{m} \lambda_i g_i(\mathbf{x}) + \sum_{i=1}^{p} \nu_i q_i(\mathbf{x}).
$$

A triple $(\mathbf{x}, \boldsymbol{\lambda}, \boldsymbol{\nu})$ is said to satisfy the Karush–Kuhn–Tucker (KKT) condition of problem (G) if the following holds

$$
\begin{aligned}
\nabla f(\mathbf{x}) + \sum_{i=1}^{m} \lambda_i \nabla g_i(\mathbf{x}) + \sum_{j=1}^{p} \nu_j \nabla q_j(\mathbf{x}) &= \mathbf{0}, \\
q_j(\mathbf{x}) = 0, g_i(\mathbf{x}) \leq 0, \quad &\forall j \in [p], i \in [m]; \\
\boldsymbol{\lambda} \geq \mathbf{0}, \ \lambda_i g_i(\mathbf{x}) = 0, \quad &\forall i \in [m].
\end{aligned} \tag{14}
$$

where $\boldsymbol{\lambda}$ (or $\boldsymbol{\nu}$) is the dual variable corresponding to inequality (resp. equality) constraints.

**Definition D.2** (KKT stationary point)**.** A point $\mathbf{x}^*$ is said to be a KKT stationary point of (G) if there exists $\boldsymbol{\lambda}^* \in \mathbb{R}^m_{\geq 0}, \boldsymbol{\nu}^* \in \mathbb{R}^p$ such that $(\mathbf{x}^*, \boldsymbol{\lambda}^*, \boldsymbol{\nu}^*)$ satisfies KKT condition (14).

**Definition D.3** (Strict complementary slackness)**.** It is said that the strict complementary slackness condition holds for problem (G), if

$$\lambda_i^* > 0 \quad \text{for} \quad g_i(\mathbf{x}^*) = 0, \quad \forall i \in [m].$$

To define the second-order KKT condition for the optimization problems, we recall that the critical cone in the following.

**Definition D.4** (Critical cone)**.** Denote the feasible region of problem (G) as $\mathcal{G}$. Then the critical cone $\mathrm{C}_{\mathcal{G}}(\mathbf{x}^*)$ at $\mathbf{x}^*$ of problem (G) is defined as (Nocedal & Wright, 1999)

$$\mathbf{w} \in \mathrm{C}_{\mathcal{G}}(\mathbf{x}^*) \Leftrightarrow \begin{cases} \nabla q_i(\mathbf{x}^*)^\top \mathbf{w} = 0, & \text{for all } i \in [p], \\ \nabla g_i(\mathbf{x}^*)^\top \mathbf{w} = 0, & \text{for all } i \in \mathcal{A}(\mathbf{x}^*) \text{ with } \lambda_i^* > 0, \\ \nabla g_i(\mathbf{x}^*)^\top \mathbf{w} \geq 0, & \text{for all } i \in \mathcal{A}(\mathbf{x}^*) \text{ with } \lambda_i^* = 0. \end{cases}$$

Here $\boldsymbol{\lambda}^*$ is the Lagrangian multiplier of inequality constraints $g_i$ and $\mathcal{A}(\mathbf{x}^*)$ is the index of active constraints.

From the definition, the critical cone of problem (P) can be written as

$$\mathbf{w} \in \mathrm{C}_{\mathcal{K}}(\mathbf{x}^*) \Leftrightarrow \begin{cases} \nabla g_i(\mathbf{x}^*)^\top \mathbf{w} = 0, & \text{for all } i \in \mathcal{A}(\mathbf{x}^*) \text{ with } \lambda_i^* > 0, \\ \nabla g_i(\mathbf{x}^*)^\top \mathbf{w} \geq 0, & \text{for all } i \in \mathcal{A}(\mathbf{x}^*) \text{ with } \lambda_i^* = 0. \end{cases}$$

Moreover, if *strict complementary slackness* holds, the critical cone is simplified as

$$\mathrm{C}_{\mathcal{K}}(\mathbf{x}^*) = \{\mathbf{w} \in \mathbb{R}^n \mid \nabla g_i(\mathbf{x}^*)^\top \mathbf{w} = 0, \text{ for all } i \in \mathcal{A}(\mathbf{x}^*)\}.$$

Suppose *strict complementary slackness* holds for problem (P) and (H). Then, we can write KKT conditions for problem (P) and (H) in the following.

*First-order KKT conditions* on $\mathbf{x}^*$. The Lagrangian of (P) is

$$\mathcal{L}_{\mathrm{P}}(\mathbf{x}, \boldsymbol{\lambda}) = f(\mathbf{x}) + \sum_{i=1}^m \lambda_i g_i(\mathbf{x}).$$

The first-order KKT conditions of (P) are: there exists $\boldsymbol{\lambda}^*$ such that

$$\nabla f(\mathbf{x}^*) + \sum_{i=1}^m \lambda_i^* \nabla g_i(\mathbf{x}^*) = \mathbf{0}, \tag{15a}$$

$$g_i(\mathbf{x}^*) \leq 0, \quad i = 1, 2, \cdots, m \tag{15b}$$

$$\boldsymbol{\lambda}^* \geq \mathbf{0}, \ \lambda_i^* g_i(\mathbf{x}^*) = 0, \quad i = 1, 2, \cdots, m. \tag{15c}$$

*Second-order KKT conditions* on $\mathbf{x}^*$. It adds the following condition

$$\mathbf{w}^\top \nabla_{\mathbf{x}}^2 \mathcal{L}_{\mathrm{P}}(\mathbf{x}^*, \boldsymbol{\lambda}^*) \mathbf{w} \geq 0 \tag{16}$$

for any $\mathbf{w}$ satisfying $\mathbf{w}^\top \nabla g_i(\mathbf{x}^*) = 0$ with $i \in \mathcal{A}(\mathbf{x}^*)$.

*First-order KKT conditions* on $\mathbf{z}^*$. The Lagrangian of (H) is

$$\mathcal{L}_{\mathrm{H}}(\mathbf{z}, \nu) = h(\mathbf{z}) + \nu(\|\mathbf{z}\|^2 - 1).$$

The first-order KKT conditions of (H) are: there exists $\nu^*$ such that

$$\nabla h(\mathbf{z}^*) + 2\nu^* \mathbf{z}^* = \mathbf{0}, \tag{17a}$$

$$\|\mathbf{z}^*\|^2 \leq 1, \tag{17b}$$

$$\nu^* \geq 0, \ \nu^*(\|\mathbf{z}^*\|^2 - 1) = 0. \tag{17c}$$

*Second-order KKT condition* on $\mathbf{z}^*$. It will add the following condition.

$$\mathbf{d}^\top \nabla_{\mathbf{z}}^2 \mathcal{L}_{\mathrm{H}}(\mathbf{z}^*, \nu^*) \mathbf{d} \geq 0 \tag{18}$$

for any $\mathbf{d} \in \mathrm{C}_{\mathcal{B}}(\mathbf{z}^*)$. Here recall that

$$\mathrm{C}_{\mathcal{B}}(\mathbf{z}^*) = \begin{cases} \mathbb{R}^n, & \text{if } \mathbf{z}^* \in \mathrm{int}(\mathcal{B}), \\ \{\mathbf{d} : \mathbf{d}^\top \mathbf{z}^* = 0\}, & \text{if } \mathbf{z}^* \in \partial\mathcal{B}. \end{cases}$$

### D.4. Relationships of KKT Stationary Points between Problem (P) and (H)

**Lemma D.5.** *Suppose strict complementary slackness holds for both problem* (P) *and* (H). *We have that* $\mathbf{x}^*$ *is a KKT stationary point of* (P) *if and only if* $\mathbf{z}^*$ *is also a KKT stationary point of* (H) *where* $\mathbf{x}^* = \boldsymbol{\psi}(\mathbf{z}^*)$.

*Proof.* 1) First, we assume that $\mathbf{x}^*$ is a KKT stationary point of (P). By assumption, there exists $\boldsymbol{\lambda}^*$ such that the KKT condition holds (15) holds. Then we have

$$\mathrm{J}_{\boldsymbol{\psi}}(\mathbf{z}^*)^\top \nabla f(\mathbf{x}^*) + \sum_{i=1}^m \lambda_i^* \mathrm{J}_{\boldsymbol{\psi}}(\mathbf{z}^*)^\top \nabla g_i(\mathbf{x}^*) = \mathbf{0},$$

$$g_i(\boldsymbol{\psi}(\mathbf{z}^*)) \leq 0, \quad i = 1, 2, \cdots, m$$
$$\boldsymbol{\lambda}^* \geq \mathbf{0}, \ \lambda_i^* g_i(\boldsymbol{\psi}(\mathbf{z}^*)) = 0, \quad i = 1, 2, \cdots, m.$$

This is equivalent to

$$\nabla h(\mathbf{z}^*) + \sum_{i=1}^m \lambda_i^* \nabla G_i(\mathbf{z}^*) = \mathbf{0}, \tag{19a}$$

$$G_i(\mathbf{z}^*) \leq 0, \quad i = 1, 2, \cdots, m \tag{19b}$$
$$\boldsymbol{\lambda}^* \geq \mathbf{0}, \ \lambda_i^* G_i(\mathbf{z}^*) = 0, \quad i = 1, 2, \cdots, m. \tag{19c}$$

Let $\nu^* = \sum_{i=1}^m \lambda_i^*$. According to the eq. (11,12), eq. (19a) is actually

$$\nabla h(\mathbf{z}^*) + 2\nu^* \mathbf{z}^* = \mathbf{0}.$$

By assumption, eq. (19b) is equivalent to

$$\|\mathbf{z}^*\|^2 \leq 1.$$

Note that if $G_i(\mathbf{z}^*) < 0$ for all $i$, then $\boldsymbol{\lambda}^* = 0$ and thus $\nu^* = 0$. In this case, $\nu^*(\|\mathbf{z}^*\|^2 - 1) = 0$. If $\mathbf{z}^*$ makes at least one $G_i(\mathbf{z}^*) = 0$, then we have $\|\mathbf{z}^*\|^2 = 1$. In this case, we also have $\nu^*(\|\mathbf{z}^*\|^2 - 1) = 0$. Hence, eq. (19c) implies

$$\nu^* \geq 0, \nu^*(\|\mathbf{z}^*\|^2 - 1) = 0.$$

In conclusion, there exists $\mathbf{z}^*, \nu^*$ such that the KKT condition holds.

2) Now, we assume $\mathbf{z}^*, \nu^*$ satisfy KKT condition for problem (H), i.e.,

$$\nabla h(\mathbf{z}^*) + 2\nu^* \mathbf{z}^* = \mathbf{0},$$
$$\|\mathbf{z}^*\|^2 \leq 1,$$
$$\nu^* \geq 0, \ \nu^*(\|\mathbf{z}^*\|^2 - 1) = 0.$$

If $\mathbf{z}^* \in \mathrm{int}(\mathcal{B})$, then $G_i(\mathbf{z}^*) < 0$ for all $i$ and $\nu^* = 0$. In this case, there exists $\boldsymbol{\lambda}^* = 0$ such that the KKT condition with eq. (15) of problem (P) holds at $\mathbf{x}^* = \boldsymbol{\psi}(\mathbf{z}^*), \boldsymbol{\lambda}^* = \mathbf{0}$.

If $\mathbf{z}^* \in \partial\mathcal{B}$, then there exists at least one $i \in \{1, 2, \cdots, m\}$ such that $G_i(\mathbf{z}^*) = 0$ and $\nu^* > 0$ from strict complementary slackness. Denote $\mathcal{A} = \{i : G_i(\mathbf{z}^*) = 0\}$. Note we define $\lambda_i^* = 0$ if $i \notin \mathcal{A}$ and $\lambda_i^* = \nu^*/|\mathcal{A}|$. Then we have $\mathbf{z}^*, \boldsymbol{\lambda}^*$ such that eq. 19 holds which implies $\mathbf{x}^* = \boldsymbol{\psi}(\mathbf{z}^*), \boldsymbol{\lambda}^*$ make the KKT condition of problem (P) hold.

$\square$

**Lemma D.6.** *Suppose strict complementary slackness condition holds for both problem* (**P**) *and* (**H**). *Then* $\mathbf{x}^*$ *is a second-order KKT stationary point of* (**P**) *if and only if* $\mathbf{z}^* = \boldsymbol{\psi}^{-1}(\mathbf{x}^*)$ *is also a second-order KKT stationary point of* (**H**).

*Proof.* From Lemma D.5, there exists $\boldsymbol{\lambda}^*$ and $\nu^*$ such that $(\mathbf{x}^*, \boldsymbol{\lambda}^*)$ holds for first-order KKT condition of (**P**) if and only if $(\mathbf{z}^*, \nu^*)$ holds for first-order KKT condition of (**H**). Hence, it suffices to show the equivalence of condition 18 and 16.

1) Let's first suppose $\mathbf{x}^*$ is a second-order KKT stationary point, i.e., eq. (16) holds.

Note

$$\nabla_{\mathbf{z}}^2 \mathcal{L}_{\mathrm{H}}(\mathbf{z}^*, \nu^*) = \nabla^2 h(\mathbf{z}^*) + 2\nu^* \mathbf{I}_n,$$

where $\mathbf{I}_n$ is identity matrix of size $n \times n$. We just need to show $\mathbf{d}^\top \nabla \mathcal{L}_{\mathrm{H}}(\mathbf{z}^*, \nu^*)\mathbf{d} \geq 0$ for any $\mathbf{d} \in \mathrm{C}_{\mathcal{B}}(\mathbf{z}^*)$. Recall that

$$\nabla^2 h(\mathbf{z}^*) = \mathrm{J}_{\boldsymbol{\psi}}(\mathbf{z}^*)^\top \nabla^2 f(\boldsymbol{\psi}(\mathbf{z}^*))\mathrm{J}_{\boldsymbol{\psi}}(\mathbf{z}^*) + \sum_{i=1}^n \frac{\partial f}{\partial \mathbf{x}_i}(\boldsymbol{\psi}(\mathbf{z}^*))\nabla^2 \psi_i(\mathbf{z}^*),$$

and

$$\nabla^2 G_i(\mathbf{z}^*) = \mathrm{J}_{\boldsymbol{\psi}}(\mathbf{z}^*)^\top \nabla^2 g_i(\boldsymbol{\psi}(\mathbf{z}^*))\mathrm{J}_{\boldsymbol{\psi}}(\mathbf{z}^*) + \sum_{k=1}^n \frac{\partial g_i}{\partial \mathbf{x}_k}(\boldsymbol{\psi}(\mathbf{z}^*))\nabla^2 \psi_k(\mathbf{z}^*), \quad i = 1, 2, \cdots, m.$$

From eq. (11), note that

$$\nabla^2 G_k(\mathbf{z}^*) = 2\mathbf{I}_n, \forall k \in \mathcal{A}(\mathbf{x}^*) \cap \{k : G_k(\mathbf{z}^*) = 0\}.$$

From Lemma D.5, $\nu^* = \sum_i \lambda_i^*$. Then we have

$$\nabla^2 \mathcal{L}_{\mathrm{H}}(\mathbf{z}^*, \nu^*) = \nabla^2 h(\mathbf{z}^*) + \sum_{i=1}^m \lambda_i^* \nabla^2 G_i(\mathbf{z}^*) \tag{20a}$$

$$= \mathrm{J}_{\boldsymbol{\psi}}(\mathbf{z}^*)^\top \nabla^2 f(\boldsymbol{\psi}(\mathbf{z}^*))\mathrm{J}_{\boldsymbol{\psi}}(\mathbf{z}^*) + \sum_{i=1}^m \mathrm{J}_{\boldsymbol{\psi}}(\mathbf{z}^*)^\top \lambda_i^* \nabla^2 g_i(\boldsymbol{\psi}(\mathbf{z}^*))\mathrm{J}_{\boldsymbol{\psi}}(\mathbf{z}^*) \tag{20b}$$

$$+ \sum_{k=1}^n \frac{\partial f}{\partial \mathbf{x}_k}(\boldsymbol{\psi}(\mathbf{z}^*))\nabla^2 \psi_k(\mathbf{z}^*) + \sum_{k=1}^n \sum_{i=1}^m \lambda_i^* \frac{\partial g_i}{\partial \mathbf{x}_k}(\boldsymbol{\psi}(\mathbf{z}^*))\nabla^2 \psi_k(\mathbf{z}^*). \tag{20c}$$

From first-order KKT stationarity of (**P**), i.e.,

$$\nabla f(\mathbf{x}^*) + \sum_{i=1}^m \lambda_i^* \nabla g_i(\mathbf{x}^*) = \mathbf{0},$$

We have

$$\frac{\partial f}{\partial \mathbf{x}_k}(\mathbf{x}^*) + \sum_{i=1}^m \lambda_i^* \frac{\partial g_i}{\partial \mathbf{x}_k}(\mathbf{x}^*) = 0.$$

Hence for any $\mathbf{d} \in \mathrm{C}_{\mathcal{B}}(\mathbf{z}^*)$, we have the second term (20c) is equal to 0.

Now we note it's trivial that $\mathrm{C}_{\mathcal{K}}(\mathbf{x}^*) = \mathrm{C}_{\mathcal{B}}(\mathbf{z}^*) = \mathbb{R}^n$ if $\mathbf{z}^* \in \mathrm{int}(\mathcal{K})$ where $\mathbf{x}^* = \boldsymbol{\psi}(\mathbf{z}^*)$. Hence in this case if $\mathbf{d} \in \mathrm{C}_{\mathcal{B}}(\mathbf{z}^*)$, we will have $\mathrm{J}_{\boldsymbol{\psi}}(\mathbf{z}^*)\mathbf{d} \in \mathrm{C}_{\mathcal{K}}(\mathbf{x}^*)$

If $\mathbf{x}^* \in \partial \mathcal{K}$, then $\mathcal{A}(\mathbf{x}^*) \neq \emptyset$. For $\mathbf{d} \in \mathrm{C}_{\mathcal{B}}(\mathbf{z}^*)$, i.e., $\mathbf{d}^\top \mathbf{z}^* = 0$, we have

$$(\mathrm{J}_{\boldsymbol{\psi}}(\mathbf{z}^*)\mathbf{d})^\top \nabla g_i(\mathbf{x}^*) = \mathbf{d}^\top \mathrm{J}_{\boldsymbol{\psi}}(\mathbf{z}^*)^\top \nabla g_i(\mathbf{x}^*) = \mathbf{d}^\top G_i(\mathbf{z}^*) = 2\mathbf{d}^\top \mathbf{z}^* = 0, \quad \text{for } i \in \mathcal{A}(\mathbf{x}^*),$$

or $\mathrm{J}_{\boldsymbol{\psi}}(\mathbf{z}^*)\mathbf{d} \in \mathrm{C}_{\mathcal{K}}(\mathbf{x}^*)$.

So for $\mathbf{d} \in \mathrm{C}_{\mathcal{B}}(\mathbf{z}^*)$, we have the following holds about the first term of $\nabla^2 \mathcal{L}_{\mathrm{H}}(\mathbf{z}^*, \nu^*)$.

$$(\mathrm{J}_{\boldsymbol{\psi}}(\mathbf{z}^*)\mathbf{d})^\top \nabla^2 f(\boldsymbol{\psi}(\mathbf{z}^*))\mathrm{J}_{\boldsymbol{\psi}}(\mathbf{z}^*)\mathbf{d} + (\mathrm{J}_{\boldsymbol{\psi}}(\mathbf{z}^*)\mathbf{d})^\top \left(\sum_{i=1}^m \lambda_i^* \nabla^2 g_i(\boldsymbol{\psi}(\mathbf{z}^*))\right) \mathrm{J}_{\boldsymbol{\psi}}(\mathbf{z}^*)\mathbf{d} \geq 0$$

where the last '$\geq$' is from the assumption that $\mathbf{x}^*$ is the second-order KKT stationary point of (P). Hence, we have $\mathbf{d}^\top \nabla^2 \mathcal{L}_{\mathrm{H}}(\mathbf{z}^*, \nu^*)\mathbf{d} \geq 0$ for any $\mathbf{d} \in \mathrm{C}_{\mathcal{B}}(\mathbf{z}^*)$, i.e., $\mathbf{z}^* = \psi^{-1}(\mathbf{x}^*)$ is also a second-order KKT stationary point.

2) Let's suppose $\mathbf{z}^*$ is a second-order KKT stationary point and show that $\mathbf{x}^*$ is a second-order KKT stationary point.

If $\mathbf{z}^* \in \mathrm{int}(\mathcal{B})$, the proof is trivial because $\nu^* = 0$ according to the similar analysis. So we assume $\mathbf{z}^* \in \partial \mathcal{B}$. Define $\mathcal{A}(\mathbf{z}^*) = \{i : G_i(\mathbf{z}^*) = 0\}$, and $\lambda_i^* = 0$ for $i \notin \mathcal{A}(\mathbf{z}^*)$, $\lambda_i^* = \nu^*/|\mathcal{A}(\mathbf{z}^*)|$ for $i \in \mathcal{A}(\mathbf{z}^*)$.

Note for any $\mathbf{w} \in \mathrm{C}_{\mathcal{K}}(\mathbf{x}^*)$, we have

$$\mathbf{0} = \mathbf{w}^\top \nabla g_i(\mathbf{x}^*) = \mathbf{w}^\top \mathrm{J}_{\psi}^{-1}(\mathbf{z}^*)^\top \nabla G_i(\mathbf{z}^*) = (\mathrm{J}_{\psi}^{-1}(\mathbf{z}^*)\mathbf{w})^\top \mathbf{z}^*, \ \text{ for } \ i \in \mathcal{A}(\mathbf{x}^*) = \mathcal{A}(\mathbf{z}^*).$$

Hence $\mathrm{J}_{\psi}^{-1}(\mathbf{z}^*)\mathbf{w} \in \mathrm{C}_{\mathcal{B}}(\mathbf{z}^*)$. Then for any $\mathbf{w} \in \mathrm{C}_{\mathcal{K}}(\mathbf{x}^*)$,

$$\begin{aligned}
&\mathbf{w}^\top \nabla_{\mathbf{x}}^2 \mathcal{L}_{\mathrm{P}}(\mathbf{x}^*, \boldsymbol{\lambda}^*)\mathbf{w} \\
&= \mathbf{w}^\top \nabla^2 f(\mathbf{x}^*)\mathbf{w} + \mathbf{w}^\top \sum_{i=1}^{m} \lambda_i^* \nabla^2 g_i(\mathbf{x}^*)\mathbf{w} \\
&= (\mathrm{J}_{\psi}^{-1}(\mathbf{z}^*)\mathbf{w})^\top \mathrm{J}_{\psi}(\mathbf{z}^*)\nabla^2 f(\mathbf{x}^*)\mathrm{J}_{\psi}(\mathbf{z}^*)\mathrm{J}_{\psi}^{-1}(\mathbf{z}^*)\mathbf{w} \\
&\quad + (\mathrm{J}_{\psi}^{-1}(\mathbf{z}^*)\mathbf{w})^\top \mathrm{J}_{\psi}(\mathbf{z}^*)(\sum_{i=1}^{m} \lambda_i^* \nabla^2 g_i(\mathbf{x}^*))\mathrm{J}_{\psi}(\mathbf{z}^*)\mathrm{J}_{\psi}^{-1}(\mathbf{z}^*)\mathbf{w} \\
&\quad + (\mathrm{J}_{\psi}^{-1}(\mathbf{z}^*)\mathbf{w})^\top [\sum_{k=1}^{n} \frac{\partial f}{\partial \mathbf{x}_k}(\psi(\mathbf{z}^*))\nabla^2 \psi_k(\mathbf{z}^*) + \sum_{k=1}^{n}\sum_{i=1}^{m} \lambda_i^* \frac{\partial g_i}{\partial \mathbf{x}_k}(\psi(\mathbf{z}^*))\nabla^2 \psi_k(\mathbf{z}^*)]\mathrm{J}_{\psi}^{-1}(\mathbf{z}^*)\mathbf{w} \\
&= (\mathrm{J}_{\psi}^{-1}(\mathbf{z}^*)\mathbf{w})^\top \mathcal{L}_{\mathrm{H}}(\mathbf{z}^*, \nu^*)\mathrm{J}_{\psi}^{-1}(\mathbf{z}^*)\mathbf{w} \geq 0
\end{aligned}$$

where the sum of last term of the second '=' is exactly 0 and the last '$\geq$' is from the assumption that $\mathbf{z}^*$ is a second-order KKT stationary point.

$\square$

**Definition D.7** (Non-degenerate KKT stationary point). A second-order KKT point $\mathbf{x}^*$ of (P) is said to be non-degenerate if there exists $\boldsymbol{\lambda}^*$ such that

$$\mathbf{d}^\top \nabla^2 \mathcal{L}(\mathbf{x}^*, \boldsymbol{\lambda}^*)\mathbf{d} > 0$$

for all $0 \neq \mathbf{d} \in \mathrm{C}_{\mathcal{K}}(\mathbf{x}^*)$. Here the Lagrangian function is

$$\mathcal{L}(\mathbf{x}, \boldsymbol{\lambda}) = f(\mathbf{x}) + \sum_{i=1}^{m} \lambda_i g_i(\mathbf{x}).$$

**Lemma D.8.** *Suppose strict complementary slackness holds for problem (P) and (H). Then $\mathbf{x}^\star$ is a non-degenerate KKT point of optimization (P) if and only if $\mathbf{z}^\star$ satisfying $\mathbf{x}^\star = \psi(\mathbf{z}^\star)$ is also a non-degenerate KKT point of problem (H).*

*Proof.* 1) Suppose $\mathbf{x}^*$ is a non-degenerate KKT stationary point. Note that for $\mathbf{d} \in \mathrm{C}_{\mathcal{B}}(\mathbf{z}^*)$, we have $\mathrm{J}_{\psi}(\mathbf{z}^*)\mathbf{d} \in \mathrm{C}_{\mathcal{K}}(\mathbf{x}^*)$ from the proof of Lemma D.6. Moreover, from $\mathrm{J}_{\psi}(\mathbf{z}^*) \neq 0$ we have $\mathrm{J}_{\psi}(\mathbf{z}^*)\mathbf{d} \neq 0$ if and only if $\mathbf{d} \neq 0$. Then the conclusion is trivial from eq. (20) in the proof of Lemma D.6.

2) Now, we suppose $\mathbf{z}^*$ is a non-degenerate KKT stationary point. It follows from the proof of Lemma D.6 that for any $\mathbf{w} \in \mathrm{C}_{\mathcal{K}}(\mathbf{x}^*)$, we have $\mathrm{J}_{\psi}^{-1}(\mathbf{z}^*)\mathbf{w} \in \mathrm{C}_{\mathcal{B}}(\mathbf{z}^*)$. Hence, the conclusion is also trivial from the proof of item (2) of Lemma D.6.

$\square$

# E. Convergence Analysis

### E.1. Discussion on Assumption 2

In this section, we discuss the reasonability of Assumption 2 from both theoretical and practical perspectives, and further address its empirical verification.

Assumption 2 imposes bounds on the function approximation error and the Jacobian approximation error of the INN. Its reasonability is supported by the following considerations.

*Theoretical justification.* The achievability of small approximation errors follows from the *universal approximation property of invertible neural networks* (INNs). As discussed in Section 3.1, established theoretical results (Ishikawa et al., 2022) show that INNs with sufficient depth and width can approximate arbitrary smooth diffeomorphisms to arbitrary precision under the *Sobolev norm*. This universality implies that both the function approximation error and the Jacobian approximation error can, in principle, be made arbitrarily small, thereby providing a solid theoretical foundation for Assumption 2.

*Practical justification.* The two components of Assumption 2 are justified as follows.

- **Function approximation error.** The proposed INN training framework is specifically designed to minimize this error through the loss function in (6), which simultaneously enforces (a) $\Phi(\mathcal{B}) \subseteq \mathcal{K}$ via the constraint penalty term, and (b) maximal coverage of $\mathcal{K}$ via the volume maximization term. The empirical results in Appendix G.1 corroborate this design, showing that trained INNs achieve low Hausdorff distances and accurate geometric approximations across a variety of constraint sets. To practically verify it, as reported in Appendix G.1 (Figure 7), the trained INNs consistently achieve Hausdorff distances below 0.3 across all tested problem instances, confirming that this condition is satisfied in practice.

- **Jacobian approximation error.** While directly verifying this error is challenging due to the absence of ground-truth homeomorphisms, we remark that this assumption is only for the worst-case analysis. We remark that theoretical optimality bounds related to Jacobian approximation error may be loose; but experiments demonstrate optimality gaps as small as 2.9%, referring to Sec. 5.

With Assumption 2, we establish $O\left(\epsilon^{-2}\right)$ convergence for finding an approximate stationary point (Theorem 1), improving upon existing $O\left(\epsilon^{-3}\right)$ or $O\left(\epsilon^{-4}\right)$ rates (refer to Table 1). Our analysis explicitly accounts for INN approximation errors throughout optimization, providing bounds relating approximation quality to solution accuracy, which advance the state-of-the-art reparameterization optimization methods.

### E.2. Proof of Theorem 1

In this section, we show Theorem 1. Before moving on, we first introduce some definitions and notations below.

**Definition E.1** (Approximate KKT Stationary Point)**.** A point $\mathbf{x}^*$ is said to be an $\epsilon$-approximate KKT stationary point of **P** if there exists $\boldsymbol{\lambda}^* \in \mathbb{R}^m_{\geq 0}$ such that the strict complementary slackness holds and $(\mathbf{x}^*, \boldsymbol{\lambda}^*)$ satisfies

$$\left\| \nabla f(\mathbf{x}^*) + \sum_{i=1}^{m} \lambda_i^* \nabla g_i(\mathbf{x}^*) \right\| \leq \epsilon,$$
$$\| [\mathbf{g}(\mathbf{x}^*)]_+ \| \leq \epsilon, \tag{21}$$
$$\sum_{i=1}^{m} |\lambda_i^* g_i(\mathbf{x}^*)| \leq \epsilon.$$

**Definition E.2** (Approximate stationary point)**.** A point $\mathbf{x}^*$ is called $\epsilon$-stationary point for problem $\min_{\mathbf{x} \in \mathcal{K}} f(\mathbf{x})$ with convex set $\mathcal{K}$, if the gradient norm mapping

$$\operatorname{Gr}_f^{\mathcal{K}}(\mathbf{x}; \alpha) := \frac{1}{\alpha} [\mathbf{x} - \Pi_{\mathcal{K}}(\mathbf{x} - \alpha \nabla f(\mathbf{x}))]$$

satisfies $\| \operatorname{Gr}_f^{\mathcal{K}}(\mathbf{x}; \alpha) \| \leq \epsilon$ for proper $\alpha > 0$.

**Definition E.3** (Normal cone)**.** The normal cone $\operatorname{N}_S(\mathbf{x})$ of a closed and convex set $\mathcal{K}$ at $\mathbf{x} \in \mathcal{K}$ is defined as

$$\operatorname{N}_{\mathcal{K}}(\mathbf{x}) = \{ \mathbf{y} : \langle \mathbf{y}, \mathbf{z} - \mathbf{x} \rangle \leq 0 \text{ for any } \mathbf{z} \in \mathcal{K} \}.$$

**Notations.**

- Recall that $\psi$ is the exact homeomorphic mapping and $\Phi$ is the learned, approximate homeomorphic mapping. Thus, we denote $\mathcal{B} := \psi^{-1}(\mathcal{K})$ as a unit ball and $\tilde{\mathcal{B}} := \Phi^{-1}(\mathcal{K})$ as an approximate unit ball. Moreover, as Assumption 2 holds, we have

$$\|\mathrm{BP}_{\tilde{\mathcal{B}}}(\mathbf{z}) - \Pi_{\mathcal{B}}(\mathbf{z})\| \leq \epsilon_{\mathrm{inn}}.$$

- We denote $h := f \circ \psi$ and $H = f \circ \Phi$.

- We denote the bi-Lipschitz continuous constant of $\Phi$ as $l_\Phi$ and $u_\Phi$, i.e.,

$$l_\Phi \|\mathbf{u} - \mathbf{v}\| \leq \|\Phi(\mathbf{u}) - \Phi(\mathbf{v})\| \leq u_\Phi \|\mathbf{u} - \mathbf{v}\|. \tag{22}$$

Recall that the bi-Lipschitz continuous property of an INN composed of affine coupling layers is satisfied by its design (Prop. B.1). Under this condition, we have

$$\|\mathrm{J}_\Phi(\mathbf{z})\| \leq u_\Phi, \|\mathrm{J}_{\Phi^{-1}}(\mathbf{z})\| \leq \frac{1}{l_\Phi}.$$

We list some help lemmas first in the following.

**Lemma E.4.** *Suppose an error $\epsilon > 0$ is sufficiently small. Consider $\min_{\mathbf{z} \in \mathcal{B}} h(\mathbf{z})$. If $\|\mathrm{Gr}_h^{\mathcal{B}}(\mathbf{z}'; \alpha)\| \leq \epsilon$ for some $\mathbf{z}' \in \mathcal{B}$, then $\mathbf{z}'$ is an $\mathcal{O}(\epsilon)$-KKT stationary point of problem $\min_{\mathbf{z} \in \mathcal{B}} h(\mathbf{z})$. Specifically, there exists $\nu^*$ such that*

$$\|\nabla h(\mathbf{z}') + 2\nu^* \mathbf{z}'\| \leq \alpha(1 + \beta)\epsilon,$$
$$\|\mathbf{z}'\| - 1 \leq 0,$$
$$\nu^* \geq 0, |\nu^*(\|\mathbf{z}'\|^2 - 1)| \leq \beta\epsilon,$$

*where $\beta$ is a constant depending on $\mathbf{z}'$.*

*Proof.* Suppose $\mathbf{z}^+ = \Pi_{\mathcal{B}}(\mathbf{z}' - \alpha \nabla h(\mathbf{z}'))$ and $\mathrm{Gr}(\mathbf{z}') = \mathrm{Gr}_h^{\mathcal{K}}(\mathbf{z}'; \alpha)$ for conciseness of notation. Then $\mathrm{Gr}(\mathbf{z}') = \frac{1}{\alpha}(\mathbf{z}' - \mathbf{z}^+)$.

From the optimality of orthogonal projection (Prop. C.1), we have

$$\langle \mathbf{z}' - \alpha \nabla h(\mathbf{z}') - \mathbf{z}^+, \mathbf{z} - \mathbf{z}^+ \rangle \leq 0$$

for any $\mathbf{z} \in \mathcal{B}$. Let $\zeta = \mathbf{z}' - \mathbf{z}^+ - \alpha \nabla h(\mathbf{z}')$. We have $\zeta \in \mathrm{N}_{\mathcal{B}}(\mathbf{z}^+)$ by definition of the normal cone. Moreover, the normal cone of a unit ball can be written as

$$\mathrm{N}_{\mathcal{B}}(\mathbf{z}^+) = \{\beta \mathbf{z}^+ : \beta > 0\} \text{ for } \mathbf{z}^+ \in \partial \mathcal{B}; \text{ and } \mathrm{N}_{\mathcal{B}}(\mathbf{z}^+) = \{\beta \mathbf{z}^+ : \beta = 0\} \text{ for } \mathbf{z}^+ \in \mathrm{int}(\mathcal{B}).$$

Hence we have $\zeta = \beta \mathbf{z}^+$ for some $\beta \geq 0$, i.e.,

$$\alpha \nabla h(\mathbf{z}') + \beta \mathbf{z}^+ = \mathbf{z}' - \mathbf{z}^+.$$

Equivalently,

$$\nabla h(\mathbf{z}') + \frac{1}{\alpha}[\beta \mathbf{z}' + \beta(\mathbf{z}^+ - \mathbf{z}')] = \frac{1}{\alpha}[\mathbf{z}' - \mathbf{z}^+].$$

Thus,

$$\left\|\nabla h(\mathbf{z}') + \frac{\beta}{\alpha} \mathbf{z}'\right\| \leq (1 + \beta)\|\mathrm{Gr}(\mathbf{z}')\| \leq (1 + \beta)\epsilon.$$

By defining $\nu^* = \frac{\beta}{2\alpha} \geq 0$, we have

$$\|\nabla h(\mathbf{z}') + 2\nu^* \mathbf{z}'\| \leq (1 + \beta)\|\mathrm{Gr}(\mathbf{z}')\| \leq (1 + \beta)\epsilon.$$

Next, note that $\mathbf{z}'$ is feasible, thereby $\|\mathbf{z}'\| - 1 \leq 0$.

Finally, we show

$$|\nu^*(\|\mathbf{z}'\|^2 - 1)| \leq \beta\epsilon.$$

If $\mathbf{z}^+ \in \mathrm{int}(\mathcal{B})$, we have $\beta = 0$ by the definition of $\beta$, i.e., $\nu^* = 0$. In this case, the proof is trivial. Hence, we assume $\mathbf{z}^+ \in \partial \mathcal{B}$. It follows that $\|\mathbf{z}^+\|^2 = 1$. Then we have

$$|\nu^*(\|\mathbf{z}'\|^2 - 1)| = |\nu^*(\|\mathbf{z}'\|^2 - \|\mathbf{z}^+\|^2)| \leq 2\nu^* \|\mathbf{z}' - \mathbf{z}^+\| \leq 2\nu^* \alpha \|\mathrm{Gr}(\mathbf{z}')\| \leq \beta\epsilon.$$

$\square$

**Lemma E.5.** *Consider the optimization problem* $\mathbf{H}_{\mathrm{inn}}$*:* $\min_{\mathbf{z} \in \tilde{\mathcal{B}}} H(\mathbf{z})$*. Let* $\epsilon > 0$ *be a sufficiently small error and Assumption 2 holds. Suppose* $\{\mathbf{z}_k\}_{k \geq 0}$ *is a sequence generated by Hom-PGD$^+$ with step-size* $\alpha \in (0, \frac{1}{L_H}]$*. Then* $\{\mathbf{z}_k\}_{0 \leq k \leq K}$ *contains an point* $\mathbf{z}'$ *with* $K = \mathcal{O}(L_H \epsilon^{-2})$ *such that*

$$\| \mathrm{Gr}_H^{\mathcal{B}}(\mathbf{z}') \| \leq c\epsilon + \mathcal{O}(\sqrt{L_H \epsilon_{\mathrm{inn}}})$$

*where* $c$ *is a constant independent of* $\epsilon$ *that can be small arbitrarily.*

*Proof.* We denote $\mathbf{z}^+ = \Pi_{\mathcal{B}}(\mathbf{z} - \alpha \nabla H(\mathbf{z}))$ and $\mathbf{z}^- = \mathrm{BP}_{\tilde{\mathcal{B}}}(\mathbf{z} - \alpha \nabla H(\mathbf{z}))$. We know that $\|\mathbf{z}^+ - \mathbf{z}^-\| \leq \epsilon_{\mathrm{inn}}$. According to the $L_H$ smoothness of $H$, we have

$$H(\mathbf{z}^-) \leq H(\mathbf{z}) + \langle \nabla H(\mathbf{z}), \mathbf{z}^- - \mathbf{z} \rangle + \frac{L_H}{2} \|\mathbf{z} - \mathbf{z}^-\|^2$$

$$= H(\mathbf{z}) + \langle \nabla H(\mathbf{z}), \mathbf{z}^- - \mathbf{z}^+ \rangle + \langle \nabla H(\mathbf{z}), \mathbf{z}^+ - \mathbf{z} \rangle + \frac{L_H}{2} \|\mathbf{z} - \mathbf{z}^-\|^2.$$

From Prop. C.1, we have

$$\langle \mathbf{z} - \alpha \nabla H(\mathbf{z}) - \mathbf{z}^+, \mathbf{z} - \mathbf{z}^+ \rangle \leq 0,$$

i.e.,

$$\langle \nabla H(\mathbf{z}), \mathbf{z} - \mathbf{z}^+ \rangle \leq -\frac{1}{\alpha} \|\mathbf{z}^+ - \mathbf{z}\|^2.$$

Hence, we have

$$H(\mathbf{z}^-) \leq H(\mathbf{z}) + \langle \nabla H(\mathbf{z}), \mathbf{z}^- - \mathbf{z}^+ \rangle + \langle \nabla H(\mathbf{z}), \mathbf{z}^+ - \mathbf{z} \rangle + \frac{L_H}{2} \|\mathbf{z} - \mathbf{z}^-\|^2$$

$$\leq H(\mathbf{z}) + (\frac{L_H}{2} - \frac{1}{\alpha}) \|\mathbf{z}^+ - \mathbf{z}\|^2 + \frac{L_H}{2} \|\mathbf{z}^+ - \mathbf{z}^-\|^2 + \|\nabla H(\mathbf{z})\| \cdot \|\mathbf{z}^- - \mathbf{z}^+\|.$$

It follows that

$$H(\mathbf{z}_k) - H(\mathbf{z}_{k+1}) + \frac{L_H}{2} \epsilon_{\mathrm{inn}}^2 + L_{H,0} \epsilon_{\mathrm{inn}} \geq \alpha(1 - \frac{\alpha L_H}{2}) \| \mathrm{Gr}(\mathbf{z}_k) \|^2 \tag{23}$$

where we denote

$$\mathrm{Gr}(\mathbf{z}) := \mathrm{Gr}_H^{\mathcal{B}}(\mathbf{z}) = \frac{1}{\alpha} [\mathbf{z} - \Pi_{\mathcal{B}}(\mathbf{z} - \alpha \nabla H(\mathbf{z}))].$$

Let $M = \alpha(1 - \frac{\alpha L_H}{2})$. We sum up Eq. (23) from $k = 0$ to $k = K$, and then we have

$$H(\mathbf{z}_0) - H^* \geq H(\mathbf{z}_0) - H(\mathbf{z}_{K+1}) + (K+1)(\frac{L_H}{2} \epsilon_{\mathrm{inn}}^2 + L_{H,0} \epsilon_{\mathrm{inn}})$$

$$\geq M \sum_{k=1}^{K} \| \mathrm{Gr}(\mathbf{z}_k) \|^2 \geq (K+1) \| \mathrm{Gr}(\mathbf{z}') \|^2$$

where $\mathbf{z}' = \arg\min_{k=0,1,\cdots,K} \| \mathrm{Gr}(\mathbf{z}_k) \|$. It follows that

$$\| \mathrm{Gr}(\mathbf{z}') \| \leq \sqrt{\frac{H(\mathbf{z}_0) - H^*}{M(K+1)} + \frac{L_H}{2} \epsilon_{\mathrm{inn}}^2 + L_{H,0} \epsilon_{\mathrm{inn}}} = \mathcal{O}(\frac{1}{\sqrt{K}}) + \mathcal{O}(\sqrt{L_H \epsilon_{\mathrm{inn}}}).$$

With $K = \mathcal{O}(L_H \epsilon^{-2})$, we get the conclusion. $\qquad\square$

**Lemma E.6.** *If* $\mathbf{z}'$ *is a **feasible** $\epsilon$-approximate KKT point of problem* $\min_{\mathbf{z} \in \mathcal{B}} H(\mathbf{z}) = f \circ \Phi(\mathbf{z})$ *over a unit ball,i.e.,* $\mathbf{z} \in \mathcal{B}$*, then* $\mathbf{x}' = \Phi(\mathbf{z}')$ *is an* $(\epsilon/\min\{l_\Phi, 1\} + \mathcal{O}(\epsilon_{\mathrm{inn}}))$*-approximate KKT point of problem* $\mathbf{P}$*.*

*Proof.* Note that

$$\mathcal{B} := \{\|\mathbf{z}\|^2 - 1 \leq 0\} = \{G_i(\mathbf{z}) := g_i(\boldsymbol{\psi}(\mathbf{z})) \leq 0, i = 1, 2, \cdots, m\}$$

and

$$\tilde{\mathcal{B}} = \Phi^{-1}(\mathcal{K}) = \{Q_i(\mathbf{z}) := g_i(\Phi(\mathbf{z})) \leq 0, i = 1, 2, \cdots, m\}.$$

We derive

$$\|\nabla G_i(\mathbf{z}) - \nabla Q_i(\mathbf{z})\| = \|\mathrm{J}_{\boldsymbol{\psi}}(\mathbf{z})\nabla g_i(\boldsymbol{\psi}(\mathbf{z})) - \mathrm{J}_{\Phi}(\mathbf{z})\nabla g_i(\Phi(\mathbf{z}))\|$$
$$\leq \|\mathrm{J}_{\boldsymbol{\psi}}(\mathbf{z})\nabla g_i(\boldsymbol{\psi}(\mathbf{z})) - \mathrm{J}_{\boldsymbol{\psi}}(\mathbf{z})\nabla g_i(\Phi(\mathbf{z}))\| +$$
$$\|\mathrm{J}_{\boldsymbol{\psi}}(\mathbf{z})\nabla g_i(\Phi(\mathbf{z})) - \mathrm{J}_{\Phi}(\mathbf{z})\nabla g_i(\Phi(\mathbf{z}))\|$$
$$\leq L_{\boldsymbol{\psi},0}L_{g_i}\epsilon_{\mathrm{inn}} + L_{g_i,0}\epsilon_{\mathrm{inn}}.$$

By assumption, there exists $\nu' \geq 0$ such that

$$\|\nabla h(\mathbf{z}') + 2\nu'\mathbf{z}'\| \leq \epsilon,$$
$$\|\mathbf{z}'\|^2 - 1 \leq 0,$$
$$|\nu'(\|\mathbf{z}'\|^2 - 1)| \leq \epsilon.$$

First, we show it is a fact that there exists $\boldsymbol{\lambda}'$ such that

$$\|\nabla h(\mathbf{z}') + \sum_{i=1}^m \lambda_i'\nabla G_i(\mathbf{z}')\| \leq \epsilon,$$
$$G_i(\mathbf{z}') \leq 0, \quad i = 1, 2, \cdots, m$$
$$\boldsymbol{\lambda}' \geq \mathbf{0}, \ |\lambda_i'G_i(\mathbf{z}')| \leq \epsilon/|\mathcal{A}|, \quad i = 1, 2, \cdots, m.$$

where we define $\mathcal{A} := \{i \mid G_i(\mathbf{z}) = \|\mathbf{z}\|^2 - 1 \text{ at } \mathbf{z}'\}$ and denote $\lambda_i' := 0$ for $i \notin \mathcal{A}$ and $\lambda_i' := \nu'/|\mathcal{A}|$ for $i \notin \mathcal{A}$. Moreover, the second inequality is from the feasibility of $\mathbf{z}'$. Now, it is easy to check that the above approximate KKT condition holds.

Note that $G_i(\mathbf{z}') \leq 0$ implies $[G_i(\mathbf{z}^+)]_+ = 0$. Next, we derive the following.

$$\left\|\nabla h(\mathbf{z}') + \sum_{i=1}^m \lambda_i'\nabla Q_i(\mathbf{z}')\right\| \leq \left\|\nabla h(\mathbf{z}') + \sum_{i=1}^m \lambda_i'\nabla G_i(\mathbf{z}')\right\| + \left\|\sum_{i=1}^m \lambda_i'\nabla Q_i(\mathbf{z}') - \sum_{i=1}^m \lambda_i'\nabla G_i(\mathbf{z}')\right\|$$
$$\leq \epsilon + \mathcal{O}(\epsilon_{\mathrm{inn}}),$$
$$[Q_i(\mathbf{z}')]_+ \leq [G_i(\mathbf{z}')]_+ + [Q_i(\mathbf{z}') - G_i(\mathbf{z}')]_+ \leq L_{g_i,0}\epsilon_{\mathrm{inn}},$$
$$|\lambda_i'Q_i(\mathbf{z}')| \leq |\lambda_i'G_i(\mathbf{z}')| + |\lambda_i'(Q_i(\mathbf{z}') - G_i(\mathbf{z}'))| \leq \epsilon + \lambda_i'L_{g_i,0}\epsilon_{\mathrm{inn}}.$$

Moreover, we have

$$\left\|\nabla h(\mathbf{z}') + \sum_{i=1}^m \lambda_i'\nabla Q_i(\mathbf{z}')\right\| \leq \epsilon + \mathcal{O}(\epsilon_{\mathrm{inn}}),$$
$$\left\|[\mathbf{Q}(\mathbf{z}')]_+\right\| \leq \sum_{i=1}^m [Q_i(\mathbf{z}')]_+ \leq \sum_{i=1}^m \left([G_i(\mathbf{z}')]_+ + [Q_i(\mathbf{z}') - G_i(\mathbf{z}')]_+\right) \leq mL_{\mathbf{g},0}\epsilon_{\mathrm{inn}},$$
$$\sum_{i=1}^m |\lambda_i'Q_i(\mathbf{z}')| \leq \sum_{i=1}^m \left(|\lambda_i'G_i(\mathbf{z}')| + |\lambda_i'(Q_i(\mathbf{z}') - G_i(\mathbf{z}'))|\right)$$
$$= \sum_{i \in \mathcal{A}} |\lambda_i'G_i(\mathbf{z}')| + \sum_{i \notin \mathcal{A}} |\lambda_i'G_i(\mathbf{z}')| + \sum_{i=1}^m |\lambda_i'(Q_i(\mathbf{z}') - G_i(\mathbf{z}'))|$$
$$\leq \epsilon + \lambda_{\mathrm{max}}'|\mathcal{A}|L_{\mathbf{g},0}\epsilon_{\mathrm{inn}}$$

where $L_{\mathbf{g},0} = \max_{i=1,2,\ldots,m}\{L_{g_i,0}\}$ and $\lambda_{\mathrm{max}}' = \max_{i=1,2,\cdots,m}\{\lambda_i'\}$. Here in the second line we use the inequality $[a + b]_+ \leq [a]_+ + [b]_+$ for $a, b \in \mathbb{R}$.

That is $\mathbf{z}'$ is an $\epsilon + \mathcal{O}(\epsilon_{\mathrm{inn}})$-KKT points of problem H with homeomorphic mapping $\Phi$.

Next, we derive

$$\left\| \nabla f\left(\mathbf{x}'\right) + \sum_{i=1}^{m} \lambda_i' \nabla g_i\left(\mathbf{x}'\right) \right\| \leq \left\| \mathrm{J}_\Phi\left(\mathbf{z}'\right)^{-\top} \right\| \cdot \left\| \mathrm{J}_\Phi\left(\mathbf{z}'\right)^\top \nabla f\left(\mathbf{x}'\right) + \sum_{i=1}^{m} \lambda_i' \mathrm{J}_\Phi\left(\mathbf{z}'\right)^\top \nabla g_i\left(\mathbf{x}'\right) \right\|,$$

$$\leq \frac{\epsilon}{l_\Phi} + \mathcal{O}\left(\epsilon_{\mathrm{inn}}\right),$$

$$\left[\mathbf{g}\left(\mathbf{x}'\right)\right]_+ = \left[\mathbf{Q}\left(\mathbf{z}'\right)\right]_+ \leq \mathcal{O}\left(\epsilon_{\mathrm{inn}}\right),$$

$$\sum_{i=1}^{m} |\lambda_i' g_i\left(\mathbf{x}'\right)| = \sum_{i=1}^{m} |\lambda_i' Q_i\left(\mathbf{z}'\right)| \leq \epsilon + \mathcal{O}\left(\epsilon_{\mathrm{inn}}\right).$$

It follows that $\mathbf{x}' = \Phi(\mathbf{z}')$ is an $(\epsilon/\min\{1, l_\Phi\} + \mathcal{O}(\epsilon_{\mathrm{inn}}))$-approximate KKT point. $\qquad\square$

*Proof of Theorem 1.* This is the direct corollary of the above lemmas. From Lemma E.5, we have that Hom-PGD$^+$ can find an approximate stationary point $\mathbf{z}'$ such that

$$\|\mathrm{Gr}_H^{\mathcal{B}}(\mathbf{z}')\| \leq c\epsilon + \mathcal{O}(\sqrt{L_H \epsilon_{\mathrm{inn}}})$$

in $\mathcal{O}(L_H \epsilon^{-2})$ iterations.

Then, it follows from Lemma E.4 that $\mathbf{z}'$ is also an approximate KKT point of optimization $\min_{\mathbf{z}\in\mathcal{B}} H(\mathbf{z})$. Specifically, we have that there exists $\nu^* \in \mathbb{R}_{\geq 0}$

$$\|\nabla H(\mathbf{z}') + 2\nu^* \mathbf{z}'\| \leq \alpha(1 + \beta)c\epsilon + \mathcal{O}(\sqrt{L_H \epsilon_{\mathrm{inn}}}),$$

$$\|\mathbf{z}'\| - 1 \leq 0,$$

$$\nu^* \geq 0, |\nu^*(\|\mathbf{z}'\|^2 - 1)| \leq c\beta\epsilon + \mathcal{O}(\sqrt{L_H \epsilon_{\mathrm{inn}}}),$$

Finally, by Lemma E.6, $\mathbf{x}' = \Phi(\mathbf{z}')$ is an $\left[c\alpha(1 + \beta)\epsilon/\min\{1, l_\Phi\} + \mathcal{O}(\sqrt{L_H \epsilon_{\mathrm{inn}}})\right]$-approximate KKT point of problem **P**. By choosing appropriate $c$, e.g.,

$$c = \min\left\{\frac{l_\Phi}{\alpha(1 + \beta)}, 1\right\},$$

$\mathbf{x}' = \Phi(\mathbf{z}')$ becomes an $\left[\epsilon + \mathcal{O}(\sqrt{L_H \epsilon_{\mathrm{inn}}})\right]$-approximate KKT point of problem **P**. $\qquad\square$

# F. Experiments Setting

## F.1. Problem Formulations and Instance Generation

### F.1.1. NON-CONVEX QUADRATICALLY CONSTRAINED QUADRATIC PROGRAMMING

We consider the following non-convex QCQP problem:

$$\min_{L \leq \mathbf{x} \leq U} \quad \frac{1}{2}\mathbf{x}^\top \mathbf{Q}_0 \mathbf{x} + \mathbf{q}_0^\top \mathbf{x} + r_0, \tag{24}$$

$$\text{s.t.} \quad \frac{1}{2}\mathbf{x}^\top \mathbf{Q}_i \mathbf{x} + \mathbf{q}_i^\top \mathbf{x} + r_i \leq 0, \quad i = 1, \ldots, m, \tag{25}$$

where $\mathbf{x} \in [L, U]^n$ is the decision variable, $\mathbf{Q}_i \in \mathbb{R}^{n \times n}$ are symmetric matrices (not necessarily positive semidefinite), $\mathbf{q}_i \in \mathbb{R}^n$, and $r_i \in \mathbb{R}$.

**Instance Generation:** For the objective matrix $\mathbf{Q}_0$, we generate eigenvalues uniformly from $[-1, 1]$ to create a mix of positive and negative eigenvalues, ensuring non-convexity. We construct $\mathbf{Q}_0 = \mathbf{U}\mathrm{diag}(\boldsymbol{\lambda})\mathbf{U}^\top/n$, where $\mathbf{U}$ is a random orthogonal matrix obtained via QR decomposition of a standard Gaussian matrix, and $\boldsymbol{\lambda}$ contains the mixed eigenvalues. The linear term $\mathbf{p}$ is sampled from $\mathcal{N}(0, 1/n)$. For the constraint matrices $\{\mathbf{Q}_i\}_{i=1}^m$, eigenvalues are uniformly sampled from $[-1, 1]$ to maintain the non-convex structure across constraints. Each $\mathbf{Q}_i$ is constructed using the same eigendecomposition approach with independent random orthogonal matrices and normalized by $1/n$. The corresponding linear terms $\mathbf{p}_i$ are

*Table 5.* Network characteristics and DC-OPF formulation complexity for PGLib test cases

| **Power Grids** | **200-Bus** | **500-Bus** |
|---|---|---|
| **Network Topology** | | |
| Buses | 200 | 500 |
| Generators | 69 | 145 |
| Branches | 245 | 597 |
| **DC-OPF Formulation** | | |
| *Decision Variables* | | |
| Real Power Generation ($P_g$) | 69 | 145 |
| Voltage Angles ($\theta$) | 199 | 499 |
| **Total Variables** | **268** | **644** |
| *Equality Constraints* | | |
| Power Balance | 200 | 500 |
| *Inequality Constraints* | | |
| Generator Limits | 138 | 290 |
| Voltage Angle Limits | 398 | 998 |
| Line Flow Limits | 490 | 1194 |
| **Total Inequalities** | **1026** | **2482** |

sampled from $\mathcal{N}(0, 1/n)$. To ensure feasibility, we first generate a random initial point $\mathbf{x}_0 \sim \mathcal{N}(0, 0.1)$ and clip it to satisfy the box constraints with a margin of 0.1. The constraint bounds are then set as $b_i = \frac{1}{2}\mathbf{x}_0^\top \mathbf{Q}_i \mathbf{x}_0 + \mathbf{p}_i^\top \mathbf{x}_0 + \epsilon_i$, where $\epsilon_i \sim |\mathcal{N}(0,1)| \cdot 0.1$ provides a feasibility margin. This construction guarantees that $\mathbf{x}_0$ is feasible and ensures the problem has a non-empty feasible region. For the illustrative example, we sample a 2-dimensional instance with 2 quadratic constraints.

F.1.2. JOINT CHANCE CONSTRAINED DC OPTIMAL POWER FLOW

In electrical power systems, operators must satisfy stochastic demand while maintaining system reliability across multiple nodes simultaneously. This presents a challenging multi-constraint optimization problem under uncertainty, where violations at any node can compromise system-wide stability.

We first introduce the standard DC optimal power flow (DC-OPF) problem:

$$\min_{\mathbf{p}, \boldsymbol{\theta}} \quad \sum_{i=1}^{G} \left( c_i^q p_i^2 + c_i^l p_i \right), \tag{26}$$

$$\text{s.t.} \quad \mathbf{p}^{\min} \leq \mathbf{p} \leq \mathbf{p}^{\max}, \quad \boldsymbol{\theta}^{\min} \leq \boldsymbol{\theta} \leq \boldsymbol{\theta}^{\max}, \tag{27}$$

$$\mathbf{B}_{\text{bus}} \boldsymbol{\theta} = \mathbf{p} - \mathbf{d}, \tag{28}$$

$$\mathbf{B}_{\text{line}} \boldsymbol{\theta} \leq \mathbf{S}^{\max}, \tag{29}$$

where $\mathbf{p} \in \mathbb{R}^G$ is the power generation vector, $\boldsymbol{\theta} \in \mathbb{R}^B$ are voltage phase angles, and $\mathbf{d} \in \mathbb{R}^B$ is the demand vector. The matrices $\mathbf{B}_{\text{bus}} \in \mathbb{R}^{B \times B}$ and $\mathbf{B}_{\text{line}} \in \mathbb{R}^{L \times B}$ are the bus and line susceptance matrices, with $B$ buses, $L$ transmission lines, and $G$ generators. The vector $\mathbf{S}^{\max} \in \mathbb{R}^L$ denotes maximum line capacities.

To handle dependency between decision variables and uncertain parameters, we eliminate the slack bus from the system equations. Let $\tilde{\mathbf{B}}_{\text{bus}} \in \mathbb{R}^{(B-1) \times (B-1)}$ be the reduced bus susceptance matrix, and $\tilde{\mathbf{p}} \in \mathbb{R}^{G-1}$, $\tilde{\boldsymbol{\theta}} \in \mathbb{R}^{B-1}$, $\tilde{\mathbf{d}} \in \mathbb{R}^{B-1}$, $\tilde{\boldsymbol{\xi}} \in \mathbb{R}^{B-1}$ be the corresponding reduced vectors. The phase angles for non-slack buses are:

$$\tilde{\boldsymbol{\theta}}(\boldsymbol{\xi}) = \tilde{\mathbf{B}}_{\text{bus}}^{-1} \left( \tilde{\mathbf{p}} - \tilde{\mathbf{d}} - \tilde{\boldsymbol{\xi}} \right), \tag{30}$$

and the slack bus generation adjusts to maintain power balance:

$$p_s(\boldsymbol{\xi}) = \sum_{i \in \mathcal{N}} (d_i + \xi_i) - \sum_{j \in \mathcal{G} \backslash s} p_j, \tag{31}$$

where $\mathcal{N}$, $\mathcal{G}$, and $s$ denote the sets of all buses, generator buses, and the slack bus, respectively.

The joint chance-constrained optimal power flow (JCC-OPF) extends the deterministic DC-OPF to handle demand uncertainty $\boldsymbol{\xi}$ while ensuring system reliability:

$$\min_{\tilde{\mathbf{p}}} \quad \mathbb{E}_{\boldsymbol{\xi}} \left[ \sum_{i=1}^{G} \left( c_i^q p_i(\boldsymbol{\xi})^2 + c_i^l p_i(\boldsymbol{\xi}) \right) \right], \tag{32}$$

$$\text{s.t.} \quad \mathbb{P} \left( \begin{array}{c} \mathbf{p}^{\min} \le \mathbf{p}(\boldsymbol{\xi}) \le \mathbf{p}^{\max} \\ \boldsymbol{\theta}^{\min} \le \boldsymbol{\theta}(\boldsymbol{\xi}) \le \boldsymbol{\theta}^{\max} \\ \mathbf{B}_{\text{line}} \boldsymbol{\theta}(\boldsymbol{\xi}) \le \mathbf{S}^{\max} \end{array} \right) \ge 1 - \epsilon, \tag{33}$$

where $\epsilon \in (0, 1)$ is the prescribed violation probability. All operational constraints must be satisfied jointly with probability at least $1 - \epsilon$, ensuring comprehensive system reliability under uncertainty.

Given sampled scenarios $\boldsymbol{\xi}^{(k)} k = 1^N$, we have the Sample Average Approximation (SAA) for the chance constraints:

$$\frac{1}{N} \sum_{k=1}^{N} \mathbb{I} \left( \begin{array}{c} \mathbf{p}^{\min} \le \mathbf{p}(\boldsymbol{\xi}^{(k)}) \le \mathbf{p}^{\max} \\ \boldsymbol{\theta}^{\min} \le \boldsymbol{\theta}(\boldsymbol{\xi}^{(k)}) \le \boldsymbol{\theta}^{\max} \\ \mathbf{B}_{\text{line}} \boldsymbol{\theta}(\boldsymbol{\xi}^{(k)}) \le \mathbf{S}^{\max} \end{array} \right) \ge 1 - \epsilon, \tag{34}$$

where $\mathbb{I}(\cdot)$ is the indicator function that equals 1 if all constraints are satisfied and 0 otherwise.

To solve it exactly via an existing solver such as GUROBI, we can reformulate it using the mixed-integer formulations by introducing binary variables $z^{(k)} \in \{0, 1\}$ for each scenario::

$$\frac{1}{N} \sum_{k=1}^{N} z^{(k)} \ge 1 - \epsilon, \tag{35}$$

$$\mathbf{p}^{\min} - M(1 - z^{(k)}) \le \mathbf{p}(\boldsymbol{\xi}^{(k)}) \le \mathbf{p}^{\max} + M(1 - z^{(k)}), \quad k = 1, \dots, N, \tag{36}$$

$$\boldsymbol{\theta}^{\min} - M(1 - z^{(k)}) \le \boldsymbol{\theta}(\boldsymbol{\xi}^{(k)}) \le \boldsymbol{\theta}^{\max} + M(1 - z^{(k)}), \quad k = 1, \dots, N, \tag{37}$$

$$\mathbf{B}_{\text{line}} \boldsymbol{\theta}(\boldsymbol{\xi}^{(k)}) \le \mathbf{S}^{\max} + M(1 - z^{(k)}), \quad k = 1, \dots, N, \tag{38}$$

$$z^{(k)} \in \{0, 1\}, \quad k = 1, \dots, N, \tag{39}$$

where $z^{(k)}$ is a binary indicator that equals 1 if all constraints are satisfied for scenario $k$, and $M$ is a sufficiently large constant. This mixed-integer linear programming formulation provides a tractable approximation with convergence guarantees as $N$ increases.

**Instance Generation:** We use IEEE test systems from PGLIB (Babaeinejadsarookolaee et al., 2019), which provide standardized network topologies, transmission line parameters, generator characteristics, and baseline demand profiles for power system benchmarking. Uncertainty scenarios $\{\boldsymbol{\xi}^{(k)}\}_{k=1}^{N}$ are generated from multivariate normal distributions $\mathcal{N}(\mathbf{0}, \boldsymbol{\Sigma})$, where $\boldsymbol{\Sigma}$ captures spatial correlation in demand uncertainty. We construct $\boldsymbol{\Sigma}$ using an exponential decay model based on geographical distance: $\Sigma_{ij} = \sigma_i \sigma_j \exp\left(-\frac{d_{ij}}{\ell}\right)$, where $\sigma_i$ is the standard deviation of demand uncertainty at bus $i$ (set to 5% of nominal demand $d_i$), $d_{ij}$ is the electrical distance between buses $i$ and $j$ measured by the shortest path length in the network graph, and $\ell$ is the correlation length parameter that controls the spatial decay rate. We sample $\ell$ from $[1, 5]$ to generate instances with different correlation structures: small $\ell$ values produce localized correlations, while large $\ell$ values create system-wide correlated demand fluctuations.

### F.1.3. NON-UNIFORM ADVERSARIAL ATTACK

Adversarial robustness evaluation requires solving an inner maximization problem that searches for perturbations capable of changing a classifier's prediction. For a classifier $f_{\boldsymbol{\theta}}$ and loss function $\ell$, the adversarial perturbation for an input-label pair $(\mathbf{x}, y)$ is obtained by

$$\max_{\boldsymbol{\delta} \in \Delta(\mathbf{x})} \quad \ell\left(f_{\boldsymbol{\theta}}(\mathbf{x} + \boldsymbol{\delta}), y\right), \tag{40}$$

where $\Delta(\mathbf{x})$ denotes the allowed perturbation region. Standard attacks usually take $\Delta(\mathbf{x})$ as a uniform $\ell_p$ ball, which assigns the same perturbation budget to all input coordinates. This uniform modeling can be restrictive when features have different semantic roles, correlations, or scales, motivating non-uniform adversarial attacks (Erdemir et al., 2021).

In our experiment, we use the weighted perturbation set

$$\Delta(\mathbf{x}) = \left\{ \boldsymbol{\delta} \in \mathbb{R}^d : \|\mathbf{W}\boldsymbol{\delta}\|_p \leq \epsilon_p,\ 0 \leq \mathbf{x} + \boldsymbol{\delta} \leq 1 \right\}, \tag{41}$$

where $\mathbf{W}$ is a positive-definite weighting operator encoding channel-spatial structure and $\epsilon_p$ is the perturbation radius for the chosen norm. This set reduces to a standard $\ell_p$-type perturbation region when $\mathbf{W}$ is the identity, while a nontrivial $\mathbf{W}$ induces feature-dependent effective budgets. For $0 < p < 1$, the weighted $\ell_p$ constraint is non-convex, so the exact Euclidean projection required by standard PGD is difficult to compute.

However, this constraint is star-convex from the zero perturbation vector, thus admits a closed-form homeomorphism to the unit ball by gauge mapping (Liu et al., 2026). Hom-PGD$^+$ therefore optimizes Eq. (40) over the latent variable $\mathbf{z} \in \mathcal{B}$ and maps the final solution back to the perturbation space through $\boldsymbol{\delta} = \boldsymbol{\psi}_{\mathbf{W},p}(\mathbf{z})$.

**Instance Generation:** We evaluate the attack on CIFAR-10 with a trained CNN classifier. For each clean image $\mathbf{x} \in [0,1]^{32 \times 32 \times 3}$, the perturbation dimension is $d = 32 \times 32 \times 3$. We construct $\mathbf{W}$ to combine inter-channel correlation and spatial smoothing, so nearby pixels and color channels receive correlated perturbation weights. We use a shared fixed subset of $N = 500$ correctly classified test images and evaluate $p \in \{0.5, 1, 2, 4\}$, covering both non-convex ($p < 1$) and convex ($p \geq 1$) perturbation regions. All attacks are run for 500 iterations with Adam step size $\alpha = 10^{-3}$. The quantitative results are reported in Table 3, and additional implementation details and qualitative examples are provided in Appendix G.4.

### F.2. Baseline Algorithms and Hyper-Parameters

We implement the baselines as follows:

- **EPM** : Exact Penalty Method (Cartis et al., 2011). It solves an unconstrained reformulated problem of (**P**) as follows

$$\min_{\mathbf{x}} f(\mathbf{x}) + \rho\|\mathbf{g}(\mathbf{x})\| \tag{42}$$

where $\rho$ is the penalty parameter. Moreover, for a large enough parameter $\rho$, the critical points of the unconstrained reformulation (42) correspond to the KKT stationary points of the original problem (**P**), provided by usual constraint qualifications (Nocedal & Wright, 1999). Based on this reformulation, one can use any appropriate algorithm to solve (42), such as gradient descent methods, trust region methods (Cartis et al., 2011).

- **ALM**: Augmented Lagrangian Methods (Sahin et al., 2019; Xie & Wright, 2019; Birgin et al., 2003).

$$\mathbf{x}_{k+1} = \arg\min_{\mathbf{x}} \{f(\mathbf{x}) + \boldsymbol{\lambda}_k^T \mathbf{g}(\mathbf{x}) + \rho_k\|[\mathbf{g}(\mathbf{x})]_+\|^2\}, \tag{43}$$

$$\boldsymbol{\lambda}_{k+1} = [\boldsymbol{\lambda}_k + \rho_k \cdot \mathbf{g}(\mathbf{x}_{k+1})]_+, \tag{44}$$

where $\boldsymbol{\lambda}_k$ is the Lagrange multipliers, $\mathbf{g}(\mathbf{x})$ represents the constraint functions, and $\rho_k > 0$ is the dual step size. The inner unconstrained optimization problem is non-convex due to the non-convexity of the constraint functions $\mathbf{g}$ and is solved using gradient descent to a stationary point, making it an inexact method.

- **PPP** : Proximal-Point Penalty Method (Lin et al., 2022). For the optimization (**P**), let

$$\phi_k(\mathbf{x}) := f(\mathbf{x}) + \frac{\gamma_k}{2}\|\mathbf{x} - \mathbf{x}_k\|^2 + \frac{\beta_k}{2}\left(\|[\mathbf{g}(\mathbf{x})]_+\|^2\right), \tag{45}$$

where $\beta_k > 0$ is the penalty parameter and $\gamma_k > 0$ is the proximal parameter. A sufficiently large parameter $\gamma_k$ will make the problem (45) a strongly convex optimization provided by the weakly convex constraints $\mathbf{g}$.

In each iteration, one will solve the problem (45) to a stationary point using (sub)gradient descent by fining $\mathbf{x}_{k+1}$ such that

$$\|\nabla\phi_k(\mathbf{x}_{k+1})\| \leq \hat{\varepsilon}_k \tag{46}$$

given a desired error $\hat{\epsilon}_k > 0$.

- **Hom-PGD$^+$**. Given the reformulated problem ($\mathbf{H}_{\text{inn}}$) and a step-size $\alpha_k$ in each iteration, we update by the rules

$$\mathbf{z}_{k+1} = \text{BP}_{\tilde{\mathcal{B}}}(\mathbf{z}_k - \alpha_k \nabla f(\Phi(\mathbf{z}_k))) \tag{47}$$

  where BP denotes the bisected projection onto the approximate unit ball $\tilde{\mathcal{B}}$, and $\Phi$ is the INN-learned homeomorphism. The solution is mapped to the original space after convergence as $\mathbf{x}^* = \mathbf{\Phi}(\mathbf{z}^*)$.

- **IPOPT**: Interior Point Optimizer, a state-of-the-art nonlinear programming solver that implements a primal-dual interior point method with line search. It uses exact second-order information and adaptive barrier parameter updates to handle inequality constraints through logarithmic barrier functions. IPOPT is particularly effective for large-scale continuous optimization problems with smooth nonlinear constraints.

- **GUROBI**: Commercial mixed-integer programming solver that employs branch-and-bound algorithms with advanced cutting plane generation, presolving techniques, and heuristics. For the SAA formulation of the JCC-OPF problem, GUROBI provides the exact optimal solution to the mixed-integer linear program, serving as the ground truth baseline for comparison with other approximate methods.

**Gradient calculation**: For simple quadratic objective functions, gradients are calculated via closed-form formulations. Other non-trivial gradient calculations across the various algorithms are implemented using auto-differentiation in PyTorch. We note that replacing auto-differentiation with closed-form gradient implementations could further improve the computational efficiency of the algorithms.

**Handing Non-differentiable Chance Constraint**: Since the indicator-based chance constraint is non-differentiable, making direct application of all first-order algorithms challenging. To tackle this challenge, we compute the robust scenario penalty following (Nemirovski & Shapiro, 2006), which computes the constraint violation for the worst-case scenario and treats it as a penalty in INN training or as the constraint violation/residual/penalty for other first-order algorithms. Specifically, we replace the non-differentiable indicator function with a smooth approximation:

$$\frac{1}{N} \sum_{k=1}^{N} \mathbb{I}(\mathbf{g}(\mathbf{x}, \boldsymbol{\xi}_k) \leq 0) \geq 1 - \epsilon \quad \Rightarrow \quad \max_{k \in \{1, \dots, N\}} [\mathbf{g}(\mathbf{x}, \boldsymbol{\xi}^{(k)})]_+ \leq 0 \tag{48}$$

Notably, when evaluating the chance constraint feasibility, we still follow the exact indicator-based formulation, which is used in the membership oracle for our Hom-PGD$^+$ method to ensure accurate feasibility assessment during optimization or the final evaluation for solutions obtained from different algorithms.

**Step-size**: Theoretically, different algorithms employ their own step size selection strategies, such as explicit dependence on smoothness and convexity parameters, or implicit step sizes that depend on the optimal objective value (Grimmer, 2024b). For practical implementation, we initialize a fixed step size (e.g., $10^{-3}$) and decay it by a factor of 0.999 if the objective value does not decrease, which helps identify a sufficient step size for convergence.

**Computation environment**: All algorithms are implemented in Pytorch and executed on an Ubuntu server with an NVIDIA A800 GPU and an AMD EPYC 7763 64-Core Processor.

### F.3. Invertible Neural Network Implementation

We adopt the coupling layer-based INN as our homeomorphism approximator. Specifically, it consists of 3 layers, each layer containing two sub-layers:

- **Invertible Linear Layers**: Following the GLOW architecture (Kingma & Dhariwal, 2018), we employ invertible linear layers with learnable bias terms. These layers implement affine transformations of the form $\mathbf{y} = \mathbf{W}\mathbf{x} + \mathbf{b}$, where the weight matrix $\mathbf{W}$ is constrained to be invertible through LU decomposition parameterization. This parameterization ensures invertibility by construction while allowing efficient computation of the log-determinant of the Jacobian as the sum of logarithms of the diagonal elements from the decomposition.

- **Coupling layer**: We implement coupling layers using MADE (Masked Autoencoder for Distribution Estimation) (Germain et al., 2015), which enables highly efficient computation through masked forward propagation. MADE applies element-wise affine transformations in an autoregressive manner, where each output dimension is conditioned on all preceding input dimensions according to a predefined ordering. This structure maintains the coupling layer property while providing computational efficiency through parallelizable masked operations.

**Conditional Embedding**: To incorporate conditional input $\boldsymbol{\theta}$, we employ a dedicated fully connected neural network that embeds the conditional information into a latent representation. This embedding is then added to the intermediate variables at each coupling layer, allowing the transformation to adapt based on the conditioning information. For the scenario-based input in JCC-DC-OPF, where the number of scenarios can vary across problem instances, we adopt a DeepSet-based architecture (Zaheer et al., 2017) to handle the permutation invariance property inherent in scenario sets. The DeepSet encoder maps variable-size scenario collections into a fixed-dimensional embedding space (64 dimensions in our implementation), ensuring consistent representation regardless of the number of scenarios while preserving the exchangeability of individual scenarios.

**INN Training**: We apply the Adam (Kingma & Ba, 2014) optimizer to train the INN with a batch size of 64, where each batch is sampled from the unit ball and input parameter space. We set the initial learning rate to $5 \times 10^{-4}$ with a decay factor of 0.9 every 1,000 iterations. The maximum number of training iterations is set to 10,000. The coefficient for the penalty term is 10, and the Lipschitz regularizer is 0.1.

## G. Supplementary Experiments Results

### G.1. INN training details

We provide training details for the invertible neural network used in homeomorphism learning. Specifically, we examine the convergence behavior of the training loss components and demonstrate the network's ability to learn bidirectional mappings between unit balls and constraint sets.

**Training Convergence**: The INN is trained by optimizing three loss components: the volume term (ensuring volume preservation), the penalty term (enforcing constraints), and the Lipschitz term (controlling smoothness). Figure 7 (top) shows the convergence of these components across different sampled input parameters $\theta$, demonstrating stable optimization. The training dynamics include three stages:

- Initialization phase: The INN parameters are randomly initialized (e.g., Gaussian), causing the initial mapping output $\Phi(\mathcal{B})$ to violate the constraint $\Phi(\mathcal{B}) \subseteq \mathcal{K}$. This results in a large constraint penalty term that dominates the total loss (as evident in the second subfigure showing high penalty loss).

- Shrinking phase: To reduce constraint violations, the network learns to shrink the mapped region and adjust its position. This shrinking decreases the volume (and thus log-volume drop), while it also reduces the constraint penalty by pushing $\Phi(\mathcal{B})$ fits within $\mathcal{K}$. During this phase, minimizing the penalty term takes priority over maximizing volume.

- Expansion phase: Once the constraint is approximately satisfied (indicated by low penalty loss in the second subfigure), the volume maximization term becomes dominant. The network then learns to expand $\Phi(\mathcal{B})$ to occupy as much of $\mathcal{K}$ as possible, ultimately approaching a homeomorphism approximately.

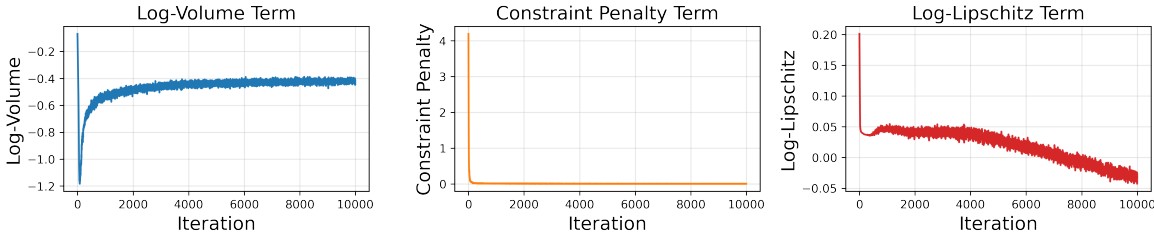

*Figure 7.* Training and evaluation of the 3-layer INN: convergence of the volume term, penalty term, and Lipschitz term across different sampled input parameters $\theta$ during training. The training algorithm stably learns homeomorphisms by maximizing volume within constraints while regularizing the Lipschitz constant, demonstrating effective approximation quality and capturing the complex constraint geometry under unseen input parameters.

**Learned Mapping Properties**: The trained INN learns parameter-dependent bidirectional mappings. In the forward direction, it maps the unit ball to constraint sets that vary with the input parameter $\theta$. In the inverse direction, it maps points from these constraint sets back to the unit ball, providing a normalized representation of the feasible region.

- Assumption 2 requires bounded homeomorphism error, meaning the trained INN must approximate the true home-omorphism between the unit ball and the constraint set with bounded error $\epsilon_{\text{inn}}$. Due to the bijective property of homeomorphisms, this is equivalent to requiring that $\Phi(\mathcal{B})$ closely approximates the true constraint set $\mathcal{K}$ (or equivalently, that $\Phi^{-1}(\mathcal{K})$ approximates $\mathcal{B}$). For straightforward visualization and comparison, we validate the forward direction by examining how well $\Phi(\mathcal{B})$ covers and matches the true constraint set $\mathcal{K}$.

- As shown in Figure 7, the mapped set $\Phi(\mathcal{B})$ accurately approximates the non-convex geometry of the target constraint set under different input parameters, demonstrating the effectiveness of our INN training method. To quantify this approximation quality, we can compute the Hausdorff distance between $\Phi(\mathcal{B})$ and $\mathcal{K}$, defined as

$$d_{\text{H}}(\Phi(\mathcal{B}), \mathcal{K}) = \max\left\{\sup_{\mathbf{x}\in\Phi(\mathcal{B})}\inf_{\mathbf{y}\in\mathcal{K}}\|\mathbf{x}-\mathbf{y}\|, \sup_{\mathbf{y}\in\mathcal{K}}\inf_{\mathbf{x}\in\Phi(\mathcal{B})}\|\mathbf{x}-\mathbf{y}\|\right\},$$

which measures the maximum distance between the two sets. if $d_H(\Phi(\mathcal{B}), \mathcal{K}) = 0$, then $\Phi(\mathcal{B}) = \mathcal{K}$ given $\mathcal{B} \cong \mathcal{K}$, meaning INN $\Phi$ is a perfect homeomorphic mapping between $\mathcal{B}$ and $\mathcal{K}$ and $\epsilon_{\text{inn}} = 0$.

### G.2. Ablation Study

We conduct ablation studies on QCQP optimization problems to analyze two key aspects of our method: **(i)** *INN Complexity and Performance*: We examine how INN depth (1/3/5 layers) affects approximation error (Assumption 2), Lipschitz constants, and downstream optimization performance, demonstrating that a 3-layer INN achieves the best balance between approximation capability and parameter complexity. **(ii)** *Bisection Complexity and Performance*: We show that reducing bisection iterations decreases per-iteration cost but may increase the optimality gap.

### G.3. More QCQP results

We visualize the comparison of Hom-PGD$^+$ and other baseline methods on QCQP optimization under different input parameters. We show the convergence with respect to iteration and total time, the constraint violation with respect to running time and per-iteration cost, and visualize the iteration trajectory of different methods.

### G.4. Adversarial Attack Experimental Details

**Feasible perturbation set.** For each clean input $\mathbf{x} \in [0, 1]^d$, perturbations are constrained by

$$\Delta(\mathbf{x}) = \left\{\boldsymbol{\delta} \in \mathbb{R}^d : \|\mathbf{W}\boldsymbol{\delta}\|_p \leq \epsilon_p,\ 0 \leq \mathbf{x} + \boldsymbol{\delta} \leq 1\right\},$$

where $\mathbf{W}$ is a positive-definite weighting operator that encodes channel-spatial structure by combining inter-channel correlation and spatial smoothing, and $\epsilon_p$ is preset for each norm level. The setting generalizes standard uniform $\ell_p$ attacks by assigning non-uniform effective perturbation budgets across input directions.

**PGD baseline.** The PGD baseline used in Table 3 does not compute the exact Euclidean projection onto $\Delta(\mathbf{x})$. Instead, after each gradient step it applies radial feasibility scaling:

$$\boldsymbol{\delta} \leftarrow s\,\boldsymbol{\delta}, \qquad s = \min\{s_{\text{ball}}, s_{\text{box}}, 1\},$$

where $s_{\text{ball}}$ enforces $\|W(s\boldsymbol{\delta})\|_p \leq \epsilon_p$ and $s_{\text{box}}$ enforces $0 \leq \mathbf{x} + s\boldsymbol{\delta} \leq 1$. Thus, the baseline uses a feasible-direction normalized update rather than the exact projector $\Pi_{\Delta(\mathbf{x})}(\cdot)$.

**Hom-PGD$^+$ implementation.** Hom-PGD$^+$ optimizes a latent variable $\mathbf{z}$ in the unit $\ell_2$ ball and maps it to a perturbation $\boldsymbol{\delta} \in \Delta(\mathbf{x})$ through the closed-form homeomorphic transformation. This enforces feasibility by construction in the perturbation space.

**Evaluation setting.** We evaluate on CIFAR-10 using a shared fixed subset of $N = 500$ correctly classified samples. All attacks use `adam` with step size $\alpha = 10^{-3}$ for 500 iterations. The success rate is computed as

$$\frac{\#\text{clean-correct} - \#\text{adv-correct}}{\#\text{clean-correct}} \times 100\%.$$

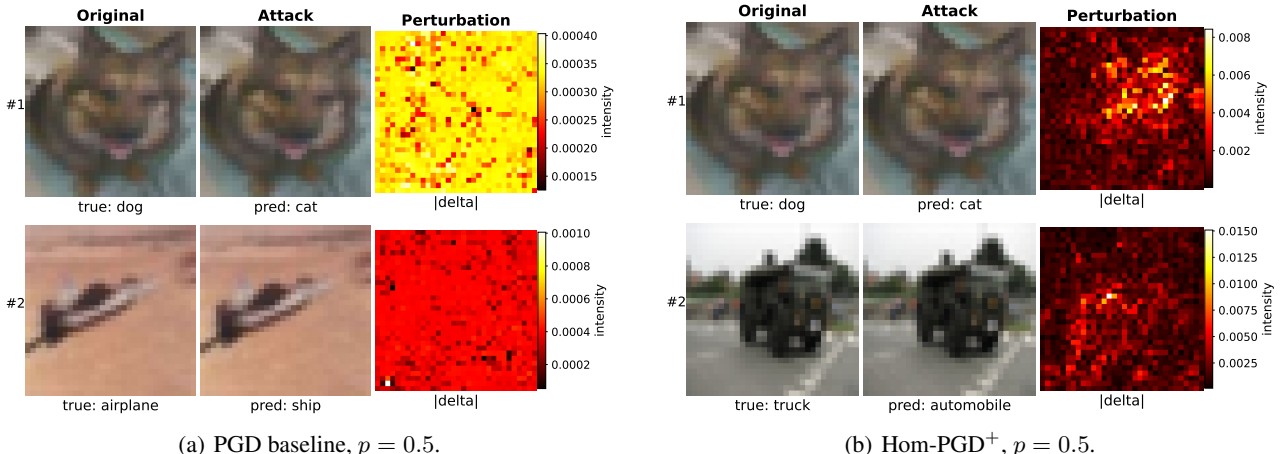

(a) PGD baseline, $p = 0.5$.        (b) Hom-PGD$^+$, $p = 0.5$.

*Figure 12.* Qualitative adversarial examples on CIFAR-10 under weighted non-uniform $\ell_{1/2}$ perturbation constraints. Each panel shows sampled successful attacks for the corresponding method. In this non-convex perturbation region, PGD with radial scaling only rescales each update along its current direction to restore feasibility, which can limit the search for stronger feasible perturbations due to the non-convex constraint geometry. Thus, PGD achieves a lower attack success rate in the weighted $\ell_{1/2}$-ball case.

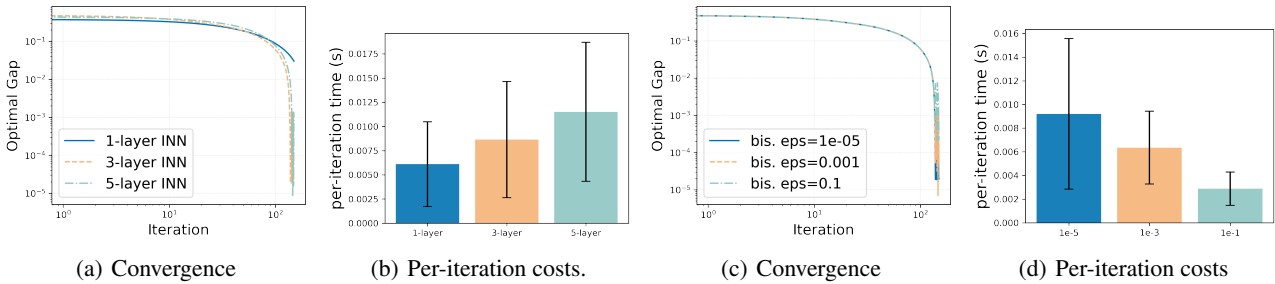

(a) Convergence     (b) Per-iteration costs.     (c) Convergence     (d) Per-iteration costs

*Figure 8.* Performance comparison of Hom-PGD$^+$ across different INN architectures (1-layer, 3-layer, 5-layer) and different bisection tolerance levels ($10^{-5}$, $10^{-3}$, $10^{-1}$). **a-b**: Single-layer INNs exhibit poor approximation capability, leading to large learning errors when approximating the constraint set. In contrast, 3-layer and 5-layer INNs provide sufficient representational capacity to capture the constraint set and demonstrate superior convergence behavior. **c-d**: Higher tolerance values accelerate the algorithm by reducing bisection iterations within the projection operator, but result in larger optimality gaps due to less precise convergence to the constraint boundary.

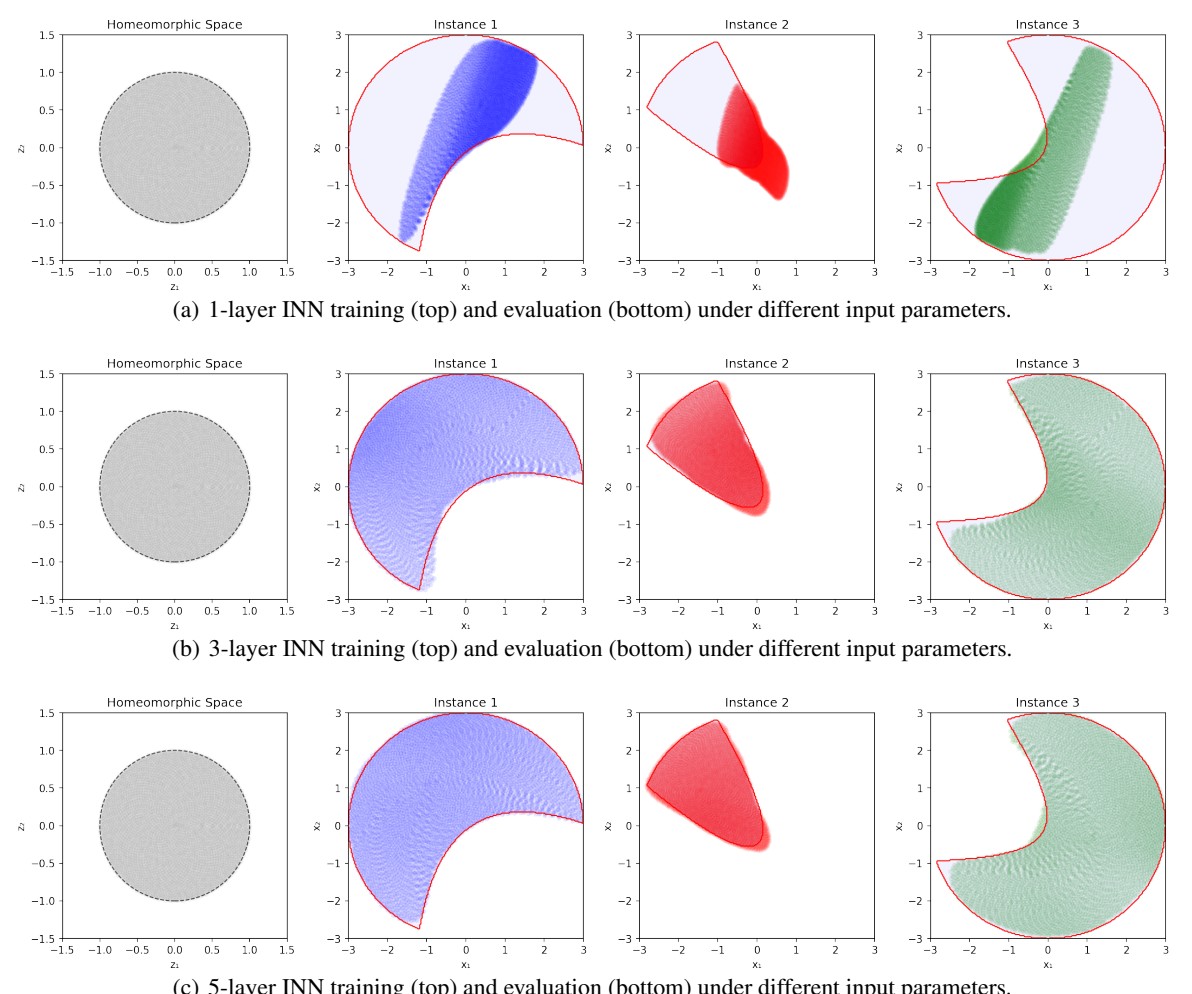

(a) 1-layer INN training (top) and evaluation (bottom) under different input parameters.

(b) 3-layer INN training (top) and evaluation (bottom) under different input parameters.

(c) 5-layer INN training (top) and evaluation (bottom) under different input parameters.

*Figure 9.* INN training and evaluation across different network depths. Panels visualize learned mappings under different input parameters. The 1-layer INN fails to capture constraint geometry accurately (average Hausdorff distance > 1.5), while 3-layer and 5-layer INNs achieve better approximation quality (average Hausdorff distance < 0.3).

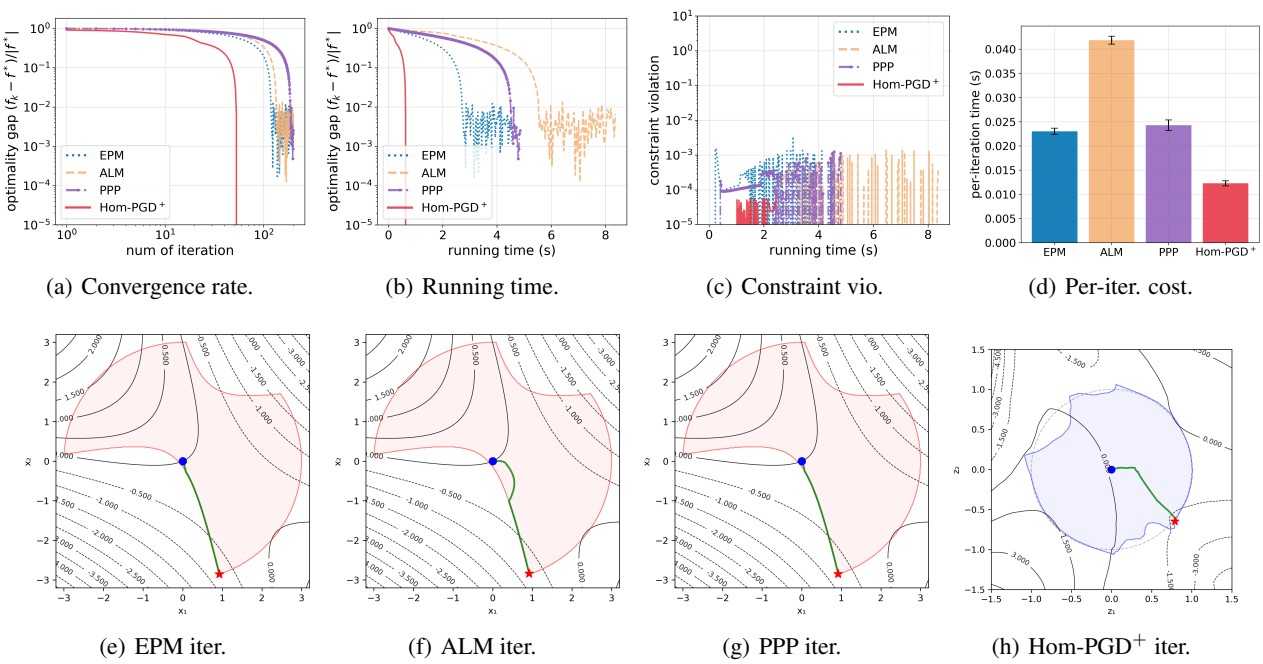

*Figure 10.* Illustrative examples of Hom-PGD$^+$ for solving QCQP with non-convex BH constraints.

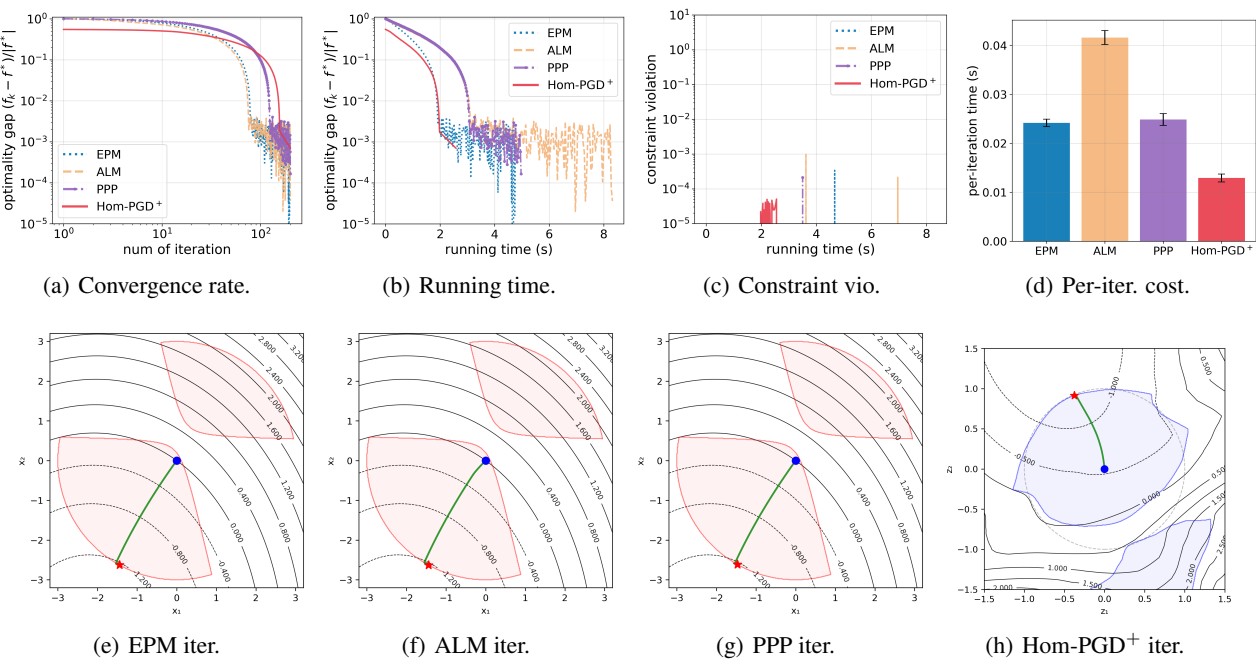

*Figure 11.* Illustrative examples of Hom-PGD$^+$ for solving QCQP with non-BH constraints.

