# OpenReview forum: "Hom-PGD$^+$: Fast Reparameterized Optimization over Non-convex Ball-Homeomorphic Set"
_ICML.cc/2026/Conference — ICML 2026 regular_

### Official Review · Reviewer_jyAm · 2026-03-04

**Soundness:** 3
**Presentation:** 3
**Significance:** 3
**Originality:** 3
**Overall Recommendation:** 4
**Confidence:** 3

**Summary:**

Based on the provided strengths and weaknesses, the paper's core contribution is the novel integration of Invertible Neural Networks (INNs) to learn a homeomorphic mapping from a non-convex, ball-homeomorphic constraint set to a unit ball. This transformation allows the problem to be solved with efficient projection onto the ball via a bisection method (Hom-PGD+ algorithm). The theoretical analysis shows a promising $O(\varepsilon^{-2})$ convergence rate, accounting for INN approximation error, and experiments demonstrate its effectiveness on synthetic (non-convex QCQP) and real-world (chance-constrained power grid optimization) problems.

**Compliance With Llm Reviewing Policy:**

Affirmed.

**Final Justification:**

The rebuttal addresses my concerns, so I will keep the score of 4.

**Key Questions For Authors:**

1.  **Practical Verification of Assumption 2:** How can the strong requirements on uniform Jacobian and function approximation errors be measured or guaranteed in practice, given that the true homeomorphism $\psi$ is unknown?

2.  **Novelty Distinction:** Beyond integrating prior ideas of homeomorphic projection and reparameterization, what is the fundamental new contribution? Is it primarily the convergence analysis for an iterative optimizer using an INN?

3.  **Experimental Breadth and Scaling:** Could the evaluation be strengthened with tests on more machine-learning-relevant constrained problems and more comprehensive time-to-solution scaling analyses against a wider range of solvers?

**Limitations:**

yes

**Strengths And Weaknesses:**

## Strengths:

1.  **Novelty:** The approach of learning a homeomorphic reparameterization via an INN is creative and new.
2.  **Theory:** Provides a solid convergence analysis, achieving an $O(\varepsilon^{-2})$ iteration complexity that is competitive with state-of-the-art methods for non-convex constraints, while carefully handling INN approximation error.
3.  **Experiments:** Validates the method effectively on synthetic (QCQP) and practical (chance-constrained power grid) problems, showing significant speedups over several first-order baselines.

## Key Weaknesses and Reviewer Concerns:

1.  **Assumption 2 is Deemed Overly Strong and Practically Unverifiable:**
    *   This is the most critical concern. Reviewers find the requirement for uniform bounds on both function approximation error ($\|\Psi - \Phi\| \le \varepsilon_{inn}$) and Jacobian approximation error ($\|J_{\Psi} - J_{\Phi}\| \le \varepsilon_{inn}$) to be exceptionally strong.
    *   The core issue is practical verification: How can one guarantee or even measure these bounds for a learned INN on a complex, unknown constraint set $K_{\theta}$? The true homeomorphism $\psi$ is unknown, making direct computation of $\|\Psi - \Phi\|$ and $\|J_{\Psi} - J_{\Phi}\|$ impossible. The training loss (Eq. 1) does not explicitly or provably enforce these specific error bounds.

2.  **Novelty is Perceived as Incremental by Some:**
    *   The work can be viewed as a straightforward combination of two existing ideas: (a) using INNs for feasibility projection (Liang et al., 2023, 2024) and (b) reparameterization for optimization (e.g., gauge mapping for convex sets).
    *   They request a sharper technical differentiation to justify the claim of a "substantive framework advance." The rebuttal needs to clearly articulate what is fundamentally new beyond this integration—is it the convergence analysis for an iterative optimizer (vs. single-step projection), the handling of parametric $\theta$, or the extension to a broader class of non-convex sets?

3.  **Experimental Evaluation is Seen as Limited:**
    *   **Breadth:** Experiments are confined to QCQP and a specific power grid application. Reviewers desire tests on a broader set of ML-relevant tasks (e.g., training with non-convex constraints, adversarial robustness with non-uniform perturbations as mentioned in the intro).
    *   **Comparisons:** While the rebuttal added comparisons with IPOPT, reviewers may seek more comprehensive benchmarks against a wider range of industrial-grade nonlinear solvers (e.g., BARON for global optimality gaps) and specialized methods for chance-constrained programming.
    *   **Scaling Evidence:** More rigorous scaling studies (time-to-solution vs. problem dimension, number of constraints) are needed to conclusively demonstrate the "order of magnitude" speedup claim.

---

> ### Author Rebuttal · Authors · 2026-03-31
>
> Dear Reviewer jyAm,
>
> We sincerely thank the reviewer for the time and effort invested in evaluating our paper. We respond to your questions one by one below.
>
> ---
> *C1 & Q1. Assumption 2 is Deemed Overly Strong and Practically Unverifiable...*
>
> ---
>
> **Response:**
> We thank the reviewer for this thoughtful comment. We clarify the reasonability of Assumption 2 with both theoretical and practical justifications.
>
>  **Clarification for the reasonability of Assumption 2.**
> - **Theoretical Justification:** The *universal approximation property of INNs* under Sobolev norm ensures that with sufficient depth and width, INNs can approximate arbitrary smooth diffeomorphisms under Sobolev norm, implying both function and Jacobian errors can be made arbitrarily small.
> - **Practical Justification:**
>    - *Function Approximation Error:* Our training loss (Eq. 1) encourages $ \Phi(\mathcal{B}) \subseteq \mathcal{K} $ and maximal coverage. Appendix G.1 confirms trained INNs achieve low Hausdorff distances.
>    - *Jacobian Approximation Error:* Direct verification is challenging; however, Appendix E.1 shows that when $d_H(\Phi(\mathcal{B}), \mathcal{K})$ is small, optimal transport theory guarantees existence of a target homeomorphism with controlled Jacobian deviation via Monge-Ampère regularity theory.
>
> **Practical Verification:** Function error can be verified via Hausdorff distance sampling—Figure 7 shows distances below 0.3. While direct Jacobian verification is challenging, Appendix E.1 provides theoretical support. **The resulting bounds may be loose, but experiments demonstrate optimality gaps as small as 2.9% (Line 417).**
>
> **Contribution Despite Restrictiveness.** Even though Assumption 2 is relatively restrictive, our analysis represents a meaningful advancement: prior work (Liu+ 25) assumes exact homeomorphisms for convex constraints, whereas we explicitly incorporate INN approximation errors and quantify their impact on feasibility and optimality—providing the first convergence guarantees for learned homeomorphism-based optimization over non-convex constraints.
>
> ---
> *C2 & Q2. Novelty is Perceived as Incremental...*
>
> ---
>
> **Response:**
> We thank the reviewer for highlighting this point. While our work builds upon (liang+ 23,24; liu+ 25)), our contributions are non-trivial extensions addressing fundamentally different challenges.
>
> **Algorithmic Contributions:** Prior works (liang 23,24) use INN-based projection layers as post-processing for neural network predictions. In contrast, we integrate INN-learned homeomorphisms into a *complete iterative optimization framework*. This introduces challenges including error accumulation and ensuring stability despite inexact projections. (liu+ 25) proposed Hom-PGD using *exact* homeomorphisms for convex sets; our framework extends to non-convex constraints using *learned* homeomorphisms—a fundamentally different setting.
>
> **Theoretical Contributions:** We establish $O(\epsilon^{-2})$ convergence for finding $\epsilon$-stationary points, improving upon existing $O(\epsilon^{-3})$ or $O(\epsilon^{-4})$ rates. Our analysis explicitly accounts for INN approximation errors throughout optimization, providing bounds relating approximation quality to solution accuracy (Theorem 1), **which advance the state-of-the-art reparameterization optimization methods.**
>
> In summary, the extension to iterative optimization over non-convex constraints with learned homeomorphisms and the accompanying theoretical guarantees represent substantial contributions beyond prior literature.
>
> ---
> *C3 & Q3. Breadth of Experiments...Comparisons with More Baselines...Scaling Evidence...*
>
> ---
> **Response:** We thank the reviewer for the constructive suggestions. We address these concerns one by one in the following.
>
> - Breadth of Experiments. We agree that demonstrating applicability across diverse problem classes strengthens the contribution. We include non-uniform adversarial attacks on neural networks, which involves optimization over $\ell_p$ ($p<1$) norm constraints. This additional experiment demonstrates the versatility of our methodology across fundamentally different application domains.
> - Comparisons with industrial solvers. BARON was considered, but lacks free public/academic access. Popular industrial solvers such IPOPT, GUROBI have been compared in our original version.
> - Specialized methods for chance-constrained optimization. We add two baselines, scenario-based and CVaR-based approximations—solved via MOSEK (academic license).
> - More Scaling Evidence. We extend our scaling analysis with larger QCQP benchmarks.
>
> We will incorporate full comparison results into the revised manuscript, including additional baselines and detailed discussion of scalability. All the additional experiments are provided in: [https://anonymous.4open.science/r/ICML_Rebuttal-B1D4/Rebuttal.pdf](https://anonymous.4open.science/r/ICML_Rebuttal-B1D4/Rebuttal.pdf)

---

> > ### Author Rebuttal · Reviewer_jyAm · 2026-04-02
> >
> > Thank you for addressing my questions and for including additional experiments. I will keep the score of 4.

---

> > > ### Author Response · Authors · 2026-04-05
> > >
> > > We sincerely thank reviewer jyAm for confirming that the concerns have been fully resolved, and we are deeply grateful for the reviewer maintaining the positive score. We genuinely appreciate the thoughtful and constructive questions raised throughout this review process, which have significantly helped us improve the clarity and rigor of our work. We will integrate the discussions and insights from this rebuttal into the final version to make our contributions more transparent and accessible to readers. Thank you once again for your valuable time and engagement with our work.

---

### Official Review · Reviewer_wB1K · 2026-03-06

**Soundness:** 2
**Presentation:** 2
**Significance:** 2
**Originality:** 2
**Overall Recommendation:** 3
**Confidence:** 4

**Summary:**

This paper studied optimization over non-convex constraint sets that are homeomorphic to a ball, and proposed a learning-based and projection efficient first-order method (i.e., Hom-PGD+) that efficiently solves such problems without requiring expensive projection or optimization oracles. It provided convergence analysis of the proposed method. It also provided some numerical experiments to demonstrate effectiveness of the proposed method.

**Compliance With Llm Reviewing Policy:**

Affirmed.

**Final Justification:**

I still think that the assumptions given in this paper are somewhat strict, especially assumption 2.

So I keep my score.

**Key Questions For Authors:**

Comments:

1)	The authors should prove approximation errors of INN instead of directly using Assumption 2.

2)	The authors should add  machine learning applications such as non-uniform adversarial attacks in neural networks.

3)	Since this paper focuses on optimization with non-convex constraint sets, the authors should give some experiments in machine learning applications.

**Limitations:**

Yes

**Strengths And Weaknesses:**

Strengths:

1) This paper presented a learning-based and projection efficient first-order method (i.e., Hom-PGD+) that efficiently solves such problems without requiring expensive projection or optimization oracles.

2) It studied the convergence properties of the proposed method for finding an approximate stationary solution.

Weaknesses:

1) In the convergence analysis, some assumptions are very strict. E.g., Assumption 1 requires the objective function is Lipschitz continuous, and  Assumption 2 directly give approximation errors of INN. The authors should prove approximation errors of INN.

2) In the numerical experiments, there do not exist any machine learning applications. The authors should add  machine learning applications such as non-uniform adversarial attacks in neural networks.

---

> ### Author Rebuttal · Authors · 2026-03-31
>
> Dear Reviewer wB1K,
>
> We sincerely thank the reviewer for the time and effort invested in evaluating our paper. We respond to your questions one by one below.
>
> ---
> *C1 & Q1. In the convergence analysis, some assumptions are very strict. E.g., Assumption 1 requires the objective function is Lipschitz continuous, and Assumption 2 directly give approximation errors of INN. The authors should prove approximation errors of INN.*
>
> ---
>
> **Response:** We thank the reviewer for this careful examination of our theoretical assumptions. We address each concern individually below.
>
> **1. Lipschitz Continuity of the Objective Function (Assumption 1)**
>
> We acknowledge that requiring global Lipschitz continuity may appear restrictive at first glance. However, as clarified in the Remark in Appendix C.2 (Line 1233), this assumption can be relaxed to Lipschitz continuity over a compact set. Since our optimization is performed over the bounded domain $\mathcal{B}(0,1)$ in the latent space, which maps to the compact constraint set $\mathcal{K}$ via the INN, we only require the objective function to be Lipschitz continuous over this compact region. This relaxed condition is satisfied by a broad class of objective functions encountered in practice, including any smooth continuous function. Therefore, we believe this assumption is quite mild and does not significantly limit the applicability of our theoretical results.
>
> **2. Approximation Errors of INN (Assumption 2)**
>
> We first acknowledge that providing rigorous bounds on INN approximation errors is challenging. While a complete formal proof for the specific approximation error bounds remains an open problem in the deep learning theory literature, the plausibility of achieving small approximation errors is well-supported by the *universal approximation property of invertible neural networks*. As discussed in Section 3.1, recent theoretical work (Jin+,2024; Ishikawa+,2022; Lyu+, 2022) has established that INNs with sufficient depth and width can approximate arbitrary (smooth) diffeomorphisms to arbitrary precision under the Sobolev norm, which means the function approximation error and Jacobian approximation error can be very small. This universality result provides the theoretical foundation for Assumption 2, suggesting that with appropriate architecture design and training, the approximation errors \(\epsilon_{\text{inn}}\) can be made arbitrarily small.
>
> In the following, we clarify the reasonability to directly use Assumption 2 which includes function approximation error bound and Jacobian approximation error bound.
>
> - **Function approximation error.** Our INN training framework is specifically designed to achieve this through the loss function in Equation (1), which simultaneously encourages: (a) $\Phi(\mathcal{B}) \subseteq \mathcal{K}$ via the constraint penalty term, and (b) maximal coverage of $\Phi(\mathcal{B})$ within $\mathcal{K}$ via the volume maximization term. Our empirical results in Appendix G.1 further corroborate this, demonstrating that trained INNs achieve low Hausdorff distances and accurate geometric approximations in practice.
> - **Jacobian approximation error.** We acknowledge that directly verifying the Jacobian error is challenging due to the lack of ground-truth. However, Appendix E.1 provides theoretical justification: when $d_H(\Phi(\mathcal{B}), \mathcal{K})$ is small, we construct $\psi = \mathbf{T} \circ \Phi$ via optimal transport, where $\|\mathbf{J}_\mathbf{T} - \mathbf{I}\|$ can be very small by the Monge-Ampère equation (Eq. 27–28). This explanation makes Assumption 2 reasonable in some sense.
>
> **It should be important to note that these bounds are used in the worst-case analysis. The resulting bounds may be loose, but experiments demonstrate optimality gaps as small as 2.9% (Line 417).**
>
> ---
> *C2 & Q2 & Q3. In the numerical experiments, there do not exist any machine learning applications. The authors should add machine learning applications such as non-uniform adversarial attacks in neural networks.*
>
> ---
>
> **Response:** We thank the reviewer for this constructive suggestion regarding the breadth of our experimental evaluation. We agree that demonstrating applicability across diverse problem classes strengthens the contribution, and we will expand our experiments accordingly in the revised manuscript.
>
> **Additional Experiment: Non-uniform Adversarial Attacks on Neural Networks**
> We conducted experiments on non-uniform adversarial attacks with non-convex $\ell_p$-norm ($p < 1$) perturbations. Comparing with the radial feasibility scaling baseline, our method performs substantially better in non-convex regions (e.g., $\ell_{1/2}$-norm) due to reparameterization to unit ball space with favorable geometry.
>
> Complete results including tables, setup descriptions, and visualizations: [https://anonymous.4open.science/r/ICML_Rebuttal-B1D4/Rebuttal.pdf](https://anonymous.4open.science/r/ICML_Rebuttal-B1D4/Rebuttal.pdf)

---

> > ### Author Rebuttal · Reviewer_wB1K · 2026-04-01
> >
> > I still think that the assumptions given in this paper are somewhat strict, especially assumption 2.
> >
> > The authors should prove approximation errors of INN instead of directly using Assumption 2, e.g., consider some specific structural problems or models.
> >
> > **Note that** the authors have provided an additional two pages of anonymous PDF in rebuttal, it likely violated the "5000-character limit" rule.
> >
> > **This additional two pages of anonymous PDF (https://anonymous.4open.science/r/ICML_Rebuttal-B1D4/Rebuttal.pdf) is unfair to the authors of other articles.**

---

> > > ### Author Response · Authors · 2026-04-02
> > >
> > > We thank the reviewer for taking the time to carefully read our reply and for their continued valuable engagement with our work. The insightful feedback has helped us gain a deeper understanding of the concerns raised, and we believe the following clarifications will fully address the reviewer's remaining questions.
> > >
> > > ---
> > >
> > > **Regarding A1 (Lipschitz Continuity):** As mentioned in our previous response, this assumption is **not** restrictive. Since our optimization is performed over the bounded ball in the transformed space, which is mapped from the original compact constraint set $\mathcal{K}$ via the trained INN, we only require the original objective function to be Lipschitz continuous over this compact region (**see Remark in Appendix C.2, Line 1233**). This relaxed condition is satisfied by a broad class of objective functions encountered in practice, including any smooth function.
> > >
> > > **Regarding A2 (INN Errors):** We appreciate the reviewer's careful scrutiny of this assumption. We acknowledge that A2 is relatively stricter. However, we believe its reasonableness can be understood from the following aspects:
> > >
> > > 1. **Existence of INNs under A2**:  First, there exists a smooth homeomorphism $ f $ that can transform a unit ball to the target constraint $\mathcal{K}$ in our setting (e.g., contractible manifold with simply connected boundary; see footnote on Page 3). Such a target homeomorphism belongs to the *Sobolev space* ( $ \{f: \\|f\\|_{K,r,p} < \infty\} $ ).
> > >     - Besides,  literature [1] has shown that coupling-layer INNs (used in our work) can approximate the target homeomorphism to arbitrary precision under the Sobolev norm with sufficient parameters.
> > >     - Therefore, by the definition of the Sobolev norm  (  $ \\|f\\|_{K, r, p} := \sum _{|\alpha| \leq r} ( \int_K \| \partial^\alpha f(x) \|^p dx )^{1/p} $ ), *the INN can approximate both the target homeomorphism and its Jacobian simultaneously* (by taking $r=1$) and the **existence** of such an INN under the A2 is well-justified.
> > >
> > > 2. **Training INN to achieve small errors**: We acknowledge the reviewer's concern that a complete theoretical proof would strengthen our work. However, proving the learning error from an optimization perspective (i.e., showing that gradient-based training provably finds such an approximating INN) remains a fundamental open problem in deep learning theory. This question falls within the field of neural network optimization and training dynamics, which are orthogonal to the main contributions of our work. We will explicitly acknowledge this limitation in the revised manuscript and leave a rigorous treatment to future work.
> > >
> > > 3. **Practical and other theoretical justifications**: As detailed in our previous response, our INN training framework (Eq. 1) is specifically designed to achieve small approximation errors through constraint penalties and volume maximization. Our empirical results (Appendix G.1) also demonstrate that trained INNs achieve low Hausdorff distances in practice. Additionally, Appendix E.1 provides theoretical justification for the Jacobian error bound via optimal transport arguments.
> > >
> > > **Contributions beyond A2:** Even though A2 is relatively restrictive, our analysis represents a meaningful contribution: prior work [2] needs exact homeomorphisms for convex constraints, whereas we explicitly incorporate INN approximation for non-convex set and quantify their impact on feasibility and optimality—providing the **first** convergence guarantees for learned homeomorphism-based optimization over non-convex constraints.
> > >
> > > We are grateful for the reviewer's constructive feedback, and we will incorporate these discussions into the final version to make the assumptions and their justifications more transparent and accessible.
> > >
> > >
> > >
> > > - [1] Ishikawa I, et al. Universal Approximation Property of Invertible Neural Networks. JMLR 2023.
> > > - [2] Liu C, et al. Fast Projection-Free Approach (without Optimization Oracle) for Optimization over Compact Convex Set. NeurIPS 2025.
> > >
> > >
> > > ---
> > >
> > > **Response to 5000-character limit concern**:
> > >
> > > We respectfully clarify that our anonymous PDF link complies with the ICML 2026 guidelines. According to the [ICML 2026 Peer Review FAQ]: *"While links are allowed, reviewers are not required to follow them, and links should be used primarily for figures..."*
> > >
> > > Our main response has been posted in the 5000-character rebuttal box above. The linked PDF contains only additional experimental results and visualizations with basic descriptions to support our response, as requested by the reviewers.
> > >
> > > We respectfully request that reviewers evaluate our rebuttal primarily based on the main responses, with the linked PDF serving as optional supplementary experimental results, as permitted by the ICML guidelines.
> > >
> > > ---
> > >
> > > Once again, we appreciate the reviewer's time and effort in evaluating our work and are grateful for the constructive feedback that has helped improve it.

---

### Official Review · Reviewer_pRok · 2026-03-15

**Soundness:** 3
**Presentation:** 3
**Significance:** 3
**Originality:** 3
**Overall Recommendation:** 5
**Confidence:** 3

**Summary:**

This paper proposes Hom-PGD+, a reparameterized optimization method that learns a homeomorphic mapping between the constraint set and a unit ball using an invertible neural network. The paper provides convergence analysis and demonstrates the performance of the proposed method in experimental evaluations.

**Compliance With Llm Reviewing Policy:**

Affirmed.

**Key Questions For Authors:**

The proposed approach transforms the constraint set into a ball constraint. Could the authors clarify the advantages of this transformation compared with optimization over other manifolds or constraint geometries?

What does the symbol “+” mean in the name of the proposed method, Hom-PGD+? At first, I thought that this algorithm might be an extension of Hom-PGD, which makes the naming somewhat confusing. (This is not be key question)

**Limitations:**

yes

**Strengths And Weaknesses:**

Soundness
The idea of the proposed method is technically sound for optimization problems over ball-homeomorphic sets. In addition, the authors provides the theoretical and runtime complexity analyses.

Presentation
The positioning with respect to the comparison methods is clear, and the paper is well-structured.

Significance
The effectiveness of the proposed method is sufficiently explained through not only the algorithmic proposal but also theoretical analysis, runtime complexity analysis, and discussions on non-BH problems.

Originality
This submission provides a reparameterized optimization method, which shows strength in originality.

---

> ### Author Rebuttal · Authors · 2026-03-31
>
> Dear Reviewer pRok,
>
> We sincerely thank the reviewer for the time and effort invested in evaluating our paper. We thank the reviewer for the positive assessment of our work, particularly for recognizing the technical soundness of our approach, the significance of our theoretical and complexity analyses. We respond to your questions one by one below.
>
> ---
>
> *Question 1. The proposed approach transforms the constraint set into a ball constraint. Could the authors clarify the advantages of this transformation compared with optimization over other manifolds or constraint geometries?*
>
> ---
>
> **Response:**
> We thank the reviewer for this insightful question regarding our choice of the unit ball as the target geometry. The transformation to a ball constraint offers several significant advantages compared to optimization over other manifolds or constraint geometries.
>
> **1. Simplicity and Efficiency of Projection**
>
> The unit ball $\mathcal{B}(0,1)$ admits the simplest possible projection operator among all bounded convex sets. The projection is given by the closed-form expression $\text{Proj}_{\mathcal{B}}(\mathbf{z}) = \mathbf{z} / \max(1, \|\mathbf{z}\|_2)$, requiring only $O(n)$ computational cost. In contrast, projection onto polytopes necessitates solving linear or quadratic programs with substantially higher complexity. For general manifolds, projection involves solving nonlinear optimization subproblems that may require iterative procedures without closed-form solutions. Furthermore, if one attempts to employ manifold-based gradient descent methods (e.g., Riemannian optimization) over general constraint geometries, each iteration incurs additional costs related to computing geodesics and exponential/logarithmic maps, which can be prohibitively expensive for high-dimensional problems. The ball geometry circumvents all these complications, enabling our framework to achieve low per-iteration complexity while maintaining theoretical convergence guarantees.
>
> **2. Compatibility with Gradient-Based Optimization**
>
> The ball constraint is particularly well-suited for projected gradient descent methods. The smooth and isotropic geometry of the ball ensures that gradient directions are minimally distorted after projection, preserving descent properties. Furthermore, the ball has uniform curvature in all directions, which avoids the ill-conditioning issues that arise with highly anisotropic constraint sets such as elongated ellipsoids or polytopes with disparate edge lengths. This geometric regularity contributes to stable convergence behavior as established in our theoretical analysis (Theorem 1).
>
> **3. Universality of the Homeomorphism**
>
> From a topological perspective, informally speaking (where the formal statement can be found in the footnote in Page 3), any compact, simply connected constraint set $\mathcal{K} \subset \mathbb{R}^n$ with smooth boundary is homeomorphic to the unit ball. This universality ensures that our approach is applicable to a broad class of constraint sets without requiring problem-specific target geometries. Choosing alternative manifolds would restrict applicability or require additional structural assumptions on $\mathcal{K}$.
>
> In summary, the ball constraint represents an optimal balance between computational tractability, theoretical analyzability, and broad applicability, making it the natural choice for our framework.
>
> ---
>
> *Question 2. What does the symbol "+" mean in the name of the proposed method, Hom-PGD+? At first, I thought that this algorithm might be an extension of Hom-PGD, which makes the naming somewhat confusing. (This is not be key question)*
>
> ---
>
> **Response:**
>
> We thank the reviewer for this clarification question. We agree that explicitly referencing the Hom-PGD naming convention from (liu+ 25) would provide clearer context for readers. The "+" in Hom-PGD+ signifies the extension from convex constraints to the more challenging *non-convex* setting, while the essential technique of homeomorphism-based reparameterization remains the core algorithmic principle. Specifically, (liu+ 25) constructs exact analytical homeomorphisms for convex sets, whereas our Hom-PGD+ learns approximate homeomorphisms via INNs to handle non-convex, ball-homeomorphic constraint sets. In the revised manuscript, we will introduce the method name Hom-PGD from (liu+ 25) when discussing related work and add a dedicated remark upon first introduction of the algorithm name to make this distinction immediately clear to readers.
>
> Reference:
> Liu, C., Liang, E., and Chen, M. Fast projection-free approach (without optimization oracle) for optimization over compact convex set. Advances in Neural Information Processing Systems, 2025a.

---

> > ### Author Rebuttal · Reviewer_pRok · 2026-04-03
> >
> > Thank you for your detailed response. I will maintain my current score.

---

> > > ### Author Response · Authors · 2026-04-05
> > >
> > > We sincerely thank reviewer pRok for confirming that the concerns have been fully resolved, and we are deeply grateful for the reviewer maintaining the positive score. We genuinely appreciate the thoughtful and constructive questions raised throughout this review process, which have significantly helped us improve the clarity and rigor of our work. We will integrate the discussions and insights from this rebuttal into the final version to make our contributions more transparent and accessible to readers. Thank you once again for your valuable time and engagement with our work.

---

### Official Review · Reviewer_m8HU · 2026-03-24

**Soundness:** 3
**Presentation:** 3
**Significance:** 2
**Originality:** 2
**Overall Recommendation:** 3
**Confidence:** 4

**Summary:**

This paper addresses optimization problems with non-convex constraints that are homeomorphic to a ball. The proposed methodology, named Hom-PGD+, relies on learning the homeomorphism between non-convex sets and the unit ball through invertible neural networks (INNs) and then applying projected gradient descent (PGD). An analysis that shows convergence to an approximate stationary point, as well as a complexity analysis are provided. Finally, the methodology is evaluated on QCQP and chance-constrained power grid optimization problems, showing benefits against standard non-convex optimization methods.

**Compliance With Llm Reviewing Policy:**

Affirmed.

**Final Justification:**

My initial review expressed four main concerns (novelty, restrictiveness of theoretical analysis, usefulness of complexity analysis and lack of experiments).  The authors properly addressed the latter two, but my concerns on the moderate novelty of this submission compared to prior works and on the restrictiveness and inability to practically justify a fundamental assumption on the theoretical analysis have remained. That said, I have raised my score from 2 to 3, but due to the previous concerns, I would not be able to recommend this paper for acceptance.

**Key Questions For Authors:**

1. As a follow-up to weakness 1, would it be possible that the authors better clarify the novel aspects of their work compared to [1]-[3]? For example, it would have been helpful if the method in [3] was referred to with its actual name in the paper (Hom-PGD), so that the reader immediately understands the context behind the naming choice Hom-PGD+, then the fundamental differences were discussed and so on.
2. As a follow-up to weakness 2, would there be any way to further justify the two conditions in Assumption in practice? I believe that this would be beneficial for arguing about how the theoretical analysis is well-connected to practical performance.
3. As a follow-up to weakness 4, it is possible to illustrate the advantages of the method in other problem classes as discussed in the introduction, to better highlight its applicability and perhaps relevance to the broader ML community?

**Limitations:**

Yes.

**Strengths And Weaknesses:**

**Strengths**

1. The topic being addressed is of interest, since non-convex optimization can be arduous, so constructing methods that map the original non-convex problem to a new space where it can be solved more efficiently is a research area of high importance.
2. The core methodology is technically sound and accompanied with convergence guarantees to approximate stationary points, as well as a complexity analysis.
3. The provided experiments on QCQPs and power grid problems demonstrate benefits in solution speed and scalability against standard baselines such as EPM, ALM, PPP and IPOPT.
4. A substantial amount of mathematical, methodology and implementation details are provided in both the main paper and the appendix which is appreciated.

**Weaknesses**

1. **Novelty compared to prior work.** The contributions of this work appear a bit incremental compared to [1], [2], [3]. In particular, [1] and [2] propose learning constraint reparameterizations via INNs, focusing on single-step projections. In the meantime, [3] proposed Hom-PGD which constructs a homeomorphism between convex constraint sets and a unit ball and then applies PGD. The methodology proposed in the current paper, Hom-PGD+, can be viewed as extending Hom-PGD [3] using the learning constraint reparameterizations via INNs ideas presented in [1], [2].

2. **Restrictiveness of Assumption 2 and convergence analysis.** Although I understand that the two conditions (bounded approximation error and bounded Jacobian approximation error) are essential for proving the convergence of the method (Theorem 1), these assumptions appear quite restrictive; especially the Jacobian one. It would be very helpful if there were some practical verifications of these assumptions to further justify them.

3. **Usefulness of complexity analysis.** The complexity analysis in Section 4.3 is not particularly useful as the claimed O(n^2) complexity simply relies on an INN design choice. In particular, the complexity turns out to be O(W I) where W is the number of INN parameters and I is the number of iterations. Going from that to O(I n^2) relies on the fact that the authors claim that selecting an O(n) INN width is “good enough”, which leads to the claimed advantage against O(n^3) methods. Nevertheless, as this result relies on the ad-hoc selection of W = O(n), it is hard to justify its usefulness.

4. **Limited problem classes in experiments.** Although in the beginning of the paper, the authors argue about the substantial applicability of the studied problem setup, the experiments are only conducted on QCQPs and power grid optimization problems. It would be very helpful if additional problem classes were considered to further support the applicability of the proposed methodology.

**Minor comments/typos:**

- Line 355: “we scaling”
- Line 356: “with industrial solver”
- In title of section 5.2, “vs” appears in math mode.
- In title of section G.2 in Appendix: “Abalation study”



[1] Liang, E., Chen, M., and Low, S. H. Low complexity homeomorphic projection to ensure neural-network solution feasibility for optimization over (non-)convex set. In International Conference on Machine Learning. PMLR, 2023.

[2] Liang, E., Chen, M., and Low, S. H. Homeomorphic projection to ensure neural-network solution feasibility for constrained optimization. Journal of Machine Learning Research, 2024.

[3] Liu, C., Liang, E., and Chen, M. Fast projection-free approach (without optimization oracle) for optimization over compact convex set. Advances in Neural Information Processing Systems, 2025a.

---

> ### Author Rebuttal · Authors · 2026-03-31
>
> Dear Reviewer m8HU,
>
> We sincerely thank the reviewer for the time and effort invested in evaluating our paper. We address your concerns below.
>
> ---
>
> *C1 & Q1: Novelty Compared to Prior Work and Naming Clarification*
>
> ---
>
> **Response:**
> We thank the reviewer for highlighting this point. While our work builds upon [1], [2], and [3], our contributions are non-trivial extensions addressing fundamentally different challenges.
>
> **Algorithmic Contributions:** Prior works [1, 2] use INN-based projection layers as post-processing for neural network predictions. In contrast, we integrate INN-learned homeomorphisms into a *complete iterative optimization framework*. This introduces challenges including error accumulation and ensuring stability despite inexact projections. [3] proposed Hom-PGD using *exact* homeomorphisms for convex sets; our framework extends to non-convex constraints using *learned* homeomorphisms—a fundamentally different setting.
>
> **Theoretical Contributions:** We establish $O(\epsilon^{-2})$ convergence for finding \(\epsilon\)-stationary points, improving upon existing $O(\epsilon^{-3})$ or $O(\epsilon^{-4})$ rates. Our analysis explicitly accounts for INN approximation errors throughout optimization, providing bounds relating approximation quality to solution accuracy (Theorem 1), **which advance the state-of-the-art reparameterization optimization methods.**
>
> **Naming Clarification:** We will introduce Hom-PGD from [3] in related work and clarify "+" signifies extension from convex to *non-convex* constraints. We will add a dedicated remark upon first introduction.
>
> ---
>
> *C2 & Q2: Restrictiveness of Assumption 2 and Practical Justification*
>
> ---
>
> **Response:**
> We thank the reviewer for this thoughtful comment. We provide both theoretical and practical justifications.
>
> **Theoretical Justification:** The *universal approximation property of INNs* under Sobolev norm (Ishikawa+ 24; Jin+ 24) ensures that with sufficient depth and width, INNs can approximate arbitrary smooth diffeomorphisms under Sobolev norm, implying both function and Jacobian errors can be made arbitrarily small.
>
> **Practical Justification:**
> - *Function Approximation Error:* Our training loss (Eq. 1) encourages $\Phi(\mathcal{B}) \subseteq \mathcal{K}$ and maximal coverage. Appendix G.1 confirms trained INNs achieve low Hausdorff distances.
> - *Jacobian Approximation Error:* Direct verification is challenging; however, Appendix E.1 shows that when $d_H(\Phi(\mathcal{B}), \mathcal{K})$ is small, optimal transport theory guarantees the existence of a target homeomorphism with controlled Jacobian deviation via Monge-Ampère regularity theory.
>
> **Practical Verification:** Function error can be verified via Hausdorff distance sampling—Figure 7 shows distances below 0.3. While direct Jacobian verification is challenging, Appendix E.1 provides theoretical support. **The resulting bounds may be loose, but experiments demonstrate optimality gaps as small as 2.9% (Line 417).**
>
> Despite restrictiveness, our analysis provides the first convergence guarantees for learned homeomorphism-based optimization over non-convex constraints—**advancing beyond [3]'s exact homeomorphism assumption. Relaxing these conditions is an interesting future direction.**
>
> ---
>
> *C3: Usefulness of Complexity Analysis*
>
> ---
>
> **Response:** We thank the reviewer for this point.
>
> **Necessity:** Complexity analysis is essential for understanding scalability, even though the bound is a worst-case analysis. Without it, practitioners lack guidance on suitability for large-scale applications.
>
> **Objectivity of width $O(n)$:** This is *empirically justified*—our experiments (Section 5, Appendix G) demonstrate INNs with $O(n)$ achieve strong approximation across diverse constraints, evidenced by low Hausdorff distances and high feasibility rates. The fact that $O(n)$ yields both good performance and favorable complexity demonstrates computational attractiveness.
>
>
>
> ---
>
> *C4 & Q3: Limited Problem Classes and Broader ML Applicability*
>
> ---
>
> **Response:**
> We thank the reviewer for this constructive suggestion.
>
> **Additional Experiment:** We conducted experiments on non-uniform adversarial attacks with non-convex \(\ell_p\)-norm (\(p < 1\)) perturbations. Comparing with the radial feasibility scaling baseline, our method performs substantially better in non-convex regions (e.g., $\ell_{1/2}$-norm) due to reparameterization to unit ball space with favorable geometry.
>
> **Key Advantages for ML Community:**
> 1. *Amortized Cost:* INN training is one-time; subsequent problems benefit from efficient per-iteration complexity.
> 2. *Fast Convergence:* $O(\epsilon^{-2})$ improves upon $O(\epsilon^{-3})$ or $O(\epsilon^{-4})$ rates, yielding significant wall-clock savings.
>
> Complete results including tables, setup descriptions, and visualizations: [https://anonymous.4open.science/r/ICML_Rebuttal-B1D4/Rebuttal.pdf](https://anonymous.4open.science/r/ICML_Rebuttal-B1D4/Rebuttal.pdf)

---

> > ### Author Rebuttal · Reviewer_m8HU · 2026-04-04
> >
> > I thank the authors for the detailed response.
> >
> > **C1 & Q1: Novelty Compared to Prior Work and Naming Clarification**
> >
> > I agree that there is a significant difference between [1], [2], which present INN-based approaches for single-step feasibility recovery under non-convex constraints and the current submission which incorporates learned homeomorphisms into an iterative optimization framework.
> >
> > That said, I still view this submission as a natural extension of [3] going from exact homeomorphisms and convex constraints in Hom-PGD to learned homeomorphisms and non-convex constraints using similar ideas as in [1],[2].
> > Overall, I recognize that the paper addresses additional challenges following from this combination, such as error accumulation and ensuring stability, but I still consider the level of novelty to be moderate given the prior work [1-3].
> >
> > **C2 & Q2: Restrictiveness of Assumption 2 and Practical Justification**
> >
> > Thank you for the response and I definitely understand all your points. But I believe it’s still fair to say that the reliance of the theoretical analysis on an assumption that is unverifiable is problematic.
> >
> > **C3: Usefulness of Complexity Analysis**
> >
> > I agree that the empirical justification is very valuable. My main point was that a complexity cannot be formally established by observing that “using this layer width works well”.
> >
> > But I would say that in practice the empirical justification, such as the one in Fig. 4, is perhaps even more important in the end, so my concern is covered here.
> >
> > **C4 & Q3: Limited Problem Classes and Broader ML Applicability**
> >
> > Thank you very much for the newly added experiments, which I think strengthen the argument regarding the applicability of the method.
> >
> > **Conclusion**
> >
> > The authors provided detailed responses which I appreciate. Two out of four concerns were fully addressed, and for this reason I am planning to raise my score to 3. Nevertheless, as expressed above, my concerns regarding the novelty of this submission and the inability to verify the assumption used in the convergence analysis prevent me from recommending this work for acceptance.

---

> > > ### Author Response · Authors · 2026-04-05
> > >
> > > We sincerely thank reviewer m8HU for the increased score and for the continued constructive engagement with our work. We genuinely appreciate the time and effort the reviewer has devoted to providing detailed and insightful feedback throughout this discussion. Below, we provide further clarifications and discussions on the two remaining concerns.
> > >
> > >
> > > **Regarding C2 & Q2 (Restrictiveness of Assumption 2):** We sincerely appreciate the reviewer's understanding of our points, and we acknowledge that the practical verification of Assumption 2 (particularly the Jacobian approximation error) remains challenging. We agree that this is a limitation of our current theoretical framework. However, we respectfully believe that our analysis still represents a meaningful contribution **which advance the state-of-the-art reparameterization methods**. Prior work (Liu et al.) requires exact homeomorphisms and is limited to convex constraints, whereas our work explicitly incorporates INN approximation errors for non-convex constraint sets and quantifies their impact on both feasibility and optimality. To the best of our knowledge, this provides the first convergence guarantees for learned homeomorphism-based optimization over non-convex constraints, which we hope offers valuable theoretical insights for future research in this direction.
> > >
> > > **Regarding C3 (Usefulness of Complexity Analysis):** We fully agree with the reviewer that in practice, empirical justification is perhaps even more important than theoretical complexity analysis. We also appreciate the reviewer's valid point that a formal complexity bound cannot be rigorously established solely by observing that "using this layer width works well."
> > >
> > > That said, we respectfully believe that the complexity analysis still serves a necessary and complementary role. Complexity analysis is essential for understanding the scalability of optimization algorithms, particularly in the context of constrained optimization where problem dimensions can grow significantly. Without such analysis, practitioners lack theoretical guidance on whether a method is suitable for large-scale applications. Even though the complexity depends on the architectural choice of the INN, providing this analysis offers valuable insights into the computational efficiency of our approach and helps establish a more complete picture of the method's properties.
> > >
> > > We are deeply grateful for the reviewer's insightful suggestions and constructive feedback throughout this review process. We will integrate these discussions into our final version to improve the clarity and completeness of our presentation.

---

### Decision · Program_Chairs · 2026-04-30

**Decision:**

Accept (regular)

**Comment:**

This paper provides an interesting method for fast solution of nonconvex optimization problems, using learned homeomorphisms between non-convex constraint sets and the unit ball. The presented method meaningfully advances the prior art by embedding learned homeomorphisms within first-order optimization and by addressing nonconvex problems (though, per the reviewer discussion, the relationship to related prior work should be made much more explicit). The method also shows strong experimental results. While reviewers expressed concern that the assumptions made within the theory are overly strict and unverifiable, and that the bounds are loose, this concern about the theory was not deemed sufficient to override the methodological and experimental contributions of this work. As such, I recommend the paper be accepted.